# Spatially organized cellular communities form the developing human heart

Elie N. Farah[1,11], Robert K. Hu[1,11], Colin Kern[2], Qingquan Zhang[1], Ting-Yu Lu[3], Qixuan Ma[1], Shaina Tran[1], Bo Zhang[1,4], Daniel Carlin[1], Alexander Monell[2], Andrew P. Blair[1], Zilu Wang[1], Jacqueline Eschbach[2], Bin Li[5], Eugin Destici[1], Bing Ren[2,5,6,7], Sylvia M. Evans[1,8], Shaochen Chen[3,4,9,10], Quan Zhu[2 ✉] & Neil C. Chi[1,4,7,10 ✉]

The heart, which is the first organ to develop, is highly dependent on its form to function[1,2]. However, how diverse cardiac cell types spatially coordinate to create the complex morphological structures that are crucial for heart function remains unclear. Here we integrated single-cell RNA-sequencing with high-resolution multiplexed error-robust fluorescence in situ hybridization to resolve the identity of the cardiac cell types that develop the human heart. This approach also provided a spatial mapping of individual cells that enables illumination of their organization into cellular communities that form distinct cardiac structures. We discovered that many of these cardiac cell types further specified into subpopulations exclusive to specific communities, which support their specialization according to the cellular ecosystem and anatomical region. In particular, ventricular cardiomyocyte subpopulations displayed an unexpected complex laminar organization across the ventricular wall and formed, with other cell subpopulations, several cellular communities. Interrogating cell–cell interactions within these communities using in vivo conditional genetic mouse models and in vitro human pluripotent stem cell systems revealed multicellular signalling pathways that orchestrate the spatial organization of cardiac cell subpopulations during ventricular wall morphogenesis. These detailed findings into the cellular social interactions and specialization of cardiac cell types constructing and remodelling the human heart offer new insights into structural heart diseases and the engineering of complex multicellular tissues for human heart repair.

The human heart comprises complex cardiac structures that are crucial for its function[1,2]. Disruption of these structures can lead to congenital heart disease, the most common birth defect, and adult structural heart diseases such as hypertrophic cardiomyopathies and valvulopathies[3–5]. However, the cell types that create the human heart and, more importantly, how they interact and organize to form and maintain functional cardiac structures remain to be fully defined. Thus, to investigate the cooperative cellular interactions that direct heart morphogenesis, we performed comprehensive single-cell RNA sequencing (scRNA-seq) and multiplexed error-robust fluorescence in situ hybridization (MERFISH) of entire developing human hearts[6–8]. This strategy combines the power of single-cell transcriptomics with spatial biology to analyse, visualize and count RNA transcripts from hundreds to thousands of genes in individual cells. Integrative multimodal analysis of scRNA-seq transcriptomics and MERFISH-based imaging spatial information revealed the molecular and spatial identification of a broad range of cell lineages that organize into cellular communities to create distinct structures of the human heart, including previously uncharacterized

cardiac cell populations. This approach also revealed the signalling pathways that coordinate interactions between the cardiac cell populations that form such structures. Examining the crosstalk between specific combinations of cell populations within these communities revealed differential signalling pathways, including plexin–semaphorin (PLXN–SEMA), that direct multicellular interactions during ventricular wall morphogenesis. Overall, our findings provide a high-resolution single-cell molecular and spatial cardiac cell atlas that details the social interactions among distinct cell types that specialize and organize into cardiac structures that are crucial for maintaining heart function.

## Cell lineages in the developing human heart

To examine how diverse cardiovascular cell types coordinate to form complex structures that are vital for regulating human heart function, we initially investigated and identified the specific cell lineages that constitute the developing human heart. To this end, scRNA-seq was performed and analysed in replicate on human hearts between

[1]Department of Medicine, Division of Cardiology, University of California San Diego, La Jolla, CA, USA. [2]Center for Epigenomics, Department of Cellular and Molecular Medicine, University of California San Diego, La Jolla, CA, USA. [3]Materials Science and Engineering Program, University of California San Diego, La Jolla, CA, USA. [4]Department of Bioengineering, University of California San Diego, La Jolla, CA, USA. [5]Department of Cellular and Molecular Medicine, School of Medicine, University of California San Diego, La Jolla, CA, USA. [6]Ludwig Institute for Cancer Research, La Jolla, CA, USA. [7]Institute for Genomic Medicine, University of California San Diego, La Jolla, CA, USA. [8]Department of Pharmacology, Skaggs School of Pharmacy and Pharmaceutical Sciences, University of California San Diego, La Jolla, CA, USA. [9]Department of NanoEngineering, University of California San Diego, La Jolla, CA, USA. [10]Institute of Engineering in Medicine, University of California San Diego, La Jolla, CA, USA. [11]These authors contributed equally: Elie N. Farah, Robert K. Hu. ✉e-mail: quzhu@health.ucsd.edu; nchi@health.ucsd.edu

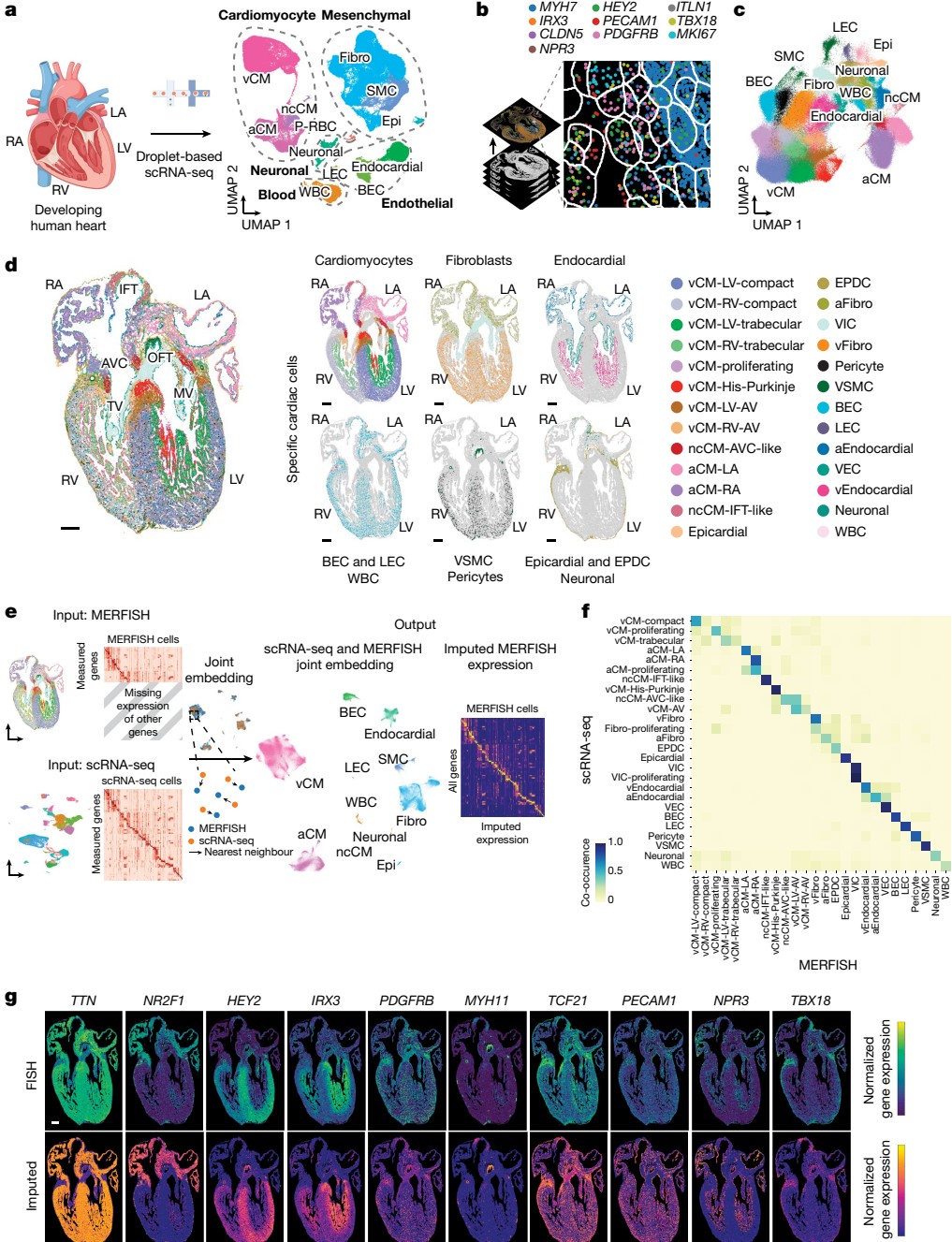

**Fig. 1 | Molecular and spatial human heart cell atlases reveal a diverse range of cell populations during heart development. a**, Left, schematic of experiment. Right, scRNA-seq identifies a diverse range of distinct cardiac cells that create the developing human heart as displayed by uniform manifold approximation and projection (UMAP) of ~143,000 cells. **b**, Schematic shows how 238 cardiac-cell-specific genes were spatially identified using MERFISH. Pseudo-coloured dots mark the location of individual molecules of ten specific RNA transcripts. **c**, Approximately 250,000 MERFISH-identified cardiac cells were clustered into specific cell populations as shown by UMAP and coloured accordingly in **d**. **d**, Identified MERFISH cells were spatially mapped across a frontal section of a 13 p.c.w. heart (left) and shown according to major cell classes (right). **e**, Joint embedding between MERFISH and age-matched scRNA-seq datasets enabled cell label transfer and MERFISH gene imputation. **f**, Co-occurrence heatmap shows the correspondence of cell annotations of MERFISH cells to those transferred from the 13 p.c.w. scRNA-seq dataset. **g**, Gene imputation performance was validated spatially by comparing normalized gene expression profiles of marker genes measured by MERFISH with the corresponding imputed gene expression profiles. Epi, epicardial; MV, mitral valve; P–RBC, platelet–red blood cell; TV, tricuspid valve. Scale bar, 250 μm (**g**). Illustration in **a** was created using BioRender (https://www.biorender.com).

9 and 16 post conception weeks (p.c.w.) (Fig. 1a, Supplementary Fig. 1a and Supplementary Table 1). Because these developing hearts were substantially smaller than adult human hearts, each collected heart was dissected into intact cardiac chambers and the interventricular septum (IVS) to increase the likelihood of identifying more cell types or states (including rare cell populations) by scRNA-seq, especially in underrepresented regions such as the atria (Supplementary Fig. 1a). Consequently, 142,946 single cells collected from these cardiac samples were analysed by scRNA-seq and were transcriptionally segregated into the following five distinct cell compartments: cardiomyocyte,

mesenchymal, endothelial, blood and neuronal (Fig. 1a). Graph-based clustering and gene marker analysis identified 12 major cell classes within the cell compartments (Fig. 1a, Supplementary Fig. 1b–d and Supplementary Table 2). Further clustering of cells from these compartments identified 39 populations that subdivided into 75 subpopulations that were assessed for their accuracy (Extended Data Fig. 1, Supplementary Figs. 2–7 and Supplementary Table 3). The identified cell lineages exhibited cellular heterogeneity that frequently corresponded to their anatomical location (atrial and ventricular cardiomyocytes, fibroblasts and endocardial cells) and developmental stage, thus providing new insights into the developmental complexity of these cells (Supplementary Figs. 3–7 and Supplementary Note 1). In summary, our single-cell analyses of the entire human heart provide a comprehensive cell atlas of the developmental heart as well as additional developmental insights for a multitude of common and rare cell types that create the human heart (Fig. 1a, Extended Data Fig. 1, Supplementary Figs. 3–12, Supplementary Tables 4–6 and Supplementary Note 2). However, how these cells interact and organize into complex morphological communities or structures crucial for heart function and cell specialization remains to be illuminated.

## MERFISH spatially maps heart cells

We next explored the interactive cellular mechanisms that direct cardiac morphogenesis and remodelling, including development of the ventricular wall. We applied MERFISH imaging[6–8] to interrogate the spatial organization of cardiovascular cells identified by scRNA-seq during a developmental time period when the ventricular wall undergoes dynamic remodelling, particularly myocardial wall compaction[9]. A list of 238 cell-subpopulation-specific genes was identified after applying a NS-Forest2 classifier on our scRNA-seq clustering analysis (Extended Data Fig. 1 and Supplementary Tables 7 and 8). These candidate genes were used to re-identify our classified cell subpopulations with an accuracy that was comparable to that of genes discovered using the Spapros classifier (Supplementary Tables 9 and 10). In particular, these 238 target genes were selected on the basis of previously reported cell lineage marker genes[10–15] and of differential or specific gene expression of more refined subpopulations identified by scRNA-seq. Using MERFISH-encoding probes designed for these selected genes (Supplementary Table 11), we performed MERFISH studies of coronal slices of 12–13 p.c.w. human hearts, which captured major cardiac structures (Fig. 1b–d). After cell segmentation and adaptive filtering, we obtained 108.2 million transcripts from 258,237 cells across three experiments (Extended Data Fig. 2a and Supplementary Table 12). On average, 365 transcripts from 85 genes per cell were detected from this analysis, whereas only 208 transcripts from 51 genes per cell were discovered by scRNA-seq using the same target gene list, a result that highlights the high RNA capture efficiency of MERFISH[6–8] (Supplementary Table 13). Additionally, the levels of RNA transcripts identified by each MERFISH experiment showed high correlation between three experimental replicates (Extended Data Fig. 2b) and our scRNA-seq datasets (Extended Data Fig. 2c). Furthermore, these imaged MERFISH genes displayed similar spatial expression patterns to those imaged using single-molecule fluorescence in situ hybridization (smFISH) (Extended Data Fig. 2d).

To identify specific cell populations from these MERFISH studies, a semi-supervised, graph-based clustering method was applied to MERFISH single-cell expression data (Fig. 1c). Cardiac gene marker analysis of this clustering revealed 27 distinct MERFISH cell populations that grouped into cell classes closely correlating with the developmental classes discovered by scRNA-seq, except for platelets–red blood cells, which is probably because of the exclusion of their marker genes from the MERFISH gene library (Fig. 1c,d, Extended Data Fig. 2e,f and Supplementary Table 14). The relative number of cells differed for some classes between MERFISH and scRNA-seq datasets (Extended Data Fig. 2g), which may be due to differences between cell capture or transcription

detection between the two methods, as previously proposed[7]. However, integration of our scRNA-seq and MERFISH datasets revealed a strong correspondence of related cells between the datasets, which facilitated the imputation and spatial mapping of additional genes beyond those examined by MERFISH (Fig. 1e–g, Supplementary Fig. 13, Supplementary Table 15 and Supplementary Note 3).

## Diverse cardiomyocytes in specific heart structures

In line with our scRNA-seq data, cardiomyocyte lineages represented the largest proportion of cells identified from our MERFISH analyses (12 out of 27 populations) (Fig. 1d and Extended Data Figs. 3 and 4). Spatial mapping of these transcripts revealed that the identified cardiomyocytes displayed distinct regional and structural distributions across the heart, corroborating our scRNA-seq regional findings (Fig. 1d and Extended Data Fig. 4a). In contrast to recent scRNA-seq and in situ RNA-seq studies of the heart[10–18], these MERFISH results provided high-resolution spatial imaging that enabled the definition of cells at finer resolution and the tracking of individual cells to detailed structures of the heart (Fig. 1d, Extended Data Figs. 3 and 4 and Supplementary Figs. 3 and 14). As a result, these cardiomyocytes were observed to populate distinct anatomical domains of chambered and non-chambered regions of the heart and were frequently spatially distinct from each other (Fig. 1d and Extended Data Fig. 4a).

Chambered cardiomyocytes were broadly divided into *NR2F1*+ and *IRX4*+ cardiomyocytes that contributed mutually exclusively to the atrial and ventricular chambers, respectively (Fig. 1d and Extended Data Figs. 2e and 4a). Atrial cardiomyocytes (aCMs) spatially segregated into those residing in the left atria (LA) and right atria (RA) (aCM-LA and aCM-RA, respectively), which were transcriptionally distinguished by *ANGPT1* as observed in our scRNA-seq analyses (Extended Data Figs. 2e and 4a and Supplementary Fig. 3d,e). By contrast, ventricular cardiomyocytes (vCMs) displayed more cellular complexity and subdivided into those that specifically occupied not only the left ventricle (LV) and right ventricle (RV) but also more distinct anatomical subdomains within the outer and inner layers of the ventricles (Fig. 1d and Extended Data Figs. 3 and 4a). Both known and new markers were enriched in these vCMs, including *SLC1A3* and *PRRX1*, which were expressed in the left vCMs (vCM-LV) and right vCMs (vCM-RV), respectively, and *HEY2* and *IRX3*, which marked outer and inner layer vCMs, thus resolving them as compact (vCM-LV-compact and vCM-RV-compact) and trabecular (vCM-LV-trabecular and vCM-RV-trabecular) cardiomyocytes, respectively[19–21] (Extended Data Fig. 2e). Within the inner ventricle layer, we discovered an additional cardiomyocyte not defined by scRNA-seq analyses that extended along the luminal portion of the ventricle to the atrioventricular canal (AVC). This specific cardiomyocyte type expressed *IRX3*, *TBX3* and *HCN4*, which are known markers of the His-Purkinje fast cardiac conduction system of the ventricle[22], as well as *IRX1* and *IRX2*, which have been observed along the subendocardial layer of the IVS of mouse hearts[21] (Fig. 1d, Extended Data Figs. 2e, 3 and 4a and Supplementary Fig. 3). Although most vCMs were observed in specific regions of the ventricle, we discovered a vCM population (vCM-proliferating) that was present throughout the ventricle and displayed moderate expression of proliferative markers but diffuse expression of cardiac structure-specific genes (Fig. 1d and Extended Data Figs. 2e, 3 and 4a), suggesting that these cardiomyocytes may be progenitor-like with the capacity to specialize within specific cardiac structures.

Although our scRNA-seq analyses uncovered cardiomyocytes (such as *BMP2*+ non-chambered cardiomyocytes (ncCMs)) beyond those reported in atrial and ventricular chambers[10–14,16,17,23], our MERFISH analyses resolved and confirmed the identification of these relatively rare but diverse specialized cardiomyocytes. In particular, inflow tract/pacemaker (ncCM-IFT-like) cardiomyocytes, which expressed the known inflow tract developmental transcriptional regulators *ISL1*

and *SHOX2*, were observed above the RA where the sinoatrial node (SAN) pacemaker[22] has been reported. Meanwhile, atrioventricular canal/node (ncCM-AVC-like) cardiomyocytes, which regulate AVC and atrioventricular node (AVN) development and co-exress *TBX3* and *RSPO3*[22,24], were located within the inner portion of the AVC (Fig. 1d and Extended Data Figs. 2e and 4a). Additionally, a population of *CNN1*+*CRABP2*+ cardiomyocytes that was also identified in our scRNA-seq analysis (Supplementary Fig. 3d,e), but not well defined, was spatially resolved and observed within the atrioventricular (AV) valve leaflets (vCM-LV/RV-AV) (Fig. 1d and Extended Data Figs. 2e and 4a). These cardiomyocytes further subdivided to those populating the tricuspid and mitral valves of the right ventricle (vCM-RV-AV) and left ventricle (vCM-LV-AV), respectively, suggesting that these cardiomyocytes, which have not been well-defined in mouse or human hearts[25], may exhibit functional differences between these valves (Fig. 1d and Extended Data Figs. 2e and 4a).

## Spatial relationships of heart lineages

Although they displayed less diversity than cardiomyocytes, MERFISH imaging revealed that non-cardiomyocyte cells, particularly those endogenous to the heart, also segregated and contributed to specific regions or structures of the heart, a result that supports similar observations from our scRNA-seq analysis (Fig. 1d and Extended Data Figs. 2e, 3 and 4b–e). However, MERFISH analyses provided detailed spatial information at single-cell resolution that resolved the identities of other less well-defined cells by scRNA-seq (Fig. 1d, Extended Data Figs. 2e, 3 and 4b–d and Supplementary Figs. 4–6). For the fibroblast-like class, we observed distinct *PDGFRA*+*TCF21*+ fibroblasts that populated specifically either the atria (aFibro) or the ventricles (vFibro), which expressed *TNC* and *HHIP*, respectively. We also observed *PENK*+ valvular interstitial cells (VICs) that contributed to the cardiac valves, and adventitial fibroblasts (adFibro) that contributed to the outflow tract (Fig. 1d and Extended Data Figs. 2e, 3 and 4b). Similarly, we discovered three distinct *LEPR*+ endocardial cells that particularly lined the luminal surfaces of the atria (aEndocardial), ventricle (vEndocardial) or cardiac valves (valve endocardial cells (VECs)) and could be molecularly distinguished by their expression of *SHISA3*+, *NSG1*+*COL26A1*− and *NSG1*+*COL26A1*+ genes, respectively (Fig. 1d and Extended Data Figs. 2e, 3 and 4c). Vascular-related cells, including *CLDN5*+*LYVE1*− blood endothelial cells (BECs), *MYH11*+ vascular smooth muscle cells (VSMCs) and *KCNJ8*+ pericytes, were distributed throughout the ventricle, revealing blood vessels, but less so within the atria, suggesting that the atria may be less vascularized, which may be due to its thinner myocardial walls (Fig. 1d and Extended Data Figs. 2e, 3 and 4b,c). Conversely, *PRPH*+ neuronal cells were observed in the outflow tract and atria, particularly near the inflow tract, a result consistent with their role in outflow tract development and innervation of the venous pole of the heart[26] (Fig. 1d and Extended Data Figs. 2e, 3 and 4d). *CLDN5*+*LYVE1*+*PROX1*+ lymphatic endothelial cells (LECs), *MOXD1*+*MMP11*+ epicardium-derived progenitor cells (EPDCs) and *ITLN1*+ epicardial cells were localized on the surface of the heart, and EPDCs were enriched within the AVC regions, as previously reported[10] (Fig. 1d and Extended Data Figs. 2e, 3 and 4b,c). Finally, many of these non-cardiomyocyte cells exclusively co-localized with each other and with corresponding cardiomyocyte counterparts within distinct cardiac regions. This finding suggests that they may assemble into cellular communities that not only influence their specialized cellular functionalization but also form anatomical structures crucial for regulating overall cardiac function (Fig. 1d and Extended Data Figs. 2e, 3 and 4).

## Cell communities form cardiac structures

We next sought to understand how specific cardiovascular cells may assemble into cellular neighbourhoods that form organized multi-cell lineage structures crucial for heart function. To this end, we identified regions of the heart that were spatially composed of distinct combinations of co-segregating cell populations ('cellular communities' (CCs)). Cellular neighbours for each cell of the heart were defined within a 150 μm radius, which represents a typical diffusion 'zone' for extracellular signalling molecules from an individual cell (Fig. 2a, cell zone, and Methods). Approximately 250,000 cell zones were identified and grouped into 13 distinct CCs on the basis of the cell composition for these cell zones (Fig. 2b and Extended Data Fig. 5a–c). These detected CCs, which were mapped to the developing heart, corresponded to and defined specific architectures of the heart at high single-cell granularity, including known and less familiar cardiac structures (Fig. 2b,c and Extended Data Fig. 5c). Each CC was composed of distinct combinations and amounts of specific cells and displayed a broad range of cellular complexity and purity. For example, some CCs contained only one or two cell populations, whereas others comprised more than ten cell populations (Fig. 2d–f and Extended Data Fig. 5d,e).

Consistent with the overall greater cellular complexity observed in the ventricular chamber compared with the atrial chamber (Fig. 2e,f and Extended Data Fig. 5d), five CCs were located in the ventricle and two in the atria (Fig. 2c and Extended Data Fig. 5c). The two atrial CCs corresponded to the left and right atria and consisted of respective left and right aCMs, aFibro cells, aEndocardial cells, epicardial cells and neuronal cells (Fig. 2c,d). By contrast, the ventricular chambers were divided into five CCs that correlated to layers of the left and right ventricular walls (outer, inner and the ventricular conduction system (VCS); Fig. 2c,d), which exhibited decreasing cellular complexity from the outside to the lumen of the ventricle (Fig. 2f and Extended Data Fig. 5d). The outer left and right ventricular CCs consisted of a broad range of cells and were enriched for the following cell types: left and right compact vCMs specific for each respective chamber, vFibro cells, vascular cells, including BECs and pericytes that form the coronary vasculature[26], and proliferating vCMs, which probably account for the increased growth rate of the outer layer of the ventricle[27] (Fig. 2d). On the other hand, the inner left and right ventricular CCs were composed of left and right trabecular vCMs, respectively, and vEndocardial cells, but substantially fewer vFibro cells, proliferating vCMs and vascular cells (Fig. 2d). Finally, a VCS CC that was luminal to the inner ventricular CCs and consisted primarily of His-Purkinje cardiomyocytes and fewer vEndocardial cells was present more predominantly in the left ventricle than the right ventricle (Fig. 2c,d and Extended Data Fig. 5c). This result provides support for the finding that the fast His-Purkinje cardiac conduction system of the ventricle may initially develop within the left ventricle[28].

In addition to the seven CCs within the cardiac chambers, we discovered six non-cardiac chamber CCs for the heart (Fig. 2b–d and Extended Data Fig. 5c). These communities corresponded with non-chambered cardiac regions of the heart, including the inflow tract (IFT) and outflow tract (OFT) and AV regions of the heart (Fig. 2c,d and Extended Data Fig. 5c). The IFT/SAN CC consisted mainly of ncCM-IFT-like and neuronal cells and may correspond to the SAN pacemaker of heart (Fig. 2c,d), whereas the OFT CC was enriched for VSMCs, adFibro cells and endothelial-related cells, consistent with cell lineages of the aorta[26] (Fig. 2c,d). Conversely, the AV region, which exhibited higher cellular complexity, was composed of four CCs within a small region between the atrial and ventricular chambers that includes the AVC and cardiac valves (Fig. 2c–f and Extended Data Fig. 5c,d). The outer portion of the AVC contained the subepicardial CC comprising EPDCs, VSMCs, LECs, neuronal cells and white blood cells (WBCs) (Fig. 2c,d). By contrast, the inner portion of the AVC, which circumscribes the cardiac valves, consisted of the AVN/AV ring CC composed of ncCM-AVC-like cardiomyocytes along with aFibro cells, and may represent a developmental structure that forms the AVN[29] (Fig. 2c,d and Extended Data Fig. 5c). Furthermore, two CCs were discovered within the cardiac valves, including the valve CC, which was composed of valve-specific cardiomyocytes,

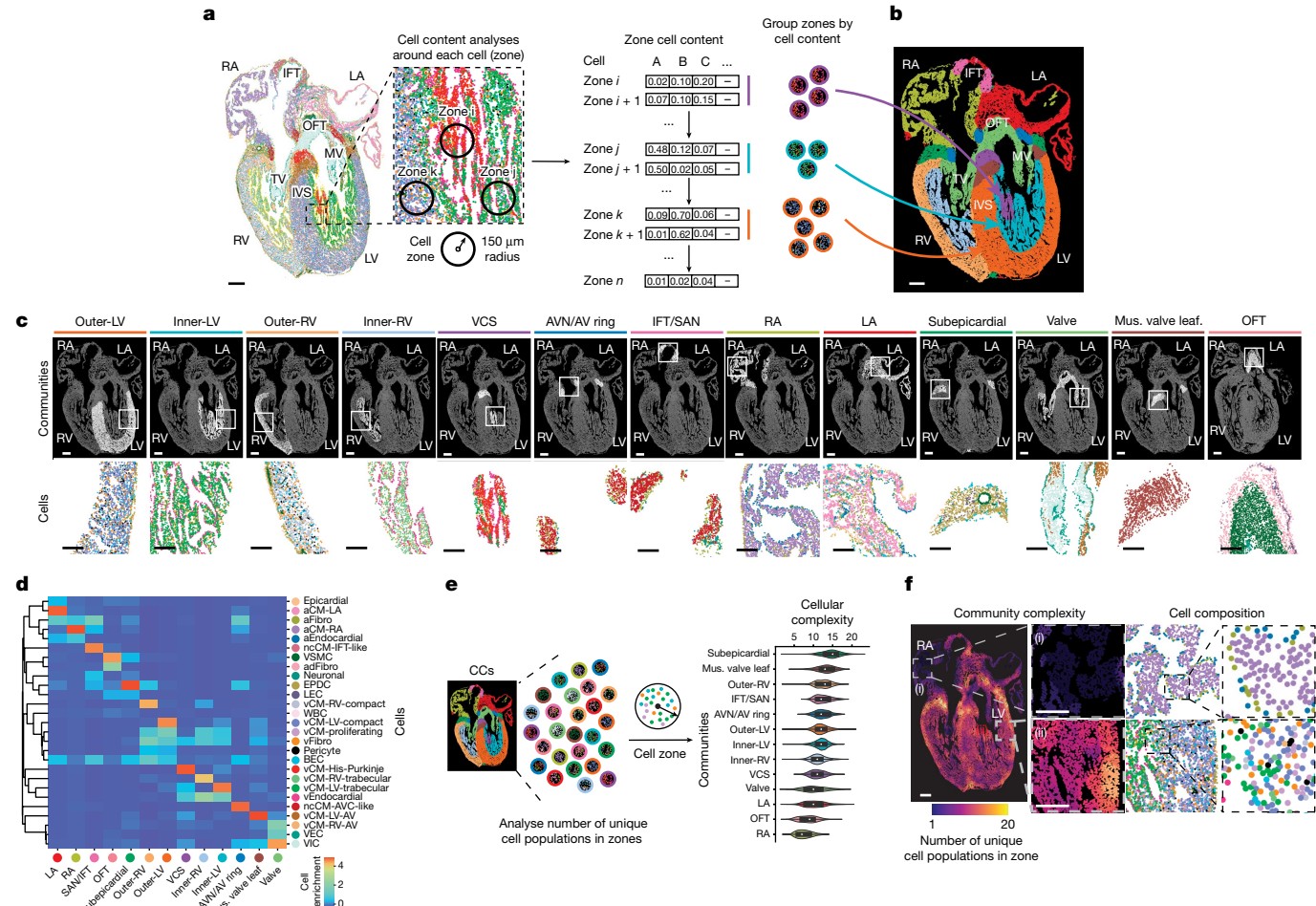

**Fig. 2 | Distinct cardiac cell populations spatially organize into CCs that form specialized cardiac structures. a**, Interrogation of the cell content around each individual cell identified cell zones or neighbourhoods, which formed defined CCs. **b**, Spatial mapping of CCs onto 13 p.c.w. hearts revealed their correspondence to distinct anatomical cardiac structures. **c**, The spatial location of each CC is displayed along with examples of their cellular composition and distribution (insets). **d**, Heatmap shows the composition of identified MERFISH cells within each defined CC. **e,f**, Analysis of the number of unique cell populations within each zone reveals the cellular complexity of each CC and cardiac region as displayed quantitatively (**e**, violin plot) and spatially (**f**, spatial complexity map). For **e**, the centre white dot represents the median, the bold black line represents the interquartile range, and the edges define minima and maxima of the distribution. Boxed areas in the spatial complexity map show regions of low (i) and high (ii) complexity. Insets (middle show the respective cellular composition, and magnified insets (right) show distinct identified cells). Mus. valve leaf., muscular valve leaflet. Scale bar, 250 μm (**b,c,f**).

endothelial cells and interstitial cells extending to both the AV and OFT valves, and a more specific muscular valve leaflet CC within the mitral valve region that was enriched for vCM-LV-AV and may reflect an earlier specialization of cells within the left ventricle (Fig. 2c,d and Extended Data Fig. 5c).

Overall, these analyses reveal at high cellular resolution how diverse cardiac cells, including those that are broadly present or more specialized, may assemble into CCs that form morphological structures of the heart. These cardiac CCs displayed not only distinct combinations of cells and cellular complexity, which may lead to functional differences among CCs and cardiac structures, but also distinct cardiac cells that frequently were enriched in specific CCs, thus supporting the idea that cardiac cells may adopt cellular specialization based on their environment and role in each community or cardiac structure (Fig. 2 and Extended Data Fig. 5).

## Multilayered organization of ventricles

Our CC analyses revealed that the developing ventricular chamber displays both high cellular complexity and low purity, particularly at the border regions of ventricular wall CCs (Fig. 2e,f and Extended Data Fig. 5d,e). This result suggested that the developing ventricular chamber may exhibit more distinct cardiac cells and complex organization than previously described[30,31]. Consistent with these findings, the ventricle exhibited intermixing of compact and trabecular vCMs at the interface between the outer and inner ventricular communities (Fig. 2f, inset of (ii)), suggesting that regions of the developing heart are dynamically remodelling, including compaction of the ventricular wall[9].

To explore this ventricular cellular and organizational complexity, MERFISH cells within only the ventricles were isolated, identified and spatially mapped to the ventricle (Fig. 3a,b and Extended Data Figs. 6a and 7). Applying gene marker analysis and spatial information to these distinct cells revealed additional populations of cardiomyocytes and fibroblasts (Fig. 3b,c, Extended Data Figs. 6a and 7 and Supplementary Table 16). In particular, the eight vCM populations initially identified by MERFISH (Fig. 1d and Extended Data Fig. 4a) were subdivided into 13 vCM subpopulations, including 11 chambered and 2 non-chambered subpopulations (Fig. 3b,c and Extended Data Figs. 6a and 7a). Whereas the non-chambered vCMs consisted of the *CNN1*+*CRABP2*+ atrioventricular valve leaflet CMs (vCM-LV/RV-AV) observed within the mitral

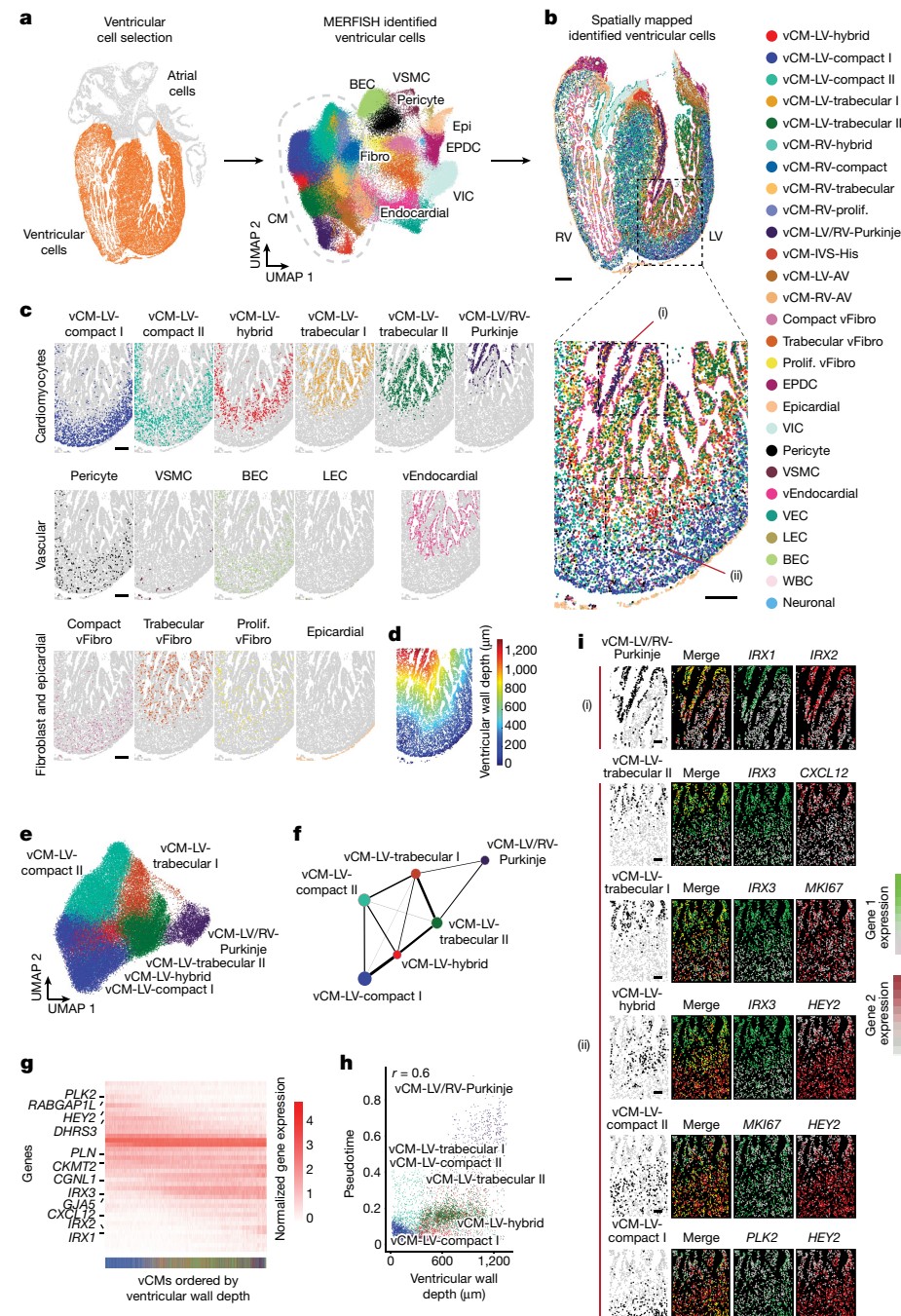

**Fig. 3 | The ventricular wall contains distinct specialized cardiac cells spatially organized into unexpected complex laminar layers. a**, MERFISH cells that constitute the ventricles (left, orange) were clustered as displayed using UMAP (right). **b**, Identified ventricular cells were spatially mapped in 13 p.c.w. ventricles. **c**, The spatial distributions of specific ventricular cells are shown for the left ventricular wall from the region outlined in the MERFISH spatial map in **b**. **d**, The ventricular wall depth distribution of ventricular cells is shown as a measured distance from the epicardial/outer surface of the ventricle for the imaged region in **b**. **e**, LV vCMs segregated into distinct vCM subpopulations. **f**, The molecular relationship of distinct vCMs is displayed in a connectivity map in which weighted edges between nodes represent their connectivity based on gene expression similarity. **g**, Heatmap shows the normalized expression of differentially expressed genes for vCMs as ordered by increasing ventricular wall depth. The coloured bar at the bottom indicates the specific vCMs as denoted in **b**. **h**, Scatter plot reveals the relationship between ventricular wall depth and pseudotime for individual vCMs in the left ventricle. **i**, MERFISH images of outlined regions in **c** ((i) and (ii)) show that specific combinations of gene markers, as shown in green and red, spatially identified specific vCMs. Scale bar, 250 μm.

and tricuspid valves as described above (Figs. 1d and 3b and Extended Data Figs. 2e, 6a and 7a), the chambered vCMs resolved into the following subpopulations: three *HEY2*[+] compact vCMs (vCM-LV-compact I, vCM-LV-compact II and vCM-RV-compact); three *IRX3*[+] trabecular vCMs (vCM-LV-trabecular I, vCM-LV-trabecular II and vCM-RV-trabecular);

two *HEY2*[+]*IRX3*[+] hybrid vCMs (vCM-LV-hybrid, vCM-RV-hybrid) that co-expressed compact and trabecular vCM genes; two *TBX3*[+] His-Purkinje CMs (vCM-IVS-His and vCM-LV/RV-Purkinje); and a right ventricular vCM (vCM-RV-proliferating) that displayed proliferative markers similar to vCM-LV-compact II and vCM-LV-trabecular I (Fig. 3b,c

and Extended Data Figs. 6a and 7a). Spatial mapping revealed that many of these vCMs were organized in a laminar distribution across the ventricular wall according to the chamber wall depth, with more vCMs and layers observed in the left than right ventricle, a result that supports the finding that the left ventricle develops earlier than the right ventricle[32] (Fig. 3c,d and Extended Data Fig. 7a). Although identified vCMs appeared in distinct layers within the ventricular wall, the ventricular fibroblasts subdivided into three subpopulations. These included a proliferative-like ventricular fibroblast that was observed throughout the ventricle and expressed mitotic markers, and two fibroblasts (compact vFibro and trabecular vFibro) more specifically located in the outer and inner regions of the ventricle, where compact and trabecular vCMs are enriched, respectively (Fig. 3b,c and Extended Data Figs. 6a and 7d). This proliferative-like vFibro (proliferating vFibro) expressed genes common to both compact and trabecular vFibro cells but at lower levels, suggesting that these fibroblasts may be progenitors that can supply differentiated compact and trabecular vFibro cells to respective regions of the ventricle (Extended Data Fig. 6a).

To understand the complexity and laminar organization of vCMs of the left ventricular wall, which displayed greater cellular complexity than that of the right ventricle (Extended Data Fig. 7a), we examined the gene expression profiles for these vCMs and their spatial distribution across the left ventricle. Consistent with the gradual transition of gene expression profiles among vCMs of the left ventricular wall (Fig. 3e), molecular connectivity analysis revealed that these vCMs exhibited a highly connected gene expression network, with the strongest connections existing between vCMs that were spatially contiguous (Fig. 3f). In support of the notion that neighbouring vCMs display high similarity in gene expression and may span a continuous spatial and molecular landscape, a progressive change in spatial gene expression for vCMs was observed along the ventricular wall depth. Moreover, the results spatially corresponded with the laminar organization and partial overlap of respective vCMs in the left ventricle (Fig. 3c,d,g and Extended Data Fig. 6b). In particular, we identified combinations of co-expressing genes that were enriched in specific vCMs and enabled their spatial tracking in the left ventricle (Fig. 3g,i and Extended Data Fig. 6b). These genes included not only known compact and ventricular markers such as *HEY2*, *IRX3* and *GJA5*, but also newly defined marker genes expressed by compact vCMs closer to the epicardium (*RABGAP1L* and *PLK2*) and trabecular vCMs nearer to the lumen (*CXCL12*) (Fig. 3g,i and Extended Data Fig. 6b). Confirming these findings, pseudotime analysis revealed that the order of vCMs correlated with their allocation along the ventricular wall (Fig. 3h and Extended Data Fig. 6c), and individual vCMs formed contiguous aggregates along both the pseudotime and ventricular wall depth axes (Fig. 3h).

To interrogate how these ventricular subpopulations may change with developmental age, we performed MERFISH on 15 p.c.w. ventricles (Extended Data Fig. 8a and Supplementary Table 17). Comparing ventricular subpopulations between 13 and 15 p.c.w. ventricles uncovered changes in the allocation of these cellular subpopulations, which included the absence of hybrid vCM subpopulations in 15 p.c.w. ventricles (Extended Data Fig. 8b–d). In support of these findings, we discovered in our scRNA-seq data that 13 p.c.w. hearts contained the highest proportion of *HEY2*+*IRX3*+ hybrid vCMs to total vCMs in the left ventricle compared with other developmental stages, which suggested that this cell population may be developmentally transient (Extended Data Fig. 8e), as suggested in mouse hearts[30]. Corresponding to this disappearance of hybrid vCMs, we observed that compact vCMs extended further across the ventricular wall depth, whereas trabecular vCMs appeared closer to the lumen of 15 p.c.w. ventricles compared with 13 p.c.w. ventricles (Extended Data Fig. 8f). Finally, comparisons of our vCM subpopulations with those from adult human hearts[33] revealed that non-failing and diseased adult vCMs are primarily compact myocardium/vCMs, as previously suggested[9] (Supplementary Figs. 15 and 16 and Supplementary Note 4). Taken together, these findings provide evidence to support the idea that developing vCMs adopt a complexity and gradient of distribution across the ventricular wall depth that correlates with the spatial expression of distinct gene profiles within the ventricular wall.

## Multicellular signalling forms ventricles

Defects in the development of the ventricular wall, particularly remodelling of the outer compact and inner trabecular layers, can lead to adult and congenital heart diseases, including left ventricular non-compaction cardiomyopathy[34] and hypoplastic left heart syndrome[3]. To understand how the human ventricle forms and organizes the myocardial layers pivotal for its development and function, we focused on investigating how distinct cardiac cells coordinate to guide ventricular wall morphogenesis at 12–13 p.c.w., a time point when human cardiac ventricles begin to refashion their walls through consolidation of the inner trabecular layer with the outer compact layer (myocardial compaction)[9]. On the basis of the distinct cardiac ventricular subpopulations discovered from our MERFISH ventricle analysis of 12–13 p.c.w. hearts, we defined ventricular CCs of spatially neighbouring ventricular cells to identify potentially interacting cells within the ventricles (Fig. 4a,b and Extended Data Fig. 9a–c). Although we observed that the right ventricular wall comprises three major CCs (outer, inner and VCS), we discovered that the LV subpopulations organized into four major CCs that include not only the outer-LV, inner-LV and VCS CCs, but also an intermediate-LV CC residing between the outer-LV and inner-LV CCs (Fig. 4a–c and Extended Data Fig. 9c). These ventricular CCs were spatially layered across the ventricular wall, similar to the laminar organization of vCMs, but disproportionately detected between the IVS and LV ventricular apical and free wall where the VCS and inner-LV CCs are enriched, respectively (Fig. 4a and Extended Data Fig. 9c). Consistent with the additional ventricular cardiac subpopulations identified in the LV, the LV-specific CCs exhibited increased cellular complexity (Extended Data Fig. 9d). Whereas the outer-LV and inner-LV CCs consisted correspondingly of compact and trabecular LV vCMs, the intermediate-LV CC contained these LV vCMs and hybrid vCMs, and displayed the greatest cellular heterogeneity but lowest cellular purity (Fig. 4a–c and Extended Data Fig. 9d,e). These findings support the notion that the LV, particularly its intermediate regions, may exhibit complex interactive multicellular events that regulate the dynamic development and remodelling of its ventricular wall.

To understand how these ventricular cardiac cells may cooperate to spatially transform the developing trabeculated ventricular layer into part of the mature functional compact ventricular wall, we interrogated cell–cell signalling events among spatially neighbouring cardiac cells using cell–cell interaction (CCI) analysis of MERFISH cardiac cells that were harmonized with age-matched scRNA-seq datasets (Fig. 1e, Supplementary Figs. 15 and 17 and Supplementary Tables 18 and 19). Consistent with its high cellular complexity, the intermediate-LV CC displayed the greatest number of ventricular cell and signalling interactions among the LV-specific CCs (outer, intermediate and inner), whereas the inner-LV CC exhibited the least (Supplementary Fig. 17a). Although outer-LV and inner-LV CCs displayed interactions between compact vCMs and compact vFibro cells, and trabecular vCMs and trabecular vFibro cells, respectively, combinatorial cross-interactions between these vCMs and vFibros were observed in the intermediate-LV CC, supporting the idea that the intermediate-LV CC may be a region of dynamic cellular developmental transformation (Supplementary Fig. 17).

Because of the highly layered organization of vCMs across the LV wall (Figs. 3c and 4c), we examined incoming signals to vCMs to identify signalling pathways that control their spatial distribution (Fig. 4d and Extended Data Fig. 10a–c). A wide range of ventricular cardiac cells was discovered to signal to distinct vCMs, with fibroblasts displaying the strongest and highest number of signalling interactions with vCMs

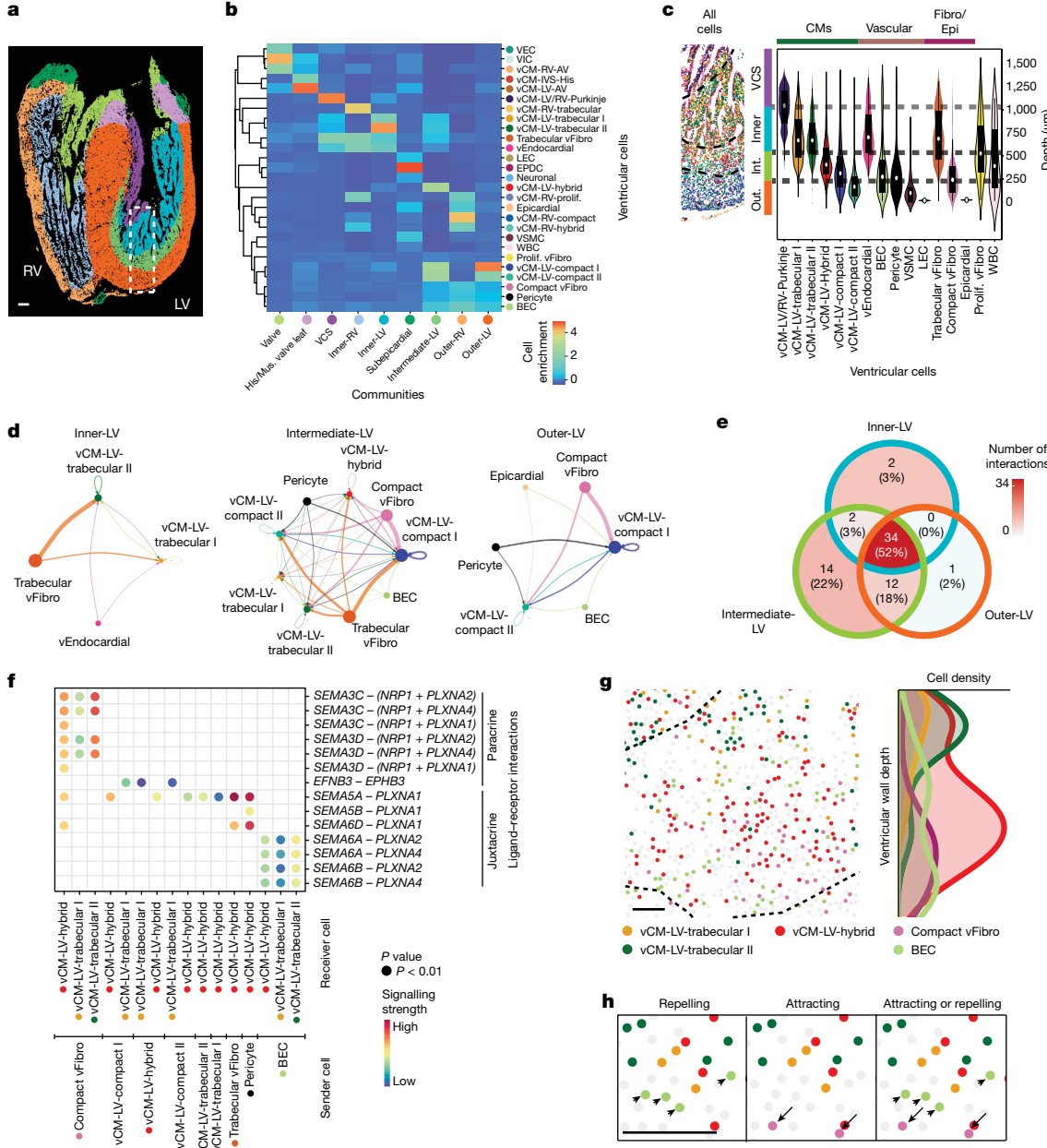

**Fig. 4 | Multicellular interactions direct the organization of specific CCs within the ventricular wall. a**, MERFISH-identified ventricular cells assembled into nine more refined CCs within the ventricle. **b**, Heatmap shows the composition of distinct ventricular cells within each ventricle CC. **c**, MERFISH image of the outlined area in **a** reveals CC layers and their cell composition. Violin plot shows the ventricular wall depth distributions for distinct ventricular cells within these layers. The centre white dot represents the median, the bold black line represents the interquartile range, and the edges define minima and maxima of the distribution. Dashed lines indicate boundaries for CC layers. **d**, Chord diagrams reveal the strength of cell–cell signalling interactions received by specific vCMs in the inner-LV, intermediate-LV and outer-LV CCs. The size of the node represents the number of cells for a distinct ventricular cell, and the width of the edge represents the interaction strength between pairs of specific ventricular cells. **e**, The Venn diagram shows the number of

specific and shared CCIs received by vCMs within the inner-LV, intermediate-LV and outer-LV communities. **f**, Dot plot shows specific signalling interactions between distinct ventricular cells within the intermediate-LV CC. **g**, Left, spatial map of cells participating in interactions between *SEMA3C*, *SEMA3D*, *SEMA6A* or *SEMA6B* with *PLXNA2* or *PLXN4* for the intermediate-LV CC. Right, normalized ventricular wall depth distribution of these cells is shown in the histogram. **h**, High-resolution spatial cell map of the intermediate-LV CC shows how cells involved in interactions with *SEMA3C*, *SEMA3D*, *SEMA6A* or *SEMA6B* with *PLXNA2* or *PLXN4* signalling may be spatially distributed to mediate attracting or repelling interactions. Arrows and arrowheads point to *SEMA3C*+*SEMA3D*+ compact vFibro cells and *SEMA6A*+*SEMA6B*+ BECs, respectively. Fibro/Epi, fibroblast and epicardial; His/mus. valve leaf., bundle of His and the muscular valve leaflet; Int., Intermediate; Out., Outer. Scale bars, 50 μm (**g**,**h**); 250 μm (**a**).

(Fig. 4d, Extended Data Fig. 10a–c and Supplementary Table 19). In line with these findings, the most predominant signalling pathways received by these vCMs were growth and extracellular-matrix-related pathways that were derived from fibroblasts across the LV CCs (Extended Data Fig. 10a and Supplementary Table 19), a finding that supports the idea

that fibroblasts may have a crucial role in the development of the LV wall[35]. Additionally, we identified CCIs differentially received by distinct vCMs among these communities (Fig. 4e and Extended Data Fig. 10b,c). For instance, neuregulin–ERBB signalling was observed between *NRG1*+ vEndocardial cells and *ERBB2*+*ERBB4*+ trabecular vCMs in the inner-LV

CC, as previously reported[31,36,37]. By contrast, the outer-LV CC exhibited pleiotrophin–syndecan growth hormone signalling between *PTN*⁺ compact vFibro and *SDC2*⁺ vCM-LV-compact II, which is consistent with the increased growth rate exhibited by the ventricular outer layer during this developmental period[27] (Extended Data Fig. 10c). Notably, we also discovered several PLXN–SEMA axon guidance signalling pathways, particularly within the intermediate-LV CC, that may mediate paracrine interactions between *PLXNA2*⁺*PLXNA4*⁺ hybrid and trabecular vCMs and *SEMA3C*⁺*SEMA3D*⁺ compact vFibro cells as well as juxtacrine interactions between *PLXNA2*⁺*PLXNA4*⁺ hybrid and trabecular vCMs and *SEMA6A*⁺*SEMA6B*⁺ BECs (Fig. 4f, Extended Data Fig. 10d).

Given the role of PLXN–SEMA signalling in regulating cell migration[38–40], this signalling pathway may mediate a complex multicellular interaction among cardiomyocytes, fibroblasts and endothelial cells that coordinate the organization of cardiomyocytes within the ventricular wall. Supporting the notion that they may participate in regulating the cells involved in remodelling the ventricular wall layers, including myocardial compaction, we observed that these cells were spatially arranged in a complementary but overlapping gradient across the LV wall, where they merged within the intermediate-LV CC to interact (Fig. 4g and Extended Data Fig. 11a,b). In particular, *PLXNA2*⁺*PLXNA4*⁺ trabecular vCMs and *SEMA3C*⁺*SEMA3D*⁺ compact vFibro cells were highest in the inner-LV CC and outer-LV CC, respectively, but progressively decreased in opposing directions along the wall depth such that these ventricular cells spatially intersected within the intermediate-LV CC (Fig. 4g,h and Extended Data Fig. 11a,b). By contrast, *SEMA6A*⁺*SEMA6B*⁺ BECs were observed throughout the intermediate-LV and outer-LV CC but tapered at the boundary between the intermediate-LV and inner-LV CCs where trabeculae exist (Fig. 4g and Extended Data Fig. 11a,b). Furthermore, *PLXNA2*⁺*PLXNA4*⁺ hybrid vCMs were mainly located in the intermediate-LV CC, with more observed at the outer half of the intermediate-LV CC (Fig. 4g and Extended Data Fig. 11a,b), suggesting that these vCMs may be transitioning between trabecular and compact vCMs during ventricular wall morphogenesis. Finally, we observed trabecular and hybrid vCMs in closer proximity to BECs than compact vFibro cells within the intermediate-LV CC, a finding that supports that there is juxtacrine and paracrine PLXN–SEMA signalling between these interacting ventricular cells (Fig. 4h and Extended Data Fig. 11c). Consistent with these cellular spatial findings, specific semaphorins and plexins for these ventricular cells generally exhibited a similar pattern of expression across the ventricular wall, as detected by virtual fluorescent in situ hybridization and confirmed by smFISH studies (Extended Data Fig. 11d,e).

## PLXN–SEMA directs ventricle organization

To explore whether PLXN–SEMA signalling pathways identified from our CCI studies participate in organizing vCMs within the ventricular wall, we used a rapid 3D bioprinting technique to create an in vitro human pluripotent stem cell (hPSC) vCM multilayer ventricular wall model[41] (hPSC-vCM) for investigating how *SEMA3C*, *SEMA3D*, *SEMA6A* and *SEMA6B* originating from the intermediate-LV CC may influence the spatial reallocation of *PLXNA2*⁺*PLXNA4*⁺ trabecular vCMs. To this end, we bioprinted enriched non-trabecular-like and trabecular-like hPSC-vCMs in layers to recapitulate the intermediate-LV CC and inner-LV trabecular CC regions of the human ventricle, respectively (Fig. 5a). Utilizing a monolayer cardiac differentiation system, we generated enriched hPSC cardiomyocytes (>90%), which were predominantly early developing *IRX4*⁺ vCMs, and were used for creating the bioprinted non-trabecular-like layers (Fig. 5a–c, Supplementary Fig. 18a–d and Supplementary Tables 20 and 21). To create trabecular-like hPSC-vCMs for bioprinting the inner-LV trabecular CC-like layer (Fig. 5a–c), hPSCs were differentiated into vCMs and then treated with neuregulin-1 (NRG1), which promotes trabecular vCM differentiation through NRG1–ERBB2–ERBB4 signalling between endocardial cells and vCMs[31,36,37], as

observed from our CCI analysis (Extended Data Fig. 10c and Supplementary Fig. 18e). Confirming their differentiation into trabecular-like hPSC-vCMs, these NRG1-treated hPSC-vCMs displayed increased expression of trabecular vCM-specific genes, including *IRX3, PLXNA2* and *PLXNA4*, and decreased expression of the compact vCM-specific marker *HEY2* (Supplementary Fig. 18e).

To investigate how intermediate-LV CC-derived SEMA3C, SEMA3D, SEMA6A and SEMA6B may affect the spatial distribution of *PLXNA2*⁺*PLXNA4*⁺ trabecular-like hPSC-vCMs in this hPSC ventricular wall model, these SEMA ligands were added in two different tiers (tier 1 and tier 2) of the intermediate-LV CC-like layer containing non-trabecular-like *TNNT2:*NLS-mKATE2 hPSC-vCMs. The spatial location of *PLXNA2*⁺*PLXNA4*⁺ trabecular-like (or control non-trabecular-like) *TNNT2:*eGFP hPSC-vCMs bioprinted in the inner-LV trabecular CC-like layer was then examined (Fig. 5b,d and Supplementary Fig. 19). When these SEMA ligands were present throughout the intermediate-LV CC-like layer, SEMA3C but not SEMA3D, SEMA6A or SEMA6B could direct the relocation of *PLXNA2*⁺*PLXNA4*⁺ trabecular-like *TNNT2:*eGFP hPSC-vCMs from the inner-LV trabecular CC-like layer to both tiers of the intermediate-LV CC-like layer (Fig. 5b,d). However, these SEMA ligands did not affect the spatial distribution of non-trabecular-like hPSC-vCMs bioprinted in either intermediate-LV (Fig. 5b) or control inner-LV CC-like layers (Supplementary Fig. 19), a finding that supports the idea that SEMA ligands may influence vCMs expressing *PLXNA2* and *PLXNA4*. Using an inducible *Tcf21-creERT2* mouse line, we investigated whether genetic deletion of *Sema3c* (*Sema3c*^fl/fl^) in cardiac fibroblasts could affect ventricular wall development in vivo, and discovered that *Tcf21-creERT2;Sema3c*^fl/fl^ cardiac ventricles exhibited hypertrabeculation and thinner compact myocardium beginning at embryonic day 14.5 (E14.5) (Fig. 5e,f and Extended Data Fig. 12). Together, these findings support the notion that SEMA3C may function as a key attractive guidance cue for driving the migration of *PLXNA2*⁺*PLXNA4*⁺ trabecular vCMs into the intermediate and outer layers of the ventricle during ventricular compaction.

Because *SEMA6A* and *SEMA6B* have been reported to repel *PLXNA2*-expressing and *PLXNA4*-expressing cells[38,39], we examined whether SEMA6A or SEMA6B could prevent SEMA3C from attracting *PLXNA2*⁺*PLXNA4*⁺ trabecular-like *TNNT2:*eGFP hPSC-vCMs from the inner-LV layer to specific tiers of the intermediate-LV CC-like layer (Fig. 5c). To this end, we exposed hPSC-vCMs in our hPSC ventricular wall model to different combinations of SEMA proteins between the two tiers of the intermediate-LV CC-like layer under the following conditions: (1) tier 1, no SEMA; tier 2, SEMA3C; (2) tier 1, SEMA6A; tier 2, SEMA3C; (3) tier 1, SEMA6B; tier 2, SEMA3C; (4) tier 1, SEMA6A and SEMA6B; tier 2, SEMA3C (Fig. 5c). SEMA condition 1 promoted the relocation of *PLXNA2*⁺*PLXNA4*⁺ trabecular-like *TNNT2:eGFP* hPSC-vCMs from the inner-LV trabecular CC-like layer to the intermediate-LV CC-like tier 1 (Fig. 5c,d). However, *PLXNA2*⁺*PLXNA4*⁺ trabecular-like *TNNT2:eGFP* hPSC-vCMs failed to migrate out of the inner-LV trabecular CC-like layer under SEMA conditions 2–4, supporting the idea that SEMA6A and SEMA6B may act as a repulsive guidance cue to prevent *PLXNA2*⁺*PLXNA*⁺ vCMs from migrating towards SEMA3C when they come in contact at the border of the inner-LV and intermediate-LV CC-like layers (Fig. 5c,d). Overall, these spatial and cell signalling findings suggest that *SEMA3C*⁺ compact vFibro cells may attract *PLXNA2*⁺*PLXNA4*⁺ trabecular and hybrid vCMs to the intermediate-LV and outer-LV CC layers, whereas *SEMA6A*⁺*SEMA6B*⁺ BECs may prevent these vCMs from migrating by repelling them after contact (Fig. 5g).

## Discussion

Our single-cell cardiac multimodal studies leveraged the combined power of scRNA-seq and MERFISH imaging to construct a comprehensive cell atlas of the developing human heart at spatial and molecular single-cell resolution. These multimodal analyses uncovered a broad

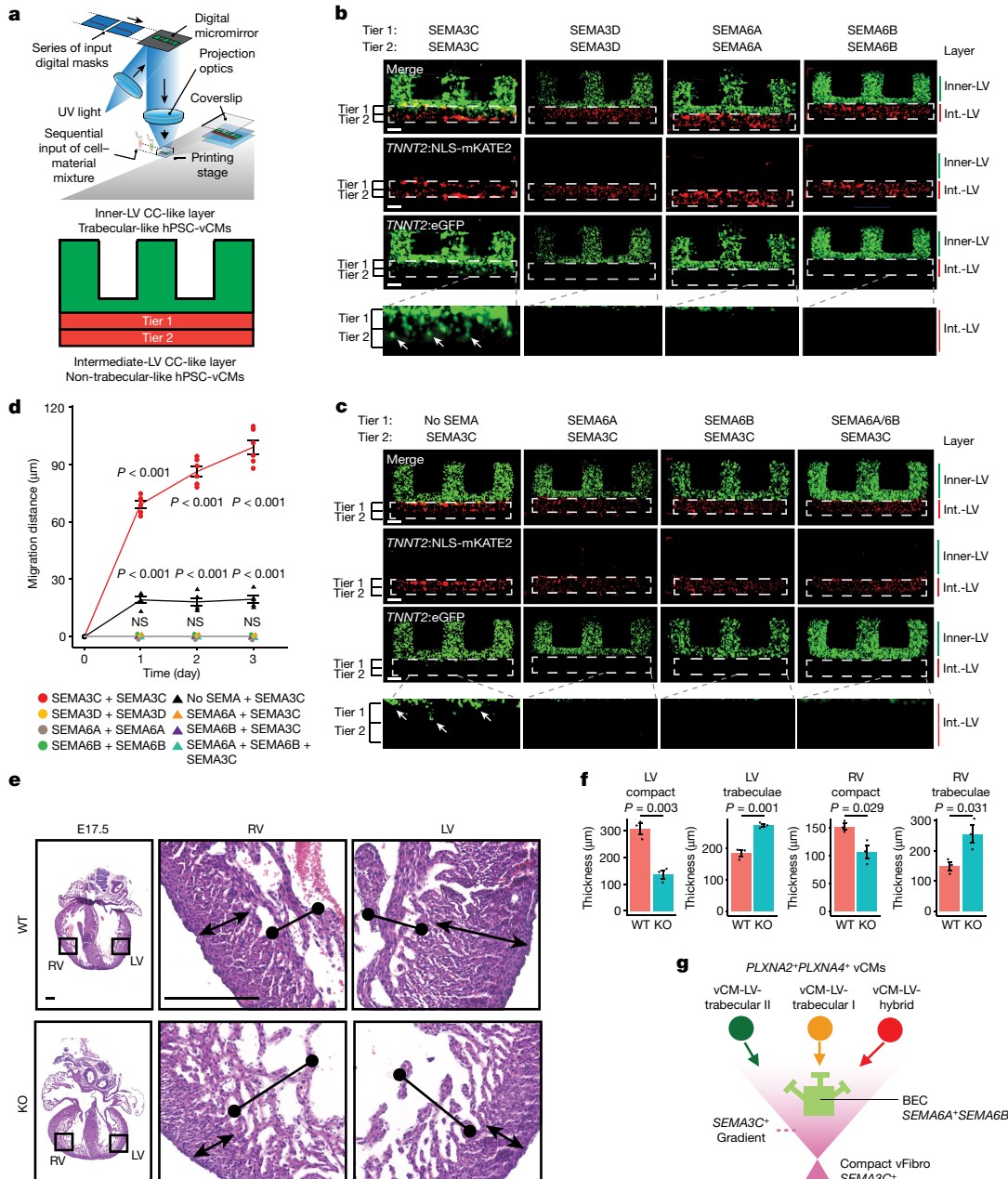

**Fig. 5 | PLXN–SEMA signalling mediates the migration of trabecular vCMs.**
**a**, NLS-mKATE2⁺ non-trabecular and GFP⁺ trabecular-like hPSC-vCMs were bioprinted into multilayered constructs modelling the ventricular wall as shown in the diagram. **b**, GFP⁺ trabecular-like hPSC-vCMs migrate to the intermediate ventricular-like layer (Int.-LV), which contains NLS-mKATE2⁺ non trabecular-like hPSC-vCMs mixed with SEMA3C but not when mixed with SEMA3D, SEMA6A or SEMA6B. White arrows point to GFP⁺ trabecular-like hPSC-vCMs migrating into the intermediate ventricular-like layer. **c**, SEMA6A or SEMA6B mixed in different combinations in the intermediate ventricular-like layer prevented SEMA3C-mediated GFP⁺ trabecular-like hPSC-vCM migration. White arrows point to GFP⁺ trabecular-like hPSC-vCMs migrating into the intermediate ventricular-like layer. **d**, GFP⁺ trabecular-like hPSC-vCM migration measurements under different intermediate-LV CC-like layer conditions. $N = 6$ and $N = 5$ independent experiments for SEMA3C+SEMA3C and no

SEMA+SEMA3C conditions, respectively. $N = 3$ independent experiments for all other conditions. Error bars are s.e.m. **e**, Representative sections of hearts from E17.5 wild type (WT) and *Tcf21-creERT2;Sema3c^{fl/fl}* knockout (KO) mouse embryos show that deletion of *Sema3c* in *Tcf21⁺* cells starting at E10.5 leads to a cardiac ventricular wall non-compaction phenotype. **f**, Graphs show the thickness of the compact and trabecular myocardium in WT and conditionally deleted *Sema3c* KO mouse hearts. $N = 3$ mice per condition. Error bars are s.e.m. **g**, Model shows how PLXN–SEMA interactions among distinct vCMs, fibroblasts and endothelial cells coordinate the organization of vCMs within the ventricular wall. White dashed lines in **b** and **c** outline the intermediate-LV CC-like layer. $P$ values in **d** and **f** determined by one-way analysis of variance. NS, not significant. Scale bars, 100 μm (**b**,**c**) or 250 μm (**e**). Schematic in **a** adapted from ref. 41, Elsevier.

range of cardiovascular lineages that participate in heart development and morphogenesis. The results also contributed new cardiac cell populations in important but underappreciated regions of the heart, such as the cardiac valves and conduction system, thus expanding the

current knowledge of cell types and states that constitute the human heart[10–18,23] (Supplementary Discussion). To gain insight into how these cell populations specialize according to their cellular and regional environment, we analysed our MERFISH-based high-resolution spatial

cardiac cell atlas, which enabled the interrogation of individual cells that form and interact within CCs that were related to distinct cardiac structures. Integrating this MERFISH imaging analysis with corresponding scRNA-seq data revealed the transcriptional profiles and the imputation of distinct genes for these spatially resolved individual cells.

Examining these particular genes with CCI algorithms helped identify distinct cell signalling ligand–receptor pairs that were expressed between spatially neighbouring cell populations to mediate their interactions. Although many of these identified signalling pathways were predicted across a wide range of cell types across the heart, we discovered that they differentially occurred between specific CCIs within distinct CCs. For instance, we observed distinct PLXN–SEMA signalling pathways among multiple combinations of interacting cell populations within specific layers of the ventricular wall that involved plexins and semaphorins previously reported in the ventricle[40]. However, we also identified an uncharacterized multicellular interaction among *PLXNA2*+*PLXNA4*+ ventricular cardiomyocytes, *SEMA3C*+*SEMA3D*+ fibroblasts and *SEMA6A*+*SEMA6B*+ endothelial cells, which may control the allocation of cardiomyocytes during the pivotal morphological process of ventricular wall compaction[30,34] (Fig. 5g and Supplementary Discussion). Overall, these findings highlight how our high-resolution molecular and spatial cardiac cell atlas offers insight into the detailed social interactions among distinct cell types that specialize and organize into cardiac structures crucial for maintaining heart function. Such information may be used in the future to not only understand the pathologic mechanisms that underlie congenital and adult structural heart diseases but also to develop new strategies for engineering complex multicellular cardiac tissues for heart repair.

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

# Methods

## Experimental procedures

**Tissue samples.** Heart samples were collected in strict observance of the legal and institutional ethical regulations. The heart samples were collected under a University of California San Diego (UCSD) Human Research Protections Program Committee Institutional Review Board (IRB)-approved protocol (IRB number 081510) by the UCSD Perinatal Biorepository's Developmental Biology Resource after informed consent was obtained from the donor families. All experiments were performed within the guidelines and regulations set forth by the IRB (IRB number 101021, registered with the Developmental Biology Resource). Ethical requirements for data privacy include that sequence-level data (for example, fastq files) be shared through controlled-access databases.

**Tissue processing.** Tissue samples were collected in buffer containing 10 mM HEPES pH 7.8, 130 mM NaCl, 5 mM KCl, 10 mM glucose, 10 mM BDM, 10 mM taurine, 1 mM EDTA and 0.5 mM $NaH_2PO_4$, and overall morphology was checked under a stereotaxic dissection microscope (Leica).

For single-cell dissociation, tissue samples from eight hearts were further cut into small pieces and enzymatically digested by incubating with collagenase type IV (Gibco) and Accutase (ThermoFisher) at 37 °C for 60 min. After removing the dissociation medium, cells were resuspended in PBS supplemented with 5% FBS and sorted using a Sony SH800 sorter. Samples were diluted to approximately 1,000 cells per µl before processing for scRNA-seq, as shown in Supplementary Fig. 1a.

Samples for MERFISH were washed with ice-cold PBS and then fixed in 4% paraformaldehyde at 4 °C overnight. On the second day, samples were washed in ice-cold PBS 3 times, 10 min each, and were incubated in 10% and 20% sucrose at 4 °C for 4 h each, and in 30% sucrose overnight, followed by immersion with OCT (Fisher, 23-730-571) and 30% sucrose (v/v) for 1 h. The samples were then embedded in OCT and stored at −80 °C until sectioning.

**hPSCs.** For the single-cell and bioprinting studies, a H9-hTnnTZ-pGZ-D2 hPSC line (*TNNT2:eGFP* hPSC cardiomyocyte reporter line) was purchased from WiCell and maintained as previously described[41]. For the bioprinting studies, an additional engineered *TNNT2:NLS-mKATE2* RUES2 hPSC cardiomyocyte transgenic reporter line that specifically expresses the mKATE2 fluorescent protein containing a nuclear localization signal (NLS-mKATE2) in differentiated cardiomyocytes was used (Supplementary Fig. 18a). Both lines were routinely authenticated with fluorescence microscopy, immunofluorescence and flow cytometry studies, and tested negative for mycoplasma contamination by PCR. To generate the *TNNT2:NLS-mKATE2-T2A-BsdR* RUES2 hPSC cardiomyocyte reporter line (*TNNT2:NLS-mKATE2*), we transfected a RUES2 hPSC line with a Piggybac (PB) construct expressing NLS-mKATE2-T2A-BsdR driven by the cardiomyocyte-specific *TNNT2* promoter. To clone the PB-*TNNT2:NLS-mKATE2-T2A-BsdR*, we used the PB plasmid pcsj532 (a gift from K. Willert, UCSD) and used Gibson assembly (SGI, GA1200) to clone in a synthesized *TNNT2* promoter[42] (Integrated DNA Technologies), PCR-amplified NLS-mKATE2-T2A-BsdR (with polyA) from pgRNA-CKB[43] (a gift from B. Conklin, Gladstone; Addgene, plasmid 73501) and PCR-amplified PGK:PuroR from RT-3GEPIR[44] (a gift from J. Zuber, IMP, Austria; Addgene, plasmid 111169). All three components were assembled in one Gibson assembly with pcsj532 digested using NheI (NEB R3131L). RUES2 hPSCs were transfected using Lipofectamine STEM (Invitrogen, STEM00015) with the PB-*TNNT2:NLS-mKATE2-T2A-BsdR* and a plasmid expressing a human-optimized PB transposase (pcsj533, a gift from K. Willert, UCSD) to integrate the PB. Two days after transfection, the cells were selected using 0.4 µg ml⁻¹ puromycin. The subsequent surviving cells behaved similarly to the parental and the *TNNT2:eGFP* hPSC lines in terms of proliferation and differentiation. Protocols were approved by the IRB (number 190561) at UCSD.

**hPSC cardiac cell differentiations and sample preparation.** hPSC lines were cultured in E8 medium and grown on Geltrex (Gibco)-coated plates. Differentiation of hPSCs into cardiomyocytes was performed using established protocols as previously described[41,45,46]. In brief, hPSCs were grown to 80% confluency, and on day 0 (D0), cells were cultured with RPMI/B27 supplement without insulin (B27 minus insulin; ThermoFisher) containing 10 µM CHIR (Fisher Scientific). After 24 h of CHIR application, the medium was replaced with fresh B27 without insulin and the cells were cultured for another 48 h. Next (D3), 5 µM IWP2 (Tocris) was supplemented to B27 without insulin and cultured for another 48 h. At D5, the B27 without insulin and with IWP2 was replaced with fresh B27 without insulin for another 48 h. From D7 onwards, cells were maintained in RPMI/B27 with insulin (B27, ThermoFisher). On D15, this B27 medium was then supplemented with either NRG1 (50 ng ml⁻¹)[47] or PBS, and further cultured until D30 and greater, refreshing the medium every 3 days.

scRNA-seq studies performed on hPSC-derived samples were prepared as described in the 'Tissue processing' section. In brief, D25 hPSC-derived cardiac cells were enzymatically digested by incubating with collagenase type IV (Gibco) and Accutase (ThermoFisher) at 37 °C for 60 min. After removing the dissociation medium, cells were resuspended in PBS supplemented with 5% FBS and sorted using a Sony SH800 sorter.

**Animal studies.** Animal studies were conducted in strict compliance with the Guide for the Care and Use of Laboratory Animals published by the National Institutes of Health and protocols approved by the Institutional Animal Care and Use Committee of UCSD (A3033-01). Mice were maintained on a 12 h–12 h light–dark cycle in a controlled temperature (20–22 °C) and humidity (30–70%) environment. The generation of *Tcf21-creERT2* and *Sema3c^{fl/fl}* mice has been previously described[48,49]. To validate the genotype of the mice, genomic DNA was extracted by adding a 2 mm tail clipping to a 75 µl solution containing 25 mM NaOH and 0.2 mM EDTA, and then heating the sample for 30 min at 98 °C. Next, 75 µl of 40 mM Tris-HCl (pH 5.5) was then added to neutralize the reaction, and a 1:50 dilution of genomic DNA template was used for genotyping PCR. Both male and female embryos were used in this study; the embryos were not genotyped to determine sex. To determine the developmental stage of embryonic development during which tamoxifen treatment was administered, noon on the day of the vaginal plug was assumed to be E0.5. Tamoxifen (Sigma, T5648-1G, 0.1 mg g⁻¹ body weight) was fed to pregnant mice by gavage at E10.5, and hearts were collected at E12.5, E14.5, E17.5 and postnatal day 1. The fixed hearts were embedded in paraffin, sectioned and stained with haematoxylin and eosin by the UCSD Histology Core. Images were taken on a Hamamatsu Nanozoomer Slide Scanning system and an Olympus VS200 slide scanner, and processed using NDP View 2 software (Hamamatsu) and QuPath (v.0.4.3)[50], respectively. Phenotypic analyses of ventricular wall thickness were performed as previously described[51]. In brief, the thicknesses of ventricular compact and trabecular zones were measured at the level of the papillary muscle, with measurements taken from at least three areas per section, and at least three sections per mouse.

**Single-cell transcriptome library preparation and sequencing.** Single-cell droplet libraries using the cell suspensions from the Sony SH800 sorter were prepared according to the manufacturer's instructions using the 10x Genomics Chromium controller, Chromium Single Cell 3′ Library and Gel Bead kit v2 (PN-120237) and Chromium i7 Multiplex kit (PN-120262). All libraries were sequenced on a HiSeq 4000 (Illumina) to a mean read depth of at least 65,000 total aligned reads per cell.

**MERFISH gene selection.** To spatially detect cell populations identified in the scRNA-seq dataset, we designed a panel of 238 genes specific for these subpopulations. These genes were then simultaneously imaged on cardiac samples using the combinatorial barcoded imaging technique MERFISH[7]. We initially identified gene markers differentially expressed for each of the 75 cell subpopulations by performing differential gene expression (DGE) analyses as well as applying a NS-Forest2 (ref. 52) classifier on scRNA-seq data obtained from the aforementioned human hearts in Supplementary Fig. 1. All markers were combined from the binary gene analysis utilizing NS-Forest2 (ref. 52) (159 genes) (Supplementary Table 7) and DGE analysis (7,557 genes) (Supplementary Table 3) of the cell subpopulations, and were then filtered for genes that were either not long enough to construct 48 target regions (each 30-nucleotides long) without overlap or for which expression levels were outside the range of 0.01–300 average unique molecular identifier (UMI) per cluster, as measured by scRNA-seq. The performance of identifying marker genes between NS-Forest2 and Spapros[53] pipelines was also compared. The initial result of Spapros produced 80 genes, which is half the number chosen by NS-Forest2. To compare a similar number of genes between NS-Forest2 and Spapros, these 80 genes were removed from the dataset and Spapros was run again, which selected another 90 genes. The combination of these two sets of genes were used for the Spapros gene list (Supplementary Table 9). To quantify the ability of the selected genes to re-identify cell subpopulations at the same granularity as annotated in the scRNA-seq data, the dimensionality reduction and neighbour graph were recalculated using only the genes selected by the algorithm (NS-Forest2 or Spapros) that was being evaluated. Each cell was then reassigned its cell subpopulation label based on the majority cell subpopulation of its five nearest neighbours in the new neighbour graph. The percentage of cells reassigned their original label was used as an accuracy metric. With this metric, we found that NS-Forest2 and Spapros chose genes with similar performance (Supplementary Table 10). Among the 238 MERFISH target genes, 63 were manually selected from the DGE and NS-Forest2 gene lists, including established markers for atrial, ventricular and non-chambered cardiomyocytes, as well as non-cardiomyocyte cell markers for fibroblasts, pericytes, VSMC, epicardial, endocardial, BEC, LEC and immune cells. Genes specific for platelet–red blood cells were not selected. To validate the final target gene list, we tested whether we could transcriptionally rederive the cell populations by cluster analyses using only the 238 target genes. To this end, we reduced the scRNA-seq dataset to only the 238 genes in the MERFISH gene panel and then performed dimensionality reduction, graph-based clustering and UMAP visualization. We observed a similar level of transcriptional separation and definition of cell classes between using the 238 target genes versus using the 3,000 variable genes chosen to annotate the cell classes in the scRNA-seq data (Fig. 1a and Extended Data Fig. 1b). In addition to the 238 MERFISH genes, we selected 11 genes that were imaged sequentially using smFISH (Supplementary Table 11), including genes that validated the combinatorial MERFISH imaging (Extended Data Fig. 2d).

**MERFISH probe library design and construction.** A total of 238 genes were identified as MERFISH target genes, which were subsequently used for probe generation and MERFISH assays as shown in Supplementary Table 11. To encode MERFISH RNA probes for spatial detection, a 22-bit modified Hamming distance 4 code was used[8]. Each of the 238 possible barcodes required at least 4 accumulated errors to be converted into another barcode. This property permitted the detection of errors up to any 2 bits, and the correction of errors to any single bit. In addition, this encoding scheme used a constant Hamming weight (that is, the number of 1 bits in each barcode) of 4 to avoid potential bias in the measurement of different barcodes due to a differential rate of 1 to 0 and 0 to 1 errors and because the optical density within each bit can interfere with resolving individual fluorescent spots, as previously described[6]. We used 238 out of the 7,315 possible barcodes to encode cellular RNAs and chose 10 barcodes that were left unassigned to serve as blank controls. The encoding probe set that we used contained 30–48 encoding probes per RNA, with each encoding probe containing 3 out of the 4 readout sequences assigned to each RNA. Encoding probes were designed using our own pipeline, namely, ProbeDesign. ProbeDesign was developed using a fully optimized algorithm in C++ for both DNA and RNA probes. ProbeDesign used the same principles of probe design utilized by various published algorithms (OligoArray[54], OligoMiner[55], OligoPaint[56] and ProbeDealer[57]), for which off-targets are based on genome-wide 17-nucleotide off-target counts. Probes were selected with similar GC content or melting temperature, and the repetitive regions were used for off-target counting but not for probe design. ProbeDesign was implemented in three steps. (1) Build a 17-nucleotide index based on the reference genome (DNA) or genome annotation files (RNA). This step is fully optimized with bit-vector and hash tables for high-performance computation; (2) Scan selected loci or genome sequences to calculate the off-targets based on the 17-nucleotide counts in the previous step. And (3), filter and rank probe candidates based on predefined selection criteria. For the RNA probe design, we used the transcript sequences derived from the human reference genome sequences (hg38) downloaded from ncbi_refseq (https://hgdownload.soe.ucsc.edu/goldenPath/hg38/bigZips/genes/). The generation of the encoding probe sets were prepared from oligonucleotide pools, as previously described[58,59]. In brief, we first used limited-cycle PCR to amplify the oligopools (Twist Biosciences). Then, we used these DNA sequences as the templates for in vitro transcription into RNA using T7 polymerase (NEB, E2040S). Subsequently, the RNA products were converted into single-stranded DNA with Maxima Reverse Transcriptase enzyme (Thermo Scientific, EP0751), and then the DNA was purified by alkaline hydrolysis (to remove the RNA templates) followed by DNA oligo purification kits (Zymo Research, D4060).

**MERFISH sample preparation.** Frozen hearts were sectioned at −20 °C on a cryostat (Leica CM3050S). A series of 12 μm coronal slices were cut at about 600 μm along the anterior–posterior axis of collected human hearts, which captured all of the major cardiac structures. MERFISH measurements of 238 genes with 10 non-targeting blank controls were performed as previously described using the encoded barcode sequences (Supplementary Table 11) and published readout probes[60]. In brief, 12-μm-thick tissue sections were mounted on 40 mm no. 1.5 coverslips that were silanized and poly-L-lysine coated[58] and subsequently pre-cleared by immersing into 50% (v/v) ethanol, 70% (v/v) ethanol and 100% ethanol, each for 5 min. The tissue was then air-dried for 5 min, followed by treatment with Protease III (ACDBio) at 40 °C for 30 min, and then washed with PBS for 5 min. Tissues were then preincubated with hybridization wash buffer (30% (v/v) formamide in 2× SSC) for 10 min at room temperature. After preincubation, the coverslip was moved to a fresh 60 mm Petri dish, and the residual hybridization wash buffer was removed with a Kimwipe laboratory tissue. In the new dish, the coverslip was immersed with 50 μl of encoding probe hybridization buffer (2× SSC, 30% (v/v) formamide, 10% (w/v) dextran sulfate, 1 mg ml$^{-1}$ yeast tRNA and a total concentration of 5 μM encoding probes and 1 μM of anchor probe: a 15-nucleotide sequence of alternating dT and thymidine-locked nucleic acid (dT+) with a 5′-acrydite modification (Integrated DNA Technologies)). The sample was then placed in a humidified 37 °C oven for 36–48 h and then washed with hybridization wash buffer for 20 min at 37 °C and 20 min at room temperature. Samples were post-fixed with 4% (v/v) paraformaldehyde in 2× SSC and then washed with 2× SSC with murine RNase inhibitor for 5 min. To anchor the RNAs in place, the encoding probe-hybridized samples were embedded in thin, 4% polyacrylamide (PA) gels as previously described[58]. In brief, the hybridized samples on coverslips were first washed with a de-gassed 4% PA solution, consisting of 4% (v/v) of 19:1 acrylamide/bis-acrylamide (Bio-Rad, 1610144), 60 mM Tris·HCl pH 8

(ThermoFisher, AM9856), 0.3 M NaCl (ThermoFisher, AM9759) and a 1:1,000 dilution of 0.1-μm-diameter blue fluorescent (350/440) beads (Life Technologies, F-8797). The beads served as fiducial markers for the alignment of images taken across multiple rounds of imaging. The coverslips were then washed again for 2 min with the same 4% PA gel solution supplemented with the polymerizing agents ammonium persulfate (Sigma, A3678) and TEMED (Sigma, T9281) at final concentrations of 0.03% (w/v) and 0.15% (v/v), respectively. The gel was then allowed to cast for 1.5 h at room temperature. The coverslip and the glass plate were then gently separated, and the PA film was incubated with a digestion buffer consisting of 50 mM Tris·HCl pH 8, 1 mM EDTA, 0.5% (v/v) Triton X-100 in nuclease-free water and 1% (v/v) proteinase K (New England Biolabs, P8107S). The sample was digested in this buffer for >36 h in a humidified, 37 °C incubator and then washed with 2× SSC 3 times. The samples were finally stained with an Alexa 488-conjugated anchor probe-readout oligonucleotide (Integrated DNA Technologies) and DAPI solution at 1 μg ml⁻¹. MERFISH measurements were conducted on a home-built system as previously described[60].

**Immunofluorescence studies.** For the immunofluorescence studies of the *TNNT2:NLS-mKATE2* hPSC line, D25 hPSC-derived cardiac cells were dissociated, replated and then cultured for another 4 days before being fixed. The fixed cells were then immunostained, as previously described[41], using an antibody for TNNT2 (A647 mouse anti-cardiac troponin T, BD 565744, 1:200). Nuclei were visualized using DAPI (1 μg ml⁻¹, Roche) staining. Immunofluorescent images were taken on a Nikon C2 confocal microscope.

**Real-time quantitative PCR.** RNA expression was measured by quantitative PCR (qPCR). RNA was extracted using TRIzol reagent (ThermoFisher) and a Direct-zol RNA MiniPrep kit (Zymo Research). cDNA was generated using 1,000 ng RNA mixed with iScript Reverse Transcription Supermix (Bio-Rad) and then diluted 1:10 in UltraPure DNase/RNase-free distilled water (ThermoFisher). qPCR was performed using Power SYBR Green master mix (ThermoFisher) according to the manufacturer's recommendations, and a two-step amplification CFX_2stepAmp protocol on a Bio-Rad CFX Connect Real-Time PCR Detection system. Data were analysed using the $2^{-\Delta\Delta Ct}$ method. All gene expression was normalized to the expression of TATA box binding protein (*TBP*). Primer sequences are listed in Supplementary Table 22.

**In vitro hPSC ventricular wall model.** To create an in vitro hPSC ventricular wall model for studying ventricular wall morphogenesis, we bioprinted cardiomyocytes in multilayered constructs as shown in Fig. 5. To this end, we differentiated *TNNT2:eGFP* and *TNNT2:NLS-mKATE2* hPSCs into D15 cardiomyocytes (hPSC-CMs) as described in the 'hPSC cardiac differentiations and sample preparation' section. D15 *TNNT2:eGFP* hPSC-CMs were further treated with NRG1 to create trabecular-like CMs as previously described[46]. As controls, D15 *TNNT2:eGFP* and *TNNT2:NLS-mKATE2* hPSC-CMs were treated with PBS. D30+ hPSC-CMs (>90% efficiency by flow cytometry) were then enzymatically dissociated with collagenase type IV (Gibco) and Accutase (ThermoFisher) and resuspended at 100 million cardiomyocytes per ml. The method to bioprint multilayered constructs involved printing D30+ hPSC-CMs that were treated with NRG1 or PBS into a rectangle with finger-like projections that was 500 × 700 × 600 μm (height × width × length) (inner-LV CC-like layer), followed by printing an adjacent rectangular structure (500 × 700 × 75 μm) (intermediate-LV CC-like layer) containing gelatin methacryloyl (GelMA)[61] mixed with 100 ng ml⁻¹ of different combinations of SEMA3C, SEMA3D, SEMA6A or SEMA6B (R&D Systems) proteins as described in Fig. 5b. The concentration of SEMA3C used for the conditions in Fig. 5c was 1,000 ng ml⁻¹ because SEMA3C was located further from the inner-LV CC-like layer. The cell-encapsulated layer was fabricated using 6.25% GelMA and 0.33% LAP with 15 s of light exposure time, and the cells were mixed with the monomer solution directly before bioprinting. The adjacent layer containing the signalling factors was printed using 4% GelMA and 0.4% LAP with 15 s of light exposure time. Using a methacrylated coverslip fixed to the controller stage, a 20 μl cell–material mixture was placed into the space between the coverslip and a polydimethylsiloxane (PDMS) film attached to a glass slide. This cell–material mixture was then exposed to UV light (365 nm, 88 mW cm⁻²) with a pattern generated by a digital micromirror device chip. After printing each layer, the construct was washed three times with warm PBS and aspirated dry. Finally, the multilayered construct was washed in both PBS and medium, and then stored in a cell culture incubator (37 °C, 5% $CO_2$). Medium was refreshed every other day.

### Data analysis
**Processing of raw sequencing reads.** Raw sequencing reads were processed using the Cell Ranger (v.3.0.1) pipeline from 10x Genomics. Reads were demultiplexed and aligned to the human hg38 genome, and UMI counts were quantified per gene per cell to generate a gene–barcode matrix.

**Cell filtering and clustering.** After generating the gene–barcode matrix file from Cell Ranger, the individual count matrices were merged together and processed using the Seurat (v.4.0.1) R package[62] (https://satijalab.org/seurat/). Further filtering and clustering analyses of the scRNA-seq cells were performed using the Seurat package, as described in the tutorials (https://satijalab.org/seurat/). Cells with at least 1,000 genes detected and a mitochondrial read percentage of less than 30% were used for downstream processing. Potential doublets were removed using DoubletFinder (v.2.0)[63] (https://github.com/chris-mcginnis-ucsf/DoubletFinder) using an anticipated doublet rate of 5%, which is the expected rate reported by 10x Genomics for the number of cells loaded onto the 10x Controller. For the aggregated dataset, gene expression was normalized for genes expressed per cell and total expression using the NormalizeData function. The top 3,000 variable genes were detected using the FindVariableFeatures function with default parameters. All of the genes were subsequently scaled using the ScaleData function, which utilizes a linear regression model to eliminate technical variability due to the number of genes detected, replicate differences and mitochondrial read percentage. Principal components were calculated using RunPCA, and the top 50 principal components (supported by ElbowPlot showing diminishing variance explained beyond the top 50 principal components) were used for creating the nearest neighbour graph utilizing the FindNeighbors function with k.param = 50. The generated nearest neighbour graph was then used for graph-based, semi-unsupervised clustering (FindClusters, default resolution of 0.8) and UMAP to project the cells into two dimensions. Marker genes were identified using a Wilcoxon rank-sum test (FindAllMarkers, default parameters) for one-versus-all comparisons for each of the cell clusters. Cell identities were assigned to the clusters by cross-referencing their marker genes with known cardiac cell type markers from both human and mouse studies, in addition to in situ hybridization data from the literature[10–15]. On occasion, a cell cluster would emerge that expressed marker genes representing multiple populations, as well as contained cells with low UMI and gene counts that escaped the first filtering step. These cells were removed from downstream analyses. The clustering approach was then repeated for each compartment of cells (cardiomyocyte, mesenchymal, endothelial, neuronal and blood) as described above, and the clustering accuracy was evaluated using SCCAF (v.0.0.10)[64] with the following parameters: linear regression with L1 regularization with a 50/50 train-test split and a fivefold cross validation. For the adult ventricle scRNA-seq comparison, we combined previously published datasets[33] with our developing heart scRNA-seq dataset and re-ran the NormalizeData function to compare gene expression between these datasets.

**Label transfer analysis.** Cell annotations from the scRNA-seq dataset were compared to a recently published adult heart dataset[18] utilizing scArches (v.0.5.9)[65]. For scArches, both the adult and developing transcriptomic datasets were integrated using scVI (v.1.0.3)[66] (n_hidden=128, n_latent=50, n_layers=3, dispersion = 'gene-batch'). A reference hierarchy tree was then constructed using the treeArches[67] workflow (https://docs.scarches.org/en/latest/treeArches_identifying_new_ct.html) with default parameters and "cell_state" labels on the adult heart published reference dataset[18]. Labels from the reference dataset were then transferred to the developing heart query dataset to predict the cell labels utilizing the scHPL.predict.predict_labels() function with default parameters. Rejected cells were calculated using the posterior probability (default option) with a 0.5 threshold.

**Gene regulatory network analysis.** To identify age-related changes in gene expression, we applied the pySCENIC (v.0.12.1)[68] gene regulatory network (GRN) inference tool to our scRNA-seq dataset. To this end, the scRNA-seq dataset was split by cell class, and pySCENIC analysis was performed to identify cell-class-specific regulons following the standard pipeline on GitHub (https://github.com/aertslab/SCENICprotocol/). In brief, we performed three steps to create a GRN for each cell class: (1) GRN inference using the GRNBoost2 algorithm, (2) transcription factor (TF) regulon predictions and (3) cellular enrichment area under the curve (AUC, measure of regulon activity) calculation for each cell. The resulting AUC matrix was then used to identify the regulons with the most significant change of activity over age by fitting a linear model to regulon activity and identifying regulons with the highest positive and negative rate of change. To find the functional pathways enriched in each set of regulons, we performed gene ontology enrichment analysis using the EnrichR Bioconductor package (v.3.1)[69]. On the same regulons, we constructed a regulatory network with the top 100 non-redundant edges of the network by importance score, and visualized the edges, transcription factors and target genes using Cytoscape (v.3.8.0)[70]. For the overall GRN of vCMs, we constructed a regulatory network with the top 50 transcription factors by centrality and then took the top 500 non-redundant edges of the network by importance score and visualized the edges and transcription factors using Cytoscape.

**CCI analysis.** We applied CellChat (v.1.6.1)[71] to our scRNA-seq dataset to identify region-specific CCIs. Atrial cells and ventricular cells were divided based on their region of dissection (LA/RA for atrial and LV/RV/IVS for ventricular) and were analysed separately. Next, we followed the suggested workflow of CellChat, using its database of human ligand–receptor interactions (with the addition of the NRG1–ERBB2 signalling pathway owing to its known biological role during cardiac development[31,36,37]), identifying overexpressed genes, computing interaction probabilities and discovering significant interactions based on default parameters. This pipeline was performed using all cell classes present in each region (except for platelet–red blood cells) to calculate potential CCIs.

**Developmental trajectory analysis.** To identify a developmental trajectory of vCMs within our scRNA-seq dataset, we used the Waddington-OT (v.1.0.8)[72] package. The vCM cell class was isolated, which represents subpopulations C1–C11 of the cardiomyocyte compartment, and the corresponding cells were used for Waddington-OT trajectory inference as outlined in GitHub (https://broadinstitute.github.io/wot/tutorial/) without the optional step of estimating initial growth rates. Transport matrices were calculated for each adjacent pair of time points (9 p.c.w.–11 p.c.w., 11 p.c.w.–13 p.c.w., and 13 p.c.w.–15 p.c.w.) and then the trajectory was computed by calculating the descendent distributions at the 9 p.c.w stage. Normalized pseudotime values used for subsequent analyses were calculated by taking the quantile for each cell ranked by raw pseudotime value. Following the construction of the developmental trajectory, the resulting pseudotime for vCMs was utilized to order the GRN and CCIs of the vCMs by determining the expression-weighted pseudotime of each respective transcription factor and receptor or ligand expressed by vCMs as previously described[73].

**MERFISH processing.** For processing MERFISH data, individual RNA molecules were decoded using MERlin (v.0.6.1) as previously described[8]. Images were aligned across hybridization rounds by maximizing phase cross-correlation on the fiducial bead channel to adjust for drift in the position of the stage from round to round. Background was reduced by applying a high-pass filter, and decoding was then performed per pixel. For each pixel, a vector was constructed of the 22 brightness values from each of the 22 rounds of imaging. These vectors were then L2-normalized, and their Euclidean distances to each of the L2-normalized barcodes from MERFISH codebook were calculated. Pixels were assigned to the gene whose barcode they were closest to, unless the closest distance was greater than 0.512, in which case the pixel was not assigned a gene. Adjacent pixels assigned to the same gene were combined into a single RNA molecule. Molecules were filtered to remove potential false positives by comparing the mean brightness, pixel size and distance to the closest barcode of molecules assigned to blank barcodes versus molecules assigned to genes to achieve an estimated misidentification rate of 5%. The exact position of each molecule was calculated as the median position of all pixels consisting of the molecule.

Cellpose (v.1.0.2)[74] was used to perform image segmentation to determine the boundaries of cells and nuclei. Distinct RNA molecules were identified and assigned to individual cells as segmented by total polyadenylated RNA staining and DAPI staining, which allowed for the detection of cellular boundaries. The nuclei boundaries were determined by running Cellpose with the 'nuclei' model using default parameters on the DAPI stain channel of the pre-hybridization images. Cytoplasm boundaries were segmented with the 'cyto' model and default parameters using the polyT stain channel. RNA molecules identified by MERlin were assigned to cells and nuclei by applying these segmentation masks to the positions of the molecules. Any segmented cells that did not have any barcodes assigned were removed before constructing the cell-by-gene matrix.

**smFISH computational analysis.** Images were flatfield-corrected for the two gene channels (750 nm and 635 nm) and the fiducial marker (405 nm) channel. To reduce background noise for each hybridization round, the images of the preceding hybridization round were reduced in intensity and subtracted to obtain new background-subtracted images. The images were then locally normalized by subtracting a 15 × 15 blur from each pixel before undergoing maximum intensity projection into two dimensions. For transcript detection, the OpenCV function adaptiveThreshold was used with a block size of 41 pixels, and a subtracted constant ranging from −80 to −70 among our replicate smFISH experiments. This constant was empirically determined by choosing a value that ensured the resulting mask only captured visible fluorescent spots across diverse imaging planes for each gene. Using the regionprops function from Scikit-Image, we filtered out spots with an eccentricity value of 0 and cells with low pixel area (<4 pixels) to combat artefactual fluorescence detection. A global threshold was identified for the images of each gene by evaluating the values of features determined as nonspecific background (for example, irregular shape, low intensity). The coordinates of local brightness maxima that remained unattenuated after applying this global threshold were stored. Coordinates lying within the adaptiveThreshold mask boundaries were identified and counted as a single identified gene transcript. The images for each of the smFISH imaging rounds were aligned to the respective initial MERFISH hybridization round images to correct for microscopic drift using the fiducial marker channel. This was done by fitting spots to the fiducial bead markers of both sets of images, then

minimizing the median distances between them. DAPI segmentation masks obtained from the MERFISH imaging were translated using this drift correction so that all identified gene transcript locations could accurately be assigned to the drift-corrected nuclei, which enabled reconstruction of a spatial mosaic of the cellular gene expression for each of our sequentially imaged gene targets. For the replicate smFISH experiments to validate MERFISH gene markers, each gene was imaged twice on the same colour channel but in different non-consecutive rounds, allowing for a more robust analysis by using the combination of the two images to reduce the effect of noise in the result. Genes were imaged on three colour channels: Alexa 750, Cy5 and Cy3. The genes were separated and analysed into batches of six, with the imaging being done in a pattern described as follows. In the first imaging round, genes A, B and C were imaged on the Alexa 750, Cy5 and Cy3 channels, respectively. On the second round, genes D, E and F were imaged on the same three channels. The third and fourth imaging rounds were then repeats of the first and second rounds. By imaging each gene twice, the data could be analysed as a pseudo-MERFISH experiment, whereby a codebook was designed with each gene having a barcode containing two 'on' bits in the two imaging rounds they were imaged. Using this codebook, the data was processed using the same method as the MERFISH data as previously described[8].

**Cell clustering analysis of MERFISH.** With the cell-by-gene matrix, we followed a standard procedure as suggested in the Scanpy (v.1.8)[75] tutorial using Python (v.3.9) for processing MERFISH data. Count normalization, principal component analysis (PCA), neighbourhood graph construction and UMAP were performed using the default parameters in Scanpy. We performed Leiden clustering utilizing a resolution of 2 because the additional clusters gained at higher resolutions either did not have differentially expressed genes or were related to technical imaging artefacts. The top 20 differential genes identified by the rank_gene_groups() function were used to annotate each cluster. We further subclustered the vCM clusters using Leiden clustering at a default resolution of 1 to further annotate vCMs as compact and trabecular vCMs for both the left and right ventricles. To investigate the cellular populations in the ventricle specifically (both 13 p.c.w. and 15 p.c.w.), we manually defined the ventricular region, subsetted the MERFISH dataset to those cells populating the ventricle and performed Leiden clustering using a similar strategy to that used in the overall clustering (resolution of 5). Again, the top 20 genes identified by the rank_gene_groups() function were used to annotate each cluster.

**Integration of the scRNAseq and MERFISH datasets.** To integrate the scRNA-seq and MERFISH datasets, we first isolated both datasets to only the 238 MERFISH target genes interrogated by both modalities. We then utilized Scanpy's implementation of Harmony to project both the scRNA-seq and MERFISH datasets into a shared PCA space[76]. The dimensionality of the joint embedding was visualized using UMAP (min_dist=0.3, 30 nearest neighbours in Pearson correlation distance). The parameters matched those used in a previous publication of Harmony[76]. To impute a complete expression profile and cell label for each MERFISH profile, we assigned the expression profile and cell label of the closest scRNA-seq cell to the MERFISH cell in the Harmony PCA space using the Euclidean distance metric (default number of PCs).

To evaluate the performance of the gene imputation method, we developed a predictability score for each gene which is the Pearson correlation between the imputed expression and measured image expression for all genes (Supplementary Fig. 14a). Because the shared embedding space is constructed using the 238 MERFISH target genes, it is expected that these genes would have higher predictability scores than genes not used in the construction. To avoid this bias, tenfold cross-validation was used to calculate independently the MERFISH and scRNA-seq gene predictability scores. To this end, a new shared

embedding utilizing only 90% of the 238 MERFISH target genes was used to calculate the MERFISH and scRNA-seq gene predictability scores for the remaining 10% of genes that were not included for constructing the embedding. This process was repeated 10 times with a different 10% of genes being imputed by a different shared embedding each time to cover the full set of 238 genes. To calculate whether a predictability score represented a prediction that is a significant improvement over random prediction, we calculated predictability scores using a randomly connected neighbour graph. In other words, rather than predicting the expression from the cell with the most similar gene expression, the prediction was made from a randomly selected cell in the dataset, and then the predictability score was calculated between the measured expression values and these randomly predicted values. This process was repeated 100 times to estimate a normal distribution of predictability scores that result from random prediction. *P* values were then determined for the true predictability scores based on rejecting the null hypothesis that the true scores originated from the null normal distributions. Finally, these *P* values were corrected for multiple hypothesis testing using the Bonferroni method. We observed that the maximum scRNA-seq predictability score for a gene that failed this significance test (adjusted *P* value > 0.01) was 0.11.

**Identifying CCs.** We sought to define CCs that represented distinct shared cellular environments defined by the neighbouring cells in close proximity. To this end, we clustered each MERFISH cell based on the cell composition of neighbouring cells within a 150 μm zone, which represented a typical diffusion distance for extracellular signalling molecules[77]. This zone sampled approximately 300 neighbours. The zone of each cell was therefore represented by a vector containing the relative proportions of each of the 27 identified cells in both the overall and ventricular subset of the MERFISH dataset. We then clustered the zones using Python's scikit-learn (v.0.22) implementation of Kmeans with $k = 13$ for the overall MERFISH dataset or $k = 9$ for the ventricular subset of the MERFISH dataset, chosen by silhouette score. Thus, each MERFISH cell was assigned to 1 of the 13 or 9 CCs in the overall or ventricular subset of the MERFISH dataset, respectively.

To infer community-specific CCIs, cell annotations derived for each MERFISH cell were transferred to the nearest scRNA-seq profile in the Harmony joint embedding space and used for the pipeline of CellChat as described in the 'CCI analysis' section. The overall and ventricular communities were each analysed separately by analysing the scRNA-seq profiles assigned to those communities. For each overall or ventricular cellular community, we only considered cells that represented at least 5% or 3.5% of the community in the MERFISH dataset for CellChat CCI analysis.

**Connectivity, ventricular wall depth and pseudotime analyses of vCMs.** To visualize the similarity in the gene expression profiles of the vCMs, we isolated the vCM-LV-compact I, vCM-LV-compact II, vCM-LV-hybrid, vCM-LV-trabecular I, vCM-LV-trabecular II and vCM-LV/RV-Purkinje populations, and reperformed PCA, nearest neighbour graph construction and UMAP utilizing Scanpy with default parameters. We then used Scanpy's implementation of partition based graph abstraction to construct a graph in which the nodes represent different vCMs, and the edges represent the degree of connectivity between the vCMs in the neighbourhood graph. This captured a measure of similarity between the vCMs.

To determine the distance of each MERFISH cell within the ventricular wall, the ventricular wall depth of each MERFISH cell was defined as the distance to the nearest epicardial cell or EPDC, which both lie on the outer layer of the heart. To compare ventricular wall depth between different ventricles, the wall depth values were normalized by dividing each value by the maximum depth of the corresponding ventricle. To identify depth correlated genes, we computed the Pearson correlation

coefficient between expression and ventricular wall depth for each gene. We considered genes with a correlation greater than 0.2 to be depth correlated. Next, the diffusion pseudotime distance on the vCM nearest neighbour graph from the medoid of the vCM-LV-compact I cluster were calculated using Scanpy's scanpy.tl.dpt() function with default parameters. We note that this metric is often reported as pseudotime to represent developmental changes, but in this case, we use it simply as a metric for expression similarity to vCM-LV-compact I cells. The scaled expression of the top genes correlating with pseudotime were plotted as a smoothed spline (Extended Data Fig. 6d) as previously described[78].

**Migration distance measurement.** The migration distance of the bioprinted hPSC-CMs was measured by using the D0 position as the starting point and calculating the distance migrated by the hPSC-CMs daily. In brief, brightfield and fluorescent (green and red) confocal images of the GFP+ and NLS-mKATE2+ hPSC-CMs were taken on a Leica SP5, with the brightfield images used to visualize the construct. Because of minor variations in size and cell number between printed constructs, we normalized the migration distances by first dividing the width of the construct into 15 even blocks for each image. Within each block, the distances from the D0 position to the furthest position (for each day) of the hPSC-CMs were calculated. We then averaged the distances measured for the 15 blocks to calculate the migration distance of each condition and line.

**Statistics and reproducibility.** Replicates and statistical tests are described in the figure legends. Sample sizes were not predetermined utilizing statistical methods. Tissue samples were not randomized, nor were the investigators blinded during collection as no subjective measurements were taken. Data for scRNA-seq and MERFISH were collected from all available samples and no randomization was necessary. For the studies utilizing human pluripotent stem cell lines, treatment with NRG1 was randomly assigned. For the animal studies, animals were randomly chosen from each genotype and stage. Blinding during analysis was not necessary as all of the results were analysed with the use of unbiased analysis and software tools that are not affected by the sample. All experiments were independently repeated at least three times with similar results, including experiments in Fig. 5e, Extended Data Figs. 2a and 12a and Supplementary Fig. 18a. To identify differentially expressed genes between clusters, a Wilcoxon rank-sum test was performed and the resulting $P$ value was corrected using the Bonferroni procedure. For the scRNA-seq predictability scores, the $P$ values were generated by using bootstrapping to generate a distribution of scores for each gene and then calculating (1− cumulative distribution function) to obtain the $P$ value. For the migration distance, ventricular wall thickness and qPCR results, we used a one-way analysis of variance using R (v.4.2.0; https://www.r-project.org/).

The sample sizes for the violin plot in Fig. 2e are listed as follows: from top to bottom, $n$ = 9,106, 7,661, 19,901, 3,791, 4,003, 60,810, 28,263, 16,369, 6,956, 21,087, 17,940, 5,135 and 27,613 cells. Fig. 4c: from left to right, $n$ = 541, 849, 1,552, 719, 2,290, 1,112, 754, 499, 335, 55, 13, 701, 338, 177, 163 and 49 cells. Extended Data Fig. 5d: from left to right, $n$ = 9,106, 7,661, 19,901, 3,791, 4,003, 60,810, 28,263, 16,369, 6,956, 21,087, 17,940, 5,135 and 27,613 cells. Extended Data Fig. 5e: from left to right, $n$ = 27,613, 5,135, 17,940, 21,087, 7,661, 6,956, 3,791, 28,263, 9,106, 4,003, 60,810, 16,369 and 19,901 cells. Extended Data Fig. 8f: top panel from left to right (13 p.c.w./15 p.c.w.), $n$ = 573/706, 976/160, 1,532/895, 354/303, 784/553, 720/NA, 187/NA, 1,440/866, 1,905/840 and 508/417 cells; bottom panel from left to right (13 p.c.w./15 p.c.w.), $n$ = 711/274, 313/305, 548/711, 387/21, 1,444/938, 800/451, 557/409, 79/80 and 66/124 cells. Extended Data Fig. 9d: from left to right, $n$ = 9,723, 18,908, 21,203, 8,042, 47,906, 16,225, 5,814, 6,307 and 18,592 cells. Extended Data Fig. 9e: from left to right, $n$ = 18,592, 8,042, 5,814, 6,307, 47,906, 21,203, 18,908, 16,225 and 9,723. Extended Data Fig. 11e: from left to right, $n$ = 81,880, 75,531,

34,953, 145,935, 19,949 and 18,485 RNA molecules. Violin plots consisting of cell numbers of ten or fewer are not shown and are labelled as 'NA' in those cases.

## Reporting summary

Further information on research design is available in the Nature Portfolio Reporting Summary linked to this article.

## Data availability

Raw sequencing data for the in vivo studies are available from dbGAP under accession number phs002031. Raw sequencing data, and related files, for the in vitro studies are available from CIRM CESCG (https://cirm.ucsc.edu) under accession number chiCardiomyocyte1. Processed scRNA-seq data are accessible from the UCSC Cell Browser[79] (https://cells.ucsc.edu/?ds=hoc). MERFISH imaging data are available from Dryad (https://doi.org/10.5061/dryad.w0vt4b8vp)[80]. The human reference genome sequences (hg38) can be downloaded from ncbi_refseq (https://hgdownload.soe.ucsc.edu/goldenPath/hg38/bigZips/genes/). Source data are provided with this paper.

## Code availability

The pipeline for generating the encoding probes used in the MERFISH studies is available from GitHub (https://github.com/bil022/Probe-Design). The pipeline for processing the MERFISH dataset, including cell segmentation and assigning barcodes to cells, are available from GitHub (https://github.com/epigen-UCSD/merfish_tools). Custom code used for analysing the scRNA-seq and MERFISH datasets in this study are available from GitHub (https://github.com/ChiLab-UCSD/Heart_MERFISH_analysis).

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

**Acknowledgements** This work was supported in part by the California Institute for Regenerative Medicine (GC1R-06673-B) for funding, and sequencing and data repository support from C. Barragan, J. Ecker, J. Stuart, C. Fischer, W. Sullivan, P. Nejad, B. Aevermann and R. Scheuermann. We thank A. R. Holman and Y. Zhang for technical assistance; members of the Chi, Chen, Ren and Evans laboratories for comments; and staff at the UCSD core facilities Perinatal Biorepository, Center for Epigenomics, Institute for Genomic Medicine, and Histology/Immunohistochemistry. This work was supported in part by grants from the NIH to B.R., S.M.E., Q. Zhu and N.C.C., the Chan Zuckerberg Initiative (2023-321230) to B.R., Q. Zhu and N.C.C., and the NSF (2135720) to S.C. Illustrations in Fig. 1a and Supplementary Fig. 1a were created using BioRender (https://www.biorender.com).

**Author contributions** E.N.F. and N.C.C. conceived the project and the overall design of the experimental strategy. Q. Zhu designed and performed MERFISH experiments. R.K.H. and C.K. designed and performed bioinformatics analyses. E.N.F., Q. Zhang, T.-Y.L., B.Z., Z.W. and E.D. conducted experiments. E.N.F., Q.M., S.T., D.C., A.M., A.P.B., J.E. and B.L. helped with bioinformatics analyses. S.C. helped provide key reagents and guidance on experimental design of the bioprinting experiments. S.M.E. and B.R. provided critical intellectual input and data interpretation. E.N.F. and N.C.C. prepared the manuscript with input from all authors.

**Competing interests** B.R. is a shareholder and consultant of Arima Genomics and co-founder of Epigenome Technologies. All other authors have no conflicts of interest.

**Additional information**
**Correspondence and requests for materials** should be addressed to Quan Zhu or Neil C. Chi.

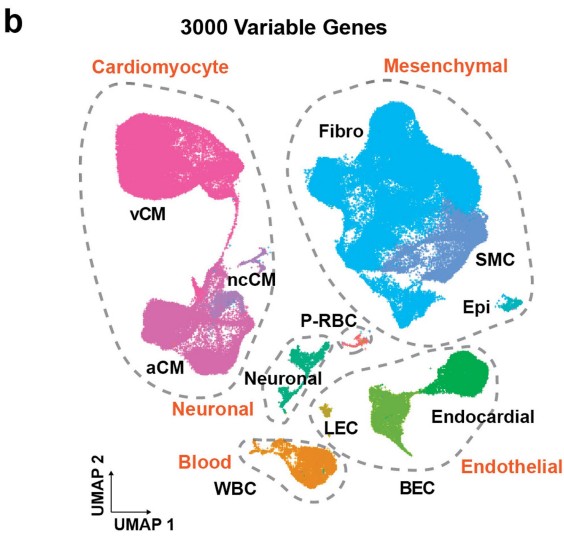

**a**

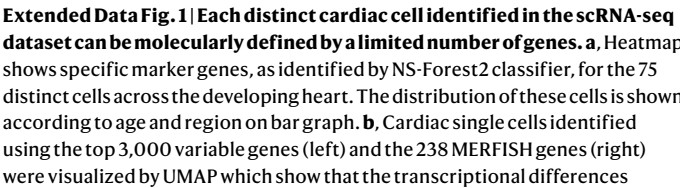

**b**

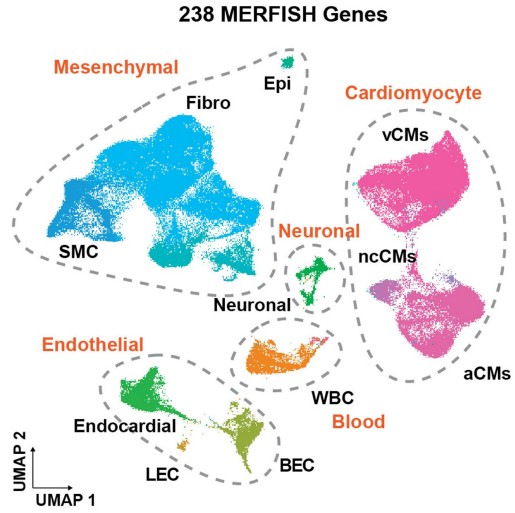

**Extended Data Fig. 1 | Each distinct cardiac cell identified in the scRNA-seq dataset can be molecularly defined by a limited number of genes. a,** Heatmap shows specific marker genes, as identified by NS-Forest2 classifier, for the 75 distinct cells across the developing heart. The distribution of these cells is shown according to age and region on bar graph. **b,** Cardiac single cells identified using the top 3,000 variable genes (left) and the 238 MERFISH genes (right) were visualized by UMAP which show that the transcriptional differences between the cell compartments (grey dashed lines) and classes (colored in **a**) are preserved with a limited set of genes. aCM, atrial cardiomyocyte; BEC, blood endothelial cell; Epi, epicardial; Fibro, fibroblast; IVS, interventricular septum; LA, left atrium; LEC, lymphatic endothelial cell; LV, left ventricle; ncCM, non-chambered cardiomyocyte; p.c.w., post conception weeks; P-RBC, platelet-red blood cell; RA, right atrium; RV, right ventricle; SMC, smooth muscle cell; vCM, ventricular cardiomyocyte; WBC, white blood cell.

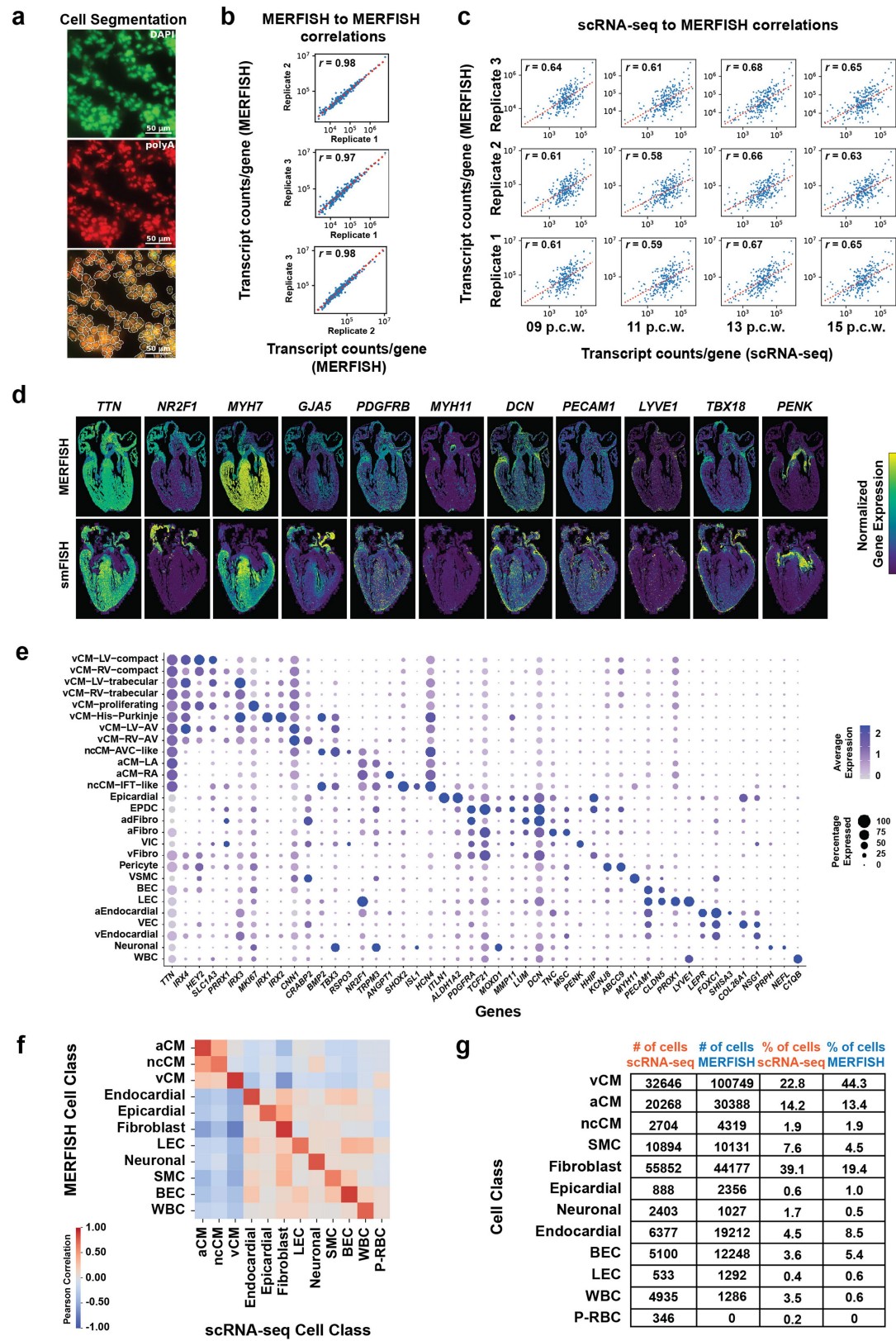

**Extended Data Fig. 2** | See next page for caption.

**Extended Data Fig. 2 | Quality control analyses of MERFISH data reveal its reproducibility and correspondence with scRNA-seq. a**, MERFISH cell boundaries were defined using CellPose[74] with DAPI and polyA staining as input images. **b**, Pearson correlation of the counts of the 238 MERFISH target genes reveals strong correlation among the three replicate MERFISH experiments (Pearson correlation coefficient ($r$) > 0.95). **c**, Pearson correlation of the transcript counts of the 238 target genes shows that the 13 p.c.w. stage displays the highest average correlation (0.67 Pearson correlation) between the MERFISH and scRNA-seq datasets. **d**, MERFISH imaging was validated spatially by comparing normalized gene expression profiles of marker genes measured by single molecule FISH (smFISH) imaging with those detected by MERFISH imaging. **e**, Marker gene analysis identified each distinct MERFISH cell. **f**, Heatmap reveals that cell classes identified in the scRNA-seq dataset are detected in the MERFISH dataset, with the exception of P-RBCs. **g**, Table shows cellular composition similarities between the scRNA-seq and MERFISH datasets. aCM, atrial cardiomyocyte; aFibro, atrial fibroblast; adFibro, adventitial fibroblast; aEndocardial, atrial endocardial; AVC, atrioventricular canal; BEC, blood endothelial cell; CM, cardiomyocyte; EPDC, epicardial-derived cell; IFT, inflow tract; LA, left atrium; LEC, lymphatic endothelial cell; LV, left ventricle; ncCM, non-chambered cardiomyocyte; p.c.w., post conception weeks; P-RBC, platelet-red blood cell; RA, right atrium; RV, right ventricle; SMC, smooth muscle cell; vCM, ventricular cardiomyocyte; vCM-LV/RV-AV, muscular valve leaflet vCM; vEndocardial, ventricular endocardial; VEC, valve endocardial cell; vFibro, ventricular fibroblast; VIC, valve interstitial cell; VSMC, vascular smooth muscle cell; WBC, white blood cell. Scale bar, 50 μm.

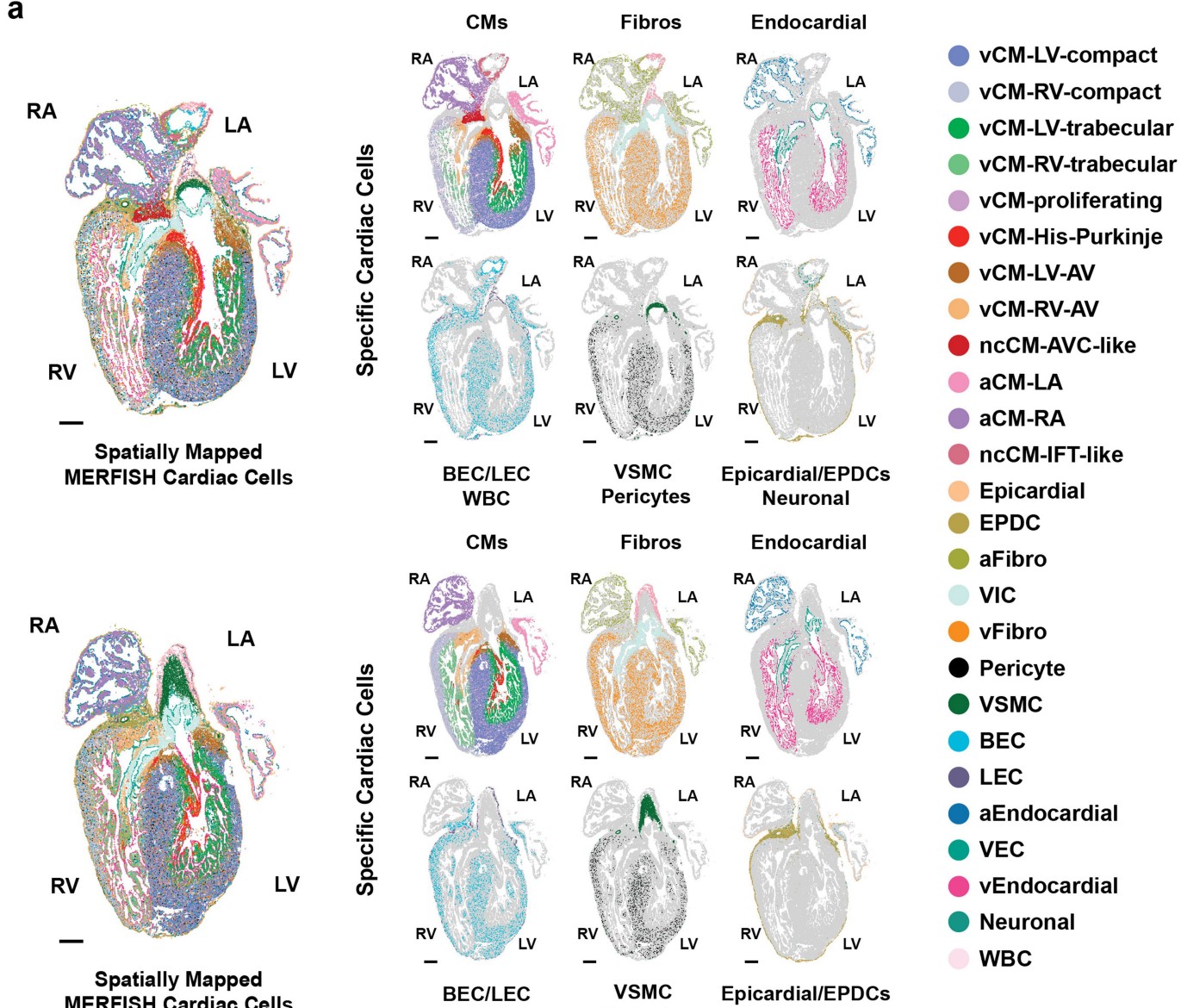

**a**

Specific Cardiac Cells

CMs   Fibros   Endocardial

BEC/LEC WBC   VSMC Pericytes   Epicardial/EPDCs Neuronal

Spatially Mapped MERFISH Cardiac Cells

- vCM-LV-compact
- vCM-RV-compact
- vCM-LV-trabecular
- vCM-RV-trabecular
- vCM-proliferating
- vCM-His-Purkinje
- vCM-LV-AV
- vCM-RV-AV
- ncCM-AVC-like
- aCM-LA
- aCM-RA
- ncCM-IFT-like
- Epicardial
- EPDC
- aFibro
- VIC
- vFibro
- Pericyte
- VSMC
- BEC
- LEC
- aEndocardial
- VEC
- vEndocardial
- Neuronal
- WBC

**Extended Data Fig. 3 | MERFISH cells were reproducibly mapped to distinct spatial regions of the developing heart. a**, Spatial mapping of identified MERFISH cells on two additional 13 p.c.w. frontal heart section replicates reveals the reproducibility of each distinct MERFISH cell and their spatial distributions. aCM, atrial cardiomyocyte; aFibro, atrial fibroblast; adFibro, adventitial fibroblast; aEndocardial, atrial endocardial; AVC, atrioventricular canal; BEC, blood endothelial cell; CM, cardiomyocyte; EPDC, epicardial-derived cell; IFT, inflow tract; LA, left atrium; LEC, lymphatic endothelial cell; LV, left ventricle; ncCM, non-chambered cardiomyocyte; p.c.w., post conception weeks; RA, right atrium; RV, right ventricle; vCM, ventricular cardiomyocyte; vCM-LV/RV-AV, muscular valve leaflet vCM; vEndocardial, ventricular endocardial; VEC, valve endocardial cell; vFibro, ventricular fibroblast; VIC, valve interstitial cell; VSMC, vascular smooth muscle cell; WBC, white blood cell. Scale bar, 250 μm.

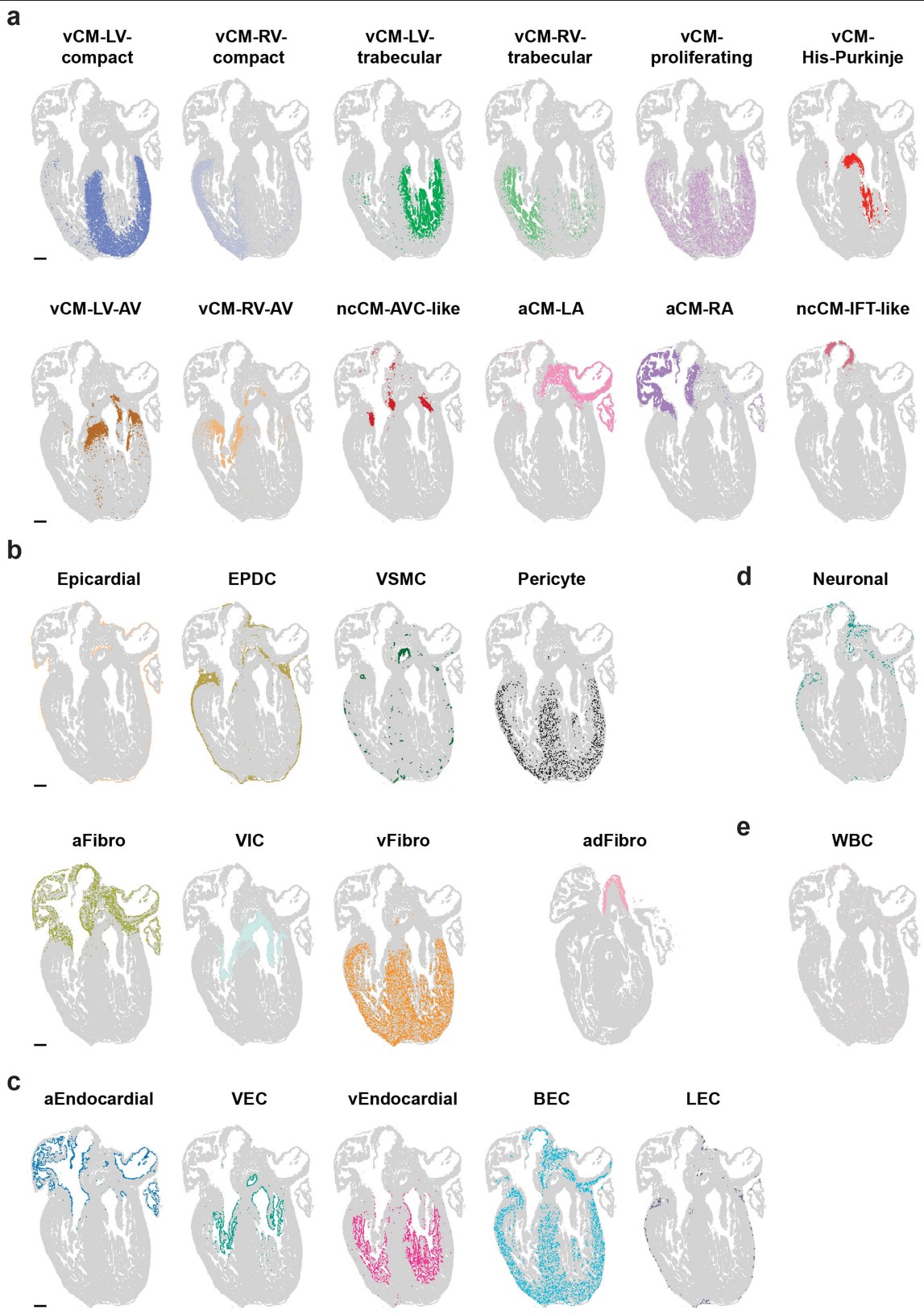

**Extended Data Fig. 4 | Identified MERFISH cardiac cells map to distinct regions and anatomical structures of the human heart.** The spatial mapping of each identified MERFISH cell is displayed accordingly: **a**, cardiomyocyte related cells, **b**, epicardial, EPDC, and vascular support related cells, **c**, endothelial related cells, **d**, neuronal cells, and **e**, blood related cells. aCM, atrial cardiomyocyte; aFibro, atrial fibroblast; adFibro, adventitial fibroblast; aEndocardial, atrial endocardial; AVC, atrioventricular canal; BEC, blood endothelial cell; EPDC, epicardial-derived cell; IFT, inflow tract; LA, left atrium; LEC, lymphatic endothelial cell; LV, left ventricle; ncCM, non-chambered cardiomyocyte; RA, right atrium; RV, right ventricle; vCM, ventricular cardiomyocyte; vCM-LV/RV-AV, muscular valve leaflet vCM; vEndocardial, ventricular endocardial; VEC, valve endocardial cell; vFibro, ventricular fibroblast; VIC, valve interstitial cell; VSMC, vascular smooth muscle cell; WBC, white blood cell. Scale bar, 250 μm.

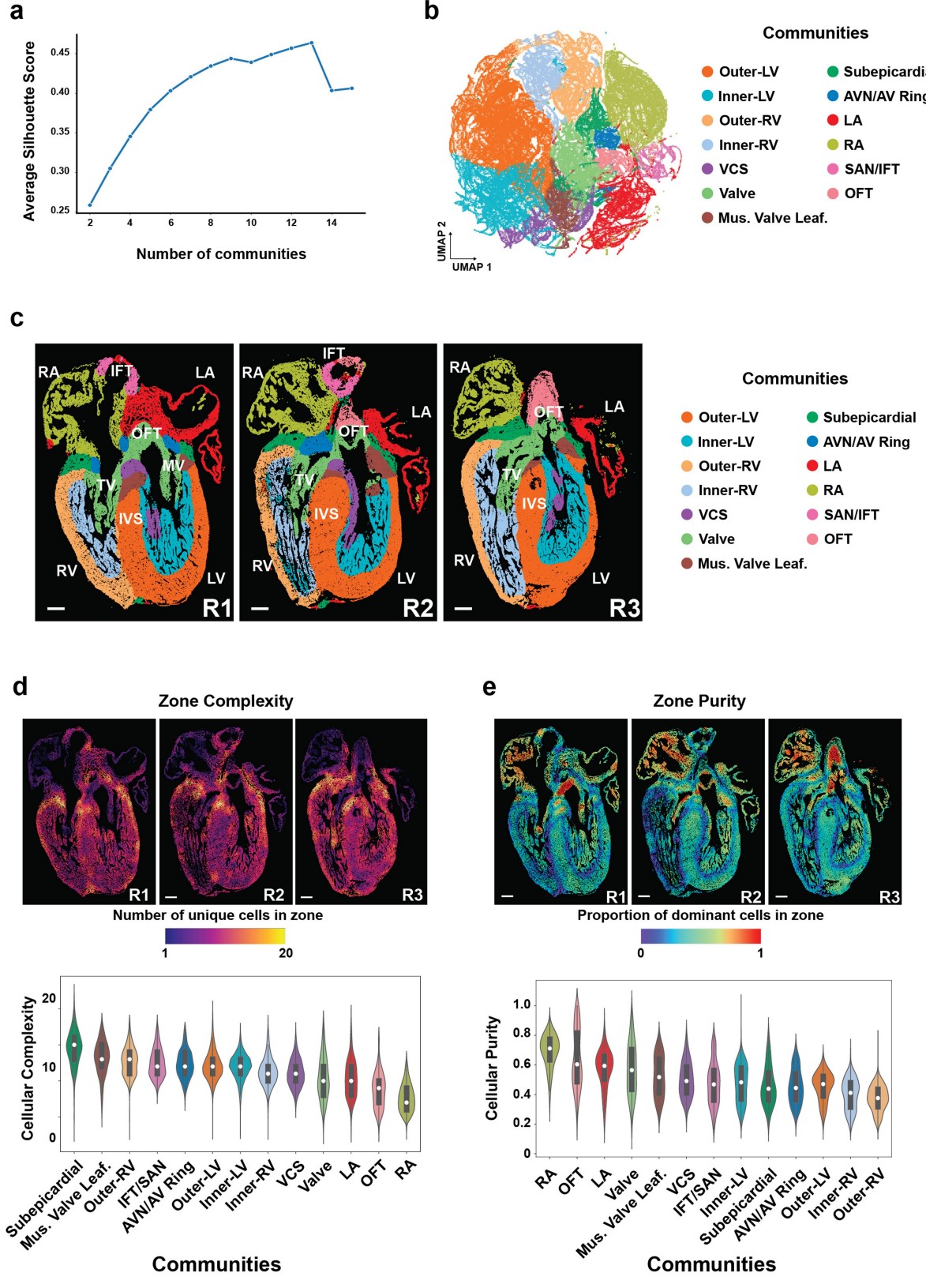

**Extended Data Fig. 5** | See next page for caption.

**Extended Data Fig. 5 | Cell zone analyses reveal the complexity and purity of the cellular communities (CCs). a**, Plot of average silhouette scores reveals that the statistically optimal number of cellular communities is thirteen. **b**, ~250,000 cell zones were grouped into specific cellular communities as shown by UMAP and colored by community. **c**, Spatial mapping of these CCs onto three different sections of the 13 p.c.w. (post conception weeks) heart shows the reproducibility of CCs corresponding to specific anatomic cardiac structures. The distribution of (**d**) cell zone complexity and (**e**) purity is displayed both spatially for replicate sections of 13 p.c.w. hearts (zone complexity/purity maps) and quantitatively in violin plots. The center white dot represents the median, the bold black line represents the interquartile range, and the edges define minima and maxima of the distribution. AVC, atrioventricular canal; AVN, atrioventricular node; IFT, inflow tract; IVS, interventricular septum; LA, left atrium; LV, left ventricle; Mus. Valve Leaf., muscular valve leaflet; MV, mitral valve; OFT, outflow tract; RA, right atrium; RV, right ventricle; SAN, sinoatrial node; TV, tricuspid valve; VCS, ventricular conduction system. Scale bar, 250 μm.

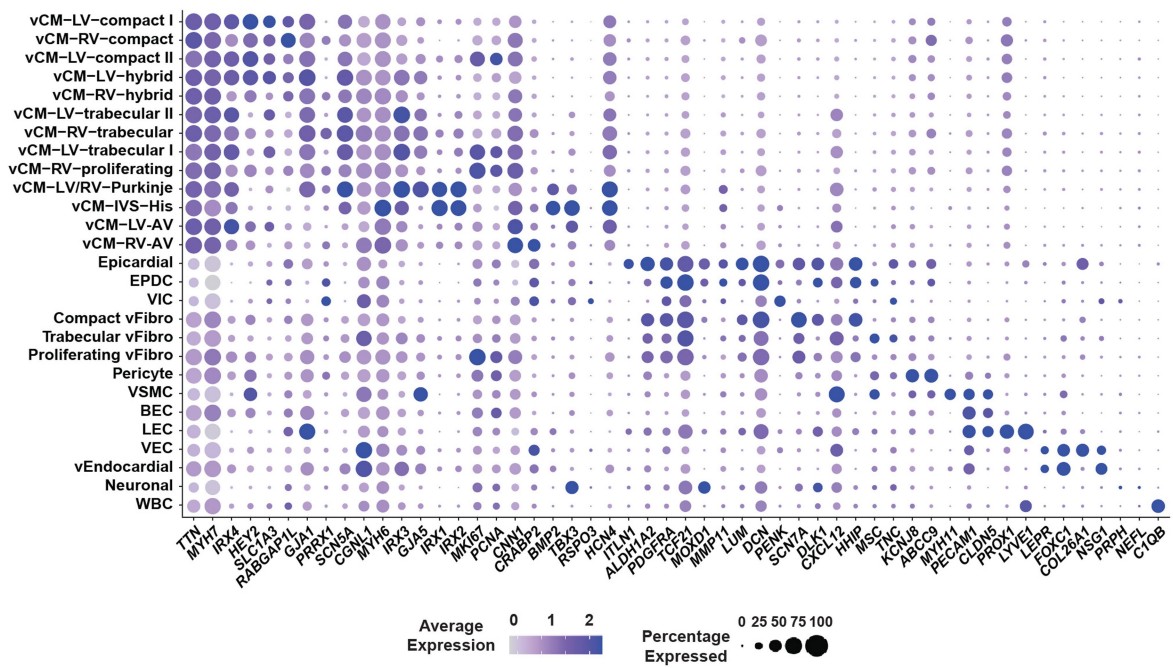

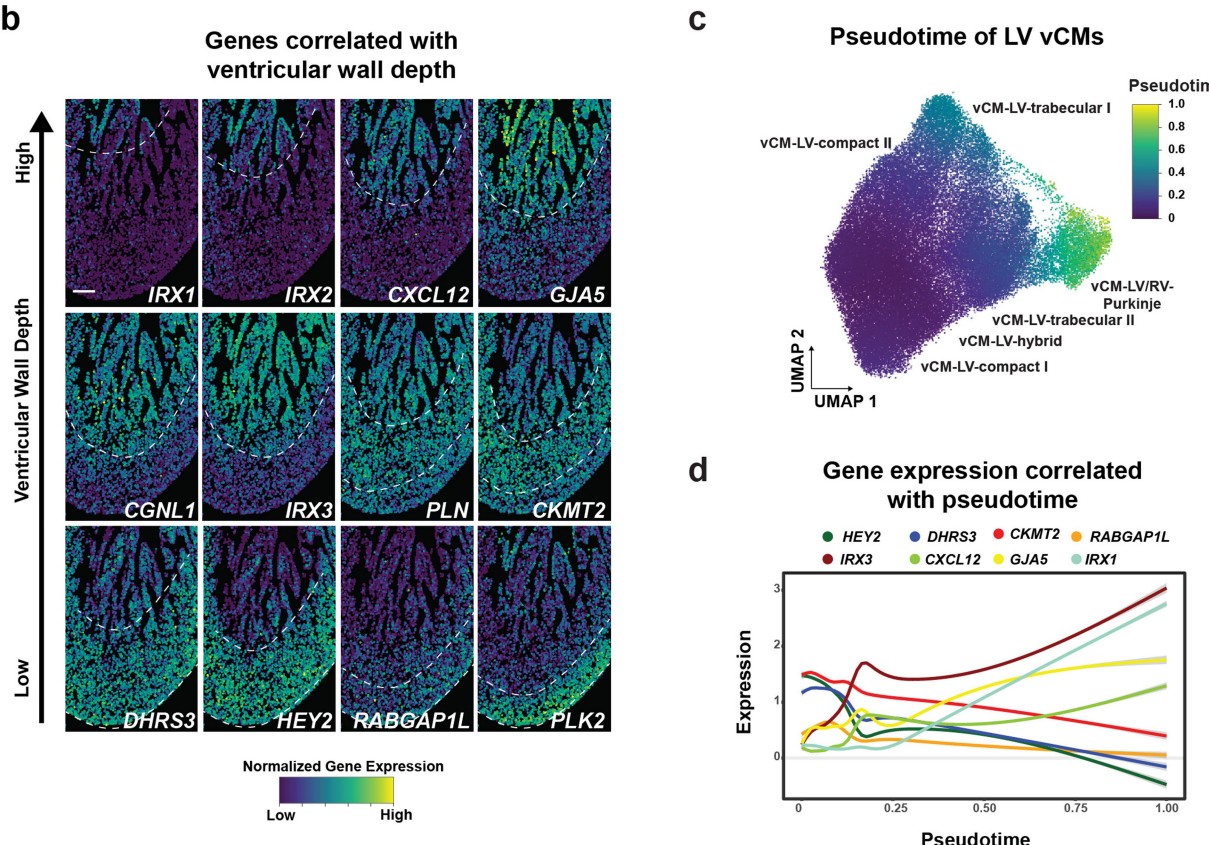

**Extended Data Fig. 6 | Gene marker analysis defined distinct ventricular MERFISH cells and their molecular relationship to ventricular wall depth and pseudotime. a**, Gene marker analysis defined MERFISH cells clustered from only the ventricles. **b**, MERFISH images reveal that spatial expression of genes related to specific vCMs correlate with ventricular wall depth. **c**, UMAP shows pseudotime of these vCMs within the left ventricular wall. **d**, Gene expression of specific markers for each distinct vCM is plotted along the pseudotime axis. Colored lines indicate each gene examined (see legend above plots). BEC, blood endothelial cell; EPDC, epicardial-derived cell; IVS, interventricular septum; LEC, lymphatic endothelial cell; LV, left ventricle; RV, right ventricle; vCM, ventricular cardiomyocyte; vCM-LV/RV-AV, muscular valve leaflet vCM; VEC, valve endocardial cell; vEndocardial, ventricular endocardial; vFibro, ventricular fibroblast; VIC, valve interstitial cell; VSMC, vascular smooth muscle cell; WBC, white blood cell. Scale bar, 250 μm.

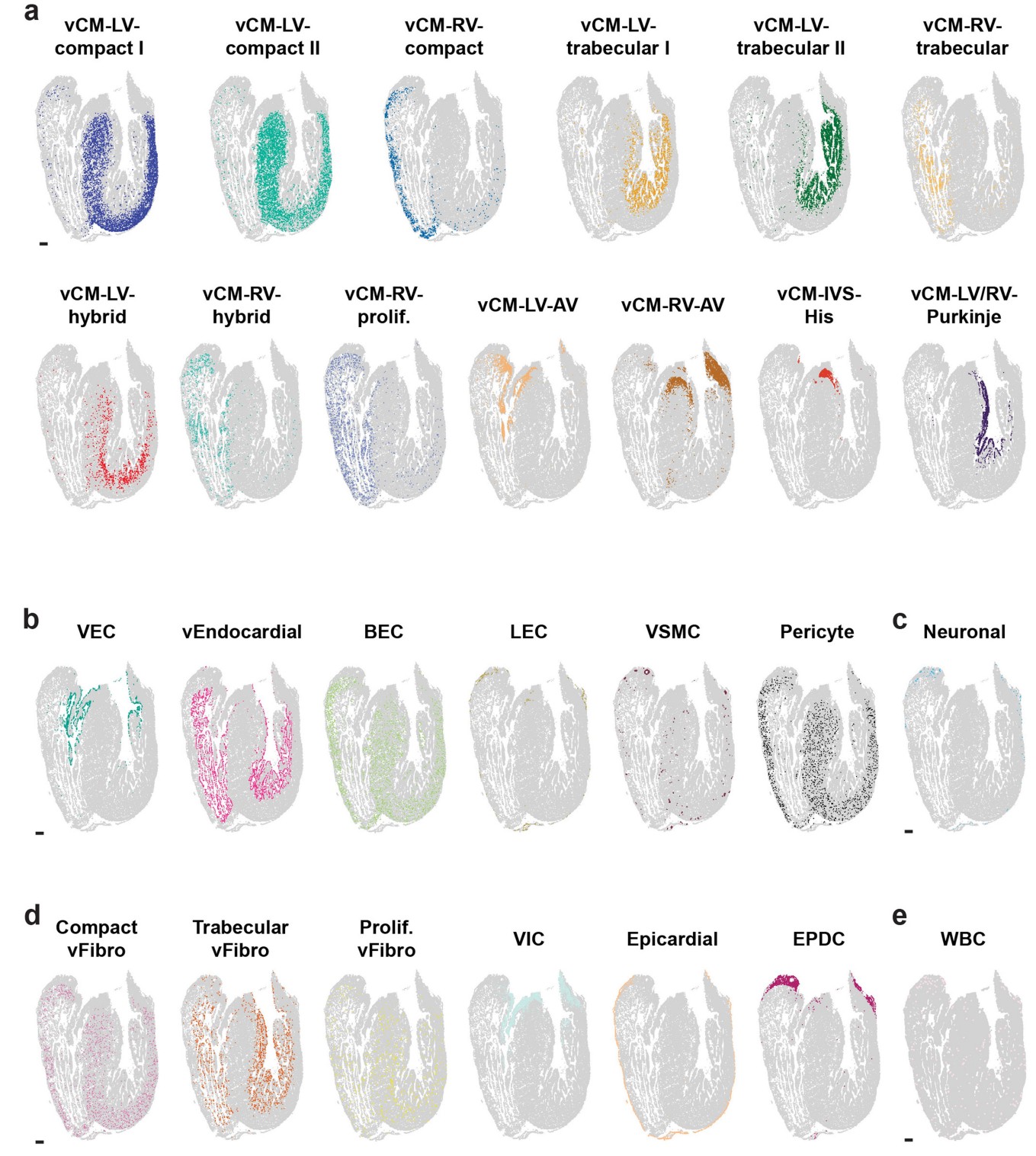

**Extended Data Fig. 7 | Distinct ventricular MERFISH cells map to specific regions of the developing human ventricle.** The spatial mapping of each identified ventricular MERFISH cell is displayed accordingly: **a**, cardiomyocyte related cells, **b**, vascular support related cells, **c**, neuronal cells, **d**, epicardial, EPDC, and fibroblast-related cells, and **e**, WBC related cells. BEC, blood endothelial cell; EPDC, epicardial-derived cell; IVS, interventricular septum; LEC, lymphatic endothelial cell; LV, left ventricle; Prolif., proliferating; RV, right ventricle; vCM, ventricular cardiomyocyte; vCM-LV/RV-AV, muscular valve leaflet vCM; VEC, valve endocardial cell; vEndocardial, ventricular endocardial; vFibro, ventricular fibroblast; VIC, valve interstitial cell; VSMC, vascular smooth muscle cell; WBC, white blood cell. Scale bar, 250 μm.

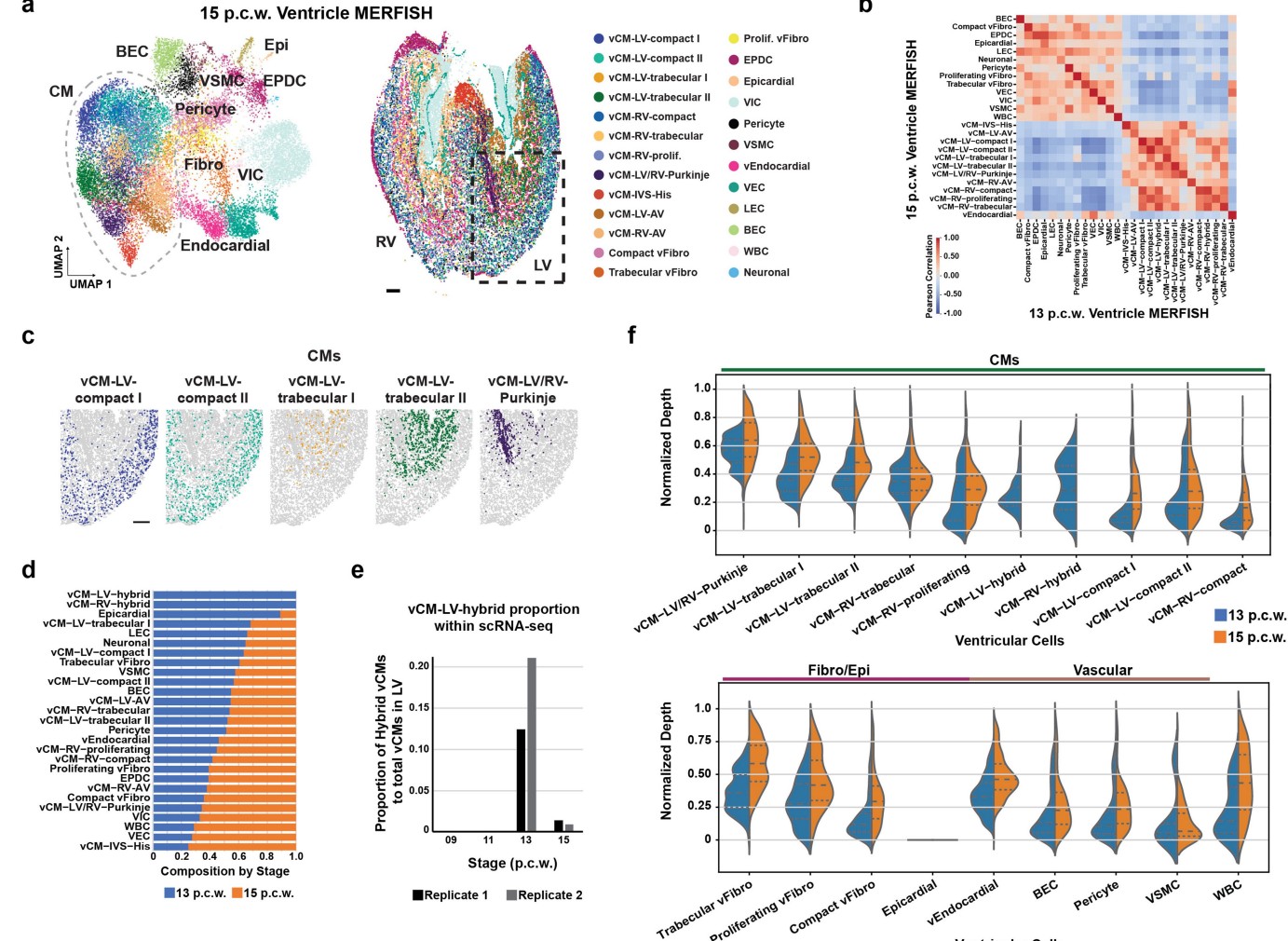

**Extended Data Fig. 8 | MERFISH imaging of 15 p.c.w. ventricles reveals how hybrid vCM subpopulations may dynamically change during development.** **a**, MERFISH cells composing 15 post conception weeks (p.c.w.) ventricles were clustered as displayed on UMAP (left), and the identified ventricular cells were spatially mapped onto the ventricles and labeled in legend (right). **b**, Heatmap of transcriptional correlation between the MERFISH ventricular subpopulations shows that the 15 p.c.w. MERFISH dataset contained all cardiac cells previously identified by the 13 p.c.w. MERFISH dataset, except for the vCM-LV-Hybrid and vCM-RV-Hybrid cardiac cell subpopulations. **c**, The spatial distribution of specific ventricular cardiomyocytes is shown for the left ventricular wall from region outlined in MERFISH spatial map (**a**). **d**, Bar graph shows the relative cell composition of 13 p.c.w. and 15 p.c.w. ventricles. **e**, Bar graph of hybrid vCMs identified at specific scRNA-seq developmental stages reveals the proportion

of hybrid vCMs to total vCMs in the LV from 9–15 p.c.w. **f**, Violin plots show the comparison of normalized ventricular wall depths of distinct ventricular cells within the apical/free walls at 13 p.c.w. and 15 p.c.w. The center dashed line represents the median, the other two dashed lines represent the interquartile range, and the edges define minima and maxima of the distribution. aFibro, atrial fibroblast; BEC, blood endothelial cell; EPDC, epicardial-derived cell; Fibro, fibroblast; IVS, interventricular septum; LEC, lymphatic endothelial cell; LV, left ventricle; Prolif., proliferating; RV, right ventricle; vCM, ventricular cardiomyocyte; vCM-AV, muscular valve leaflet vCM; vCM-LV/RV-AV, muscular valve leaflet vCM; VEC, valve endocardial cell; vEndocardial, ventricular endocardial; vFibro, ventricular fibroblast; VIC, valve interstitial cell; VSMC, vascular smooth muscle cell; WBC, white blood cell. Scale bar, 250 μm.

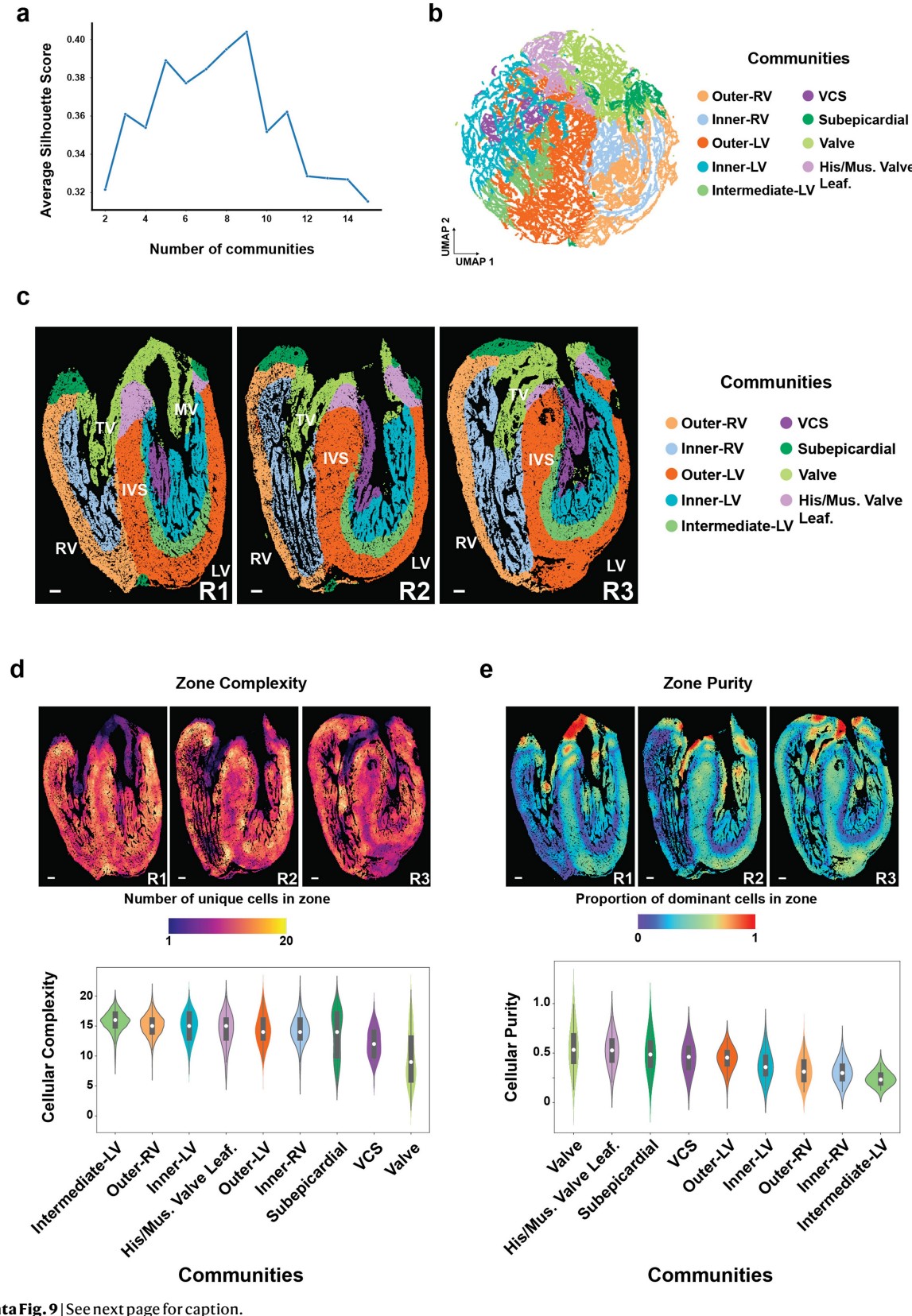

**Extended Data Fig. 9** | See next page for caption.

**Extended Data Fig. 9 | Cell zone analyses of distinct ventricular cells reveal the complexity and purity of ventricle cellular communities (CCs).** **a**, Plot of average silhouette scores shows that the statistically optimal number of cellular communities is nine for identified ventricular cells. **b**, ~180,000 ventricular cell zones were clustered into specific ventricular cellular communities as shown by UMAP and colored by community. **c**, Spatial mapping of these CCs onto three different sections of the 13 p.c.w. (post conception weeks) heart shows the reproducibility of CCs corresponding to specific anatomic cardiac ventricular structures. The distribution of (**d**) cell zone complexity and (**e**) purity is displayed both spatially for replicate sections of the 13 p.c.w. hearts (zone complexity/purity maps) and quantitatively in violin plots. The Intermediate-LV CC exhibits the highest cellular complexity and lowest cellular purity. The center white dot represents the median, the bold black line represents the interquartile range, and the edges define minima and maxima of the distribution. His/Mus. Valve Leaf., bundle of His and the muscular valve leaflet; IVS, interventricular septum; LV, left ventricle; MV, mitral valve; RV, right ventricle; TV, tricuspid valve; VCS, ventricular conduction system. Scale bar, 250 μm.

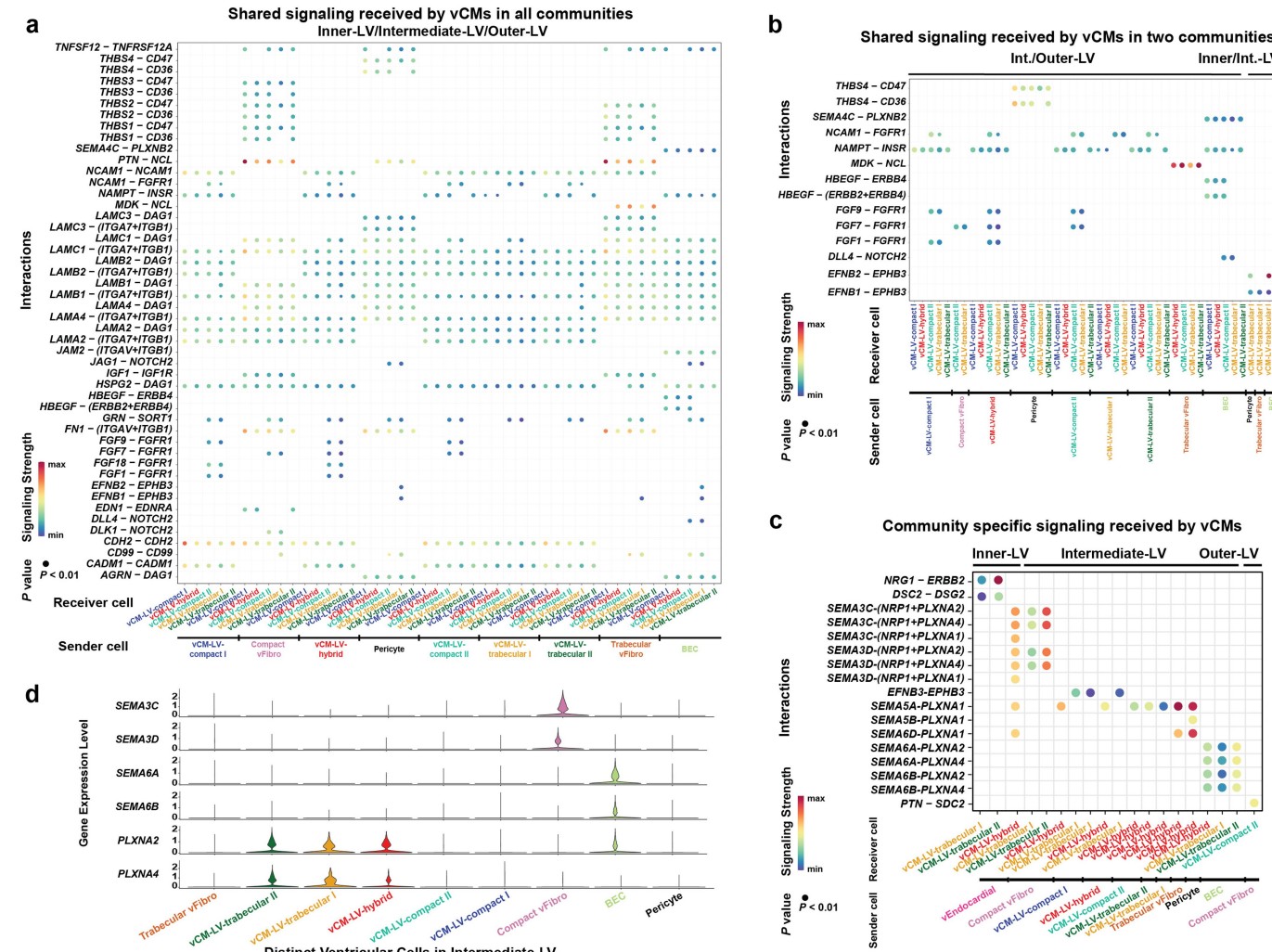

**Extended Data Fig. 10 | Ventricular cardiomyocytes interact with distinct ventricular cells to receive signals that may be specific or shared for the left ventricle (LV) cell community (CC) layers. a**, Dot plot shows the interactions received by specific vCMs within the Inner-LV, Intermediate-LV, and Outer-LV CC layers. The dots are colored by signaling strength and based on the expression of the ligand and cognate receptor. **b**, Dot plot shows shared interactions received by specific vCMs within the Inner-LV/Intermediate-LV and Intermediate-LV/Outer-LV CCs. **c**, Dot plot shows and compares specific interactions received by specific vCMs within the Inner-LV, Intermediate-LV and Outer-LV CC layers. **d**, Violin plots show the expression of specific plexins and semaphorins for each distinct ventricular cell within the Intermediate-LV CC. BEC, blood endothelial cell; CC cellular community; Int., intermediate; LV, left ventricle; vCM, ventricular cardiomyocyte; vEndocardial, ventricular endocardial; vFibro, ventricular fibroblast.

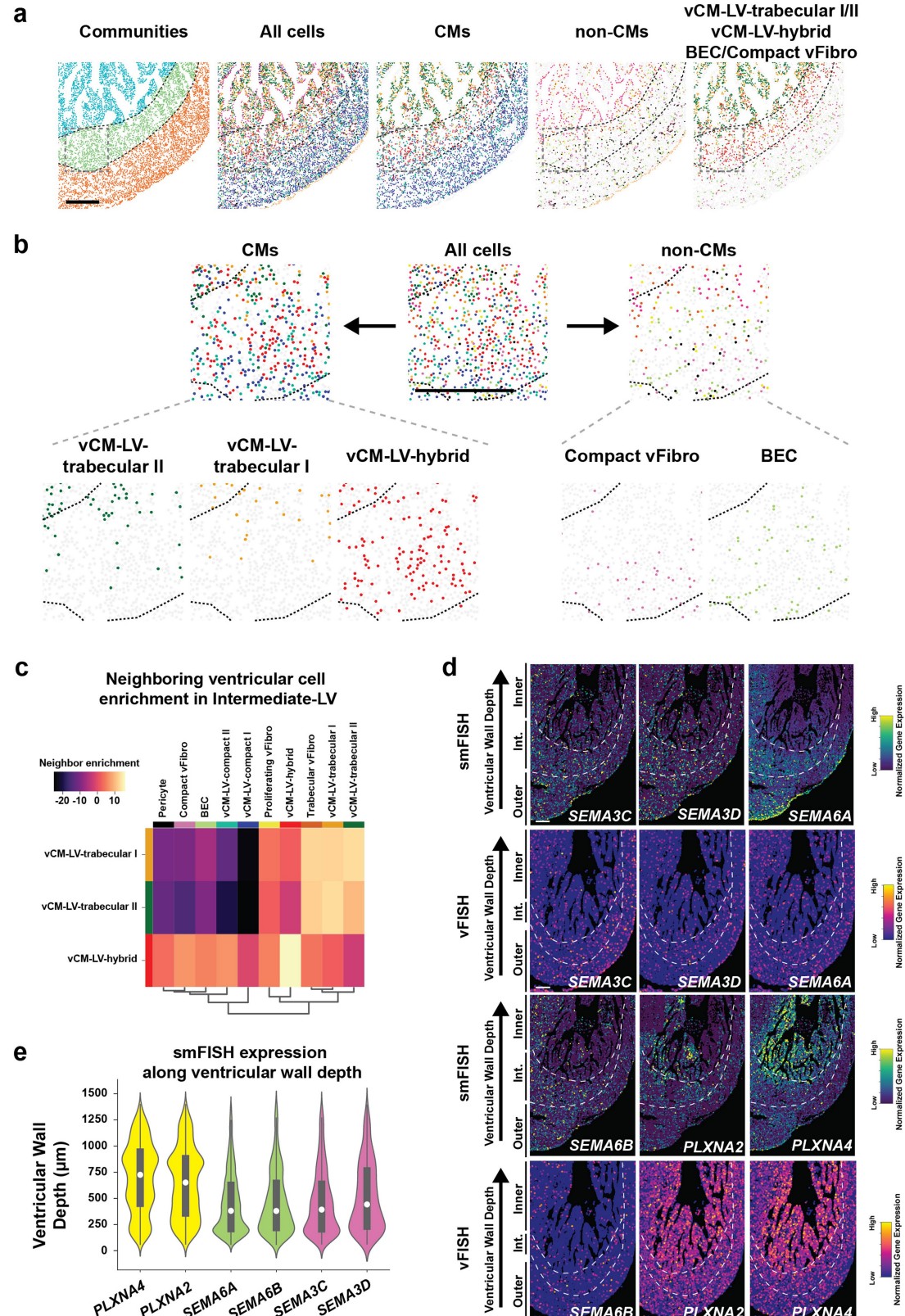

**Extended Data Fig. 11** | See next page for caption.

**Extended Data Fig. 11 | Distinct ventricular cells cooperating in plexin-semaphorin signaling display complementary but overlapping spatial distributions within the ventricular wall. a**, The distribution of distinct ventricular cardiac cells participating in *SEMA3C/3D/6 A/6B - PLXNA2/4* interactions is shown within the left ventricular wall. Cells are colored by community and identity as indicated in Fig. 3b. **b**, Magnified view of boxed area in (**a**) reveals how these cells spatially organize in the Intermediate-LV CC. **c**, Neighborhood enrichment plot of Intermediate-LV CC shows that vCM-LV-Trabecular I, vCM-LV-Trabecular II, vCM-LV-Hybrid are closer to BECs than Compact vFibro. **d**, smFISH and imputed spatial expression (vFISH) analyses show the spatial gene expression of interacting semaphorin ligands and plexin receptors. **e**, Violin plot shows the level of expression (smFISH) for each of the semaphorin ligands and plexin receptors across the ventricular wall depth. The center white dot represents the median, the bold black line represents the interquartile range, and the edges define minima and maxima of the distribution. BEC, blood endothelial cell; CC cellular community; CM, cardiomyocyte; Int., intermediate; LV, left ventricle; smFISH, single molecule fluorescent in situ hybridization; vCM, ventricular cardiomyocyte; vFibro, ventricular fibroblast; vFISH, virtual fluorescent in situ hybridization. Scale bar, 250 μm.

**a**

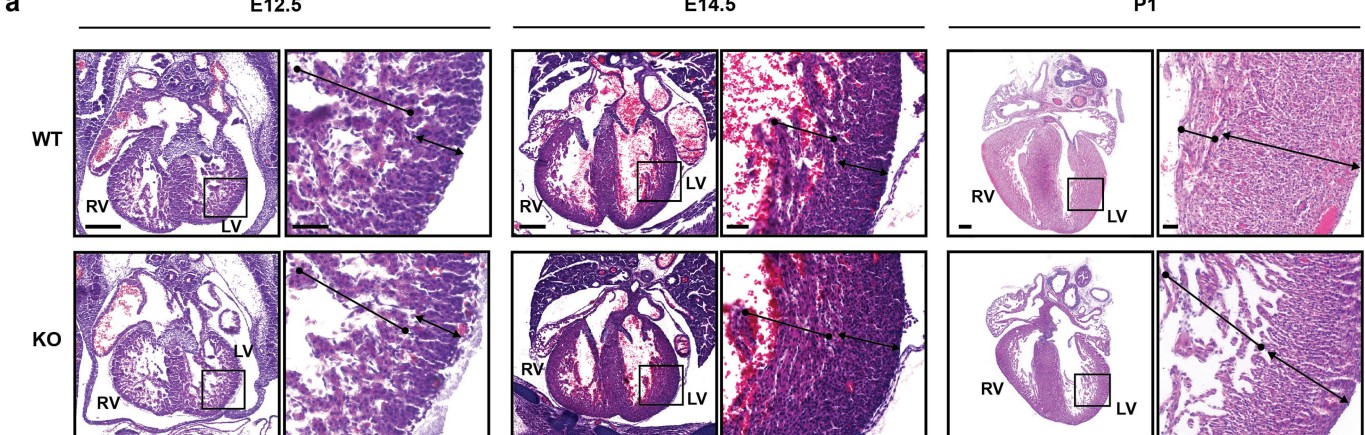

**b**

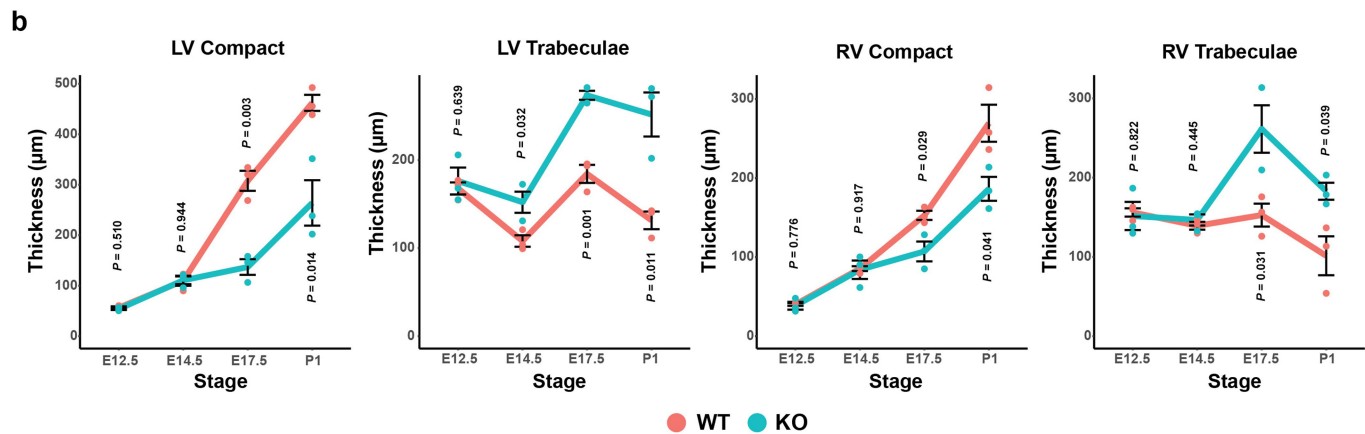

● WT  ● KO

**Extended Data Fig. 12 | *Tcf21-creERT2;Sema3c^fl/fl* knockout mice display hypertrabeculation and relatively thin compact myocardium. a**, Representative hematoxylin and eosin stained frontal sections of hearts from *Tcf21-creERT2;Sema3c^fl/fl* knockout mice at indicated stages show that deletion of *Sema3c* in *Tcf21+* cells starting at E10.5 leads to a progressive cardiac ventricular wall noncompaction phenotype (i.e., hypertrabeculation and thinner compact myocardium), which continues postnatally. Scale bar, 250 μm (50 μm in inset). **b**, Graphs show the thickness of compact and trabecular myocardium from E12.5 to P1. *N* = 3 mice per condition. KO, knockout; LV, left ventricle; RV, right ventricle; WT, wildtype. Error bars are s.e.m. *P* values determined by one-way ANOVA.

                           Neil C. Chi

# Reporting Summary

## Statistics

For all statistical analyses, confirm that the following items are present in the figure legend, table legend, main text, or Methods section.

| n/a | Confirmed | |
|---|---|---|
| ☐ | ☒ | The exact sample size (*n*) for each experimental group/condition, given as a discrete number and unit of measurement |
| ☐ | ☒ | A statement on whether measurements were taken from distinct samples or whether the same sample was measured repeatedly |
| ☐ | ☒ | The statistical test(s) used AND whether they are one- or two-sided<br>*Only common tests should be described solely by name; describe more complex techniques in the Methods section.* |
| ☐ | ☒ | A description of all covariates tested |
| ☐ | ☒ | A description of any assumptions or corrections, such as tests of normality and adjustment for multiple comparisons |
| ☐ | ☒ | A full description of the statistical parameters including central tendency (e.g. means) or other basic estimates (e.g. regression coefficient) AND variation (e.g. standard deviation) or associated estimates of uncertainty (e.g. confidence intervals) |
| ☐ | ☒ | For null hypothesis testing, the test statistic (e.g. $F$, $t$, $r$) with confidence intervals, effect sizes, degrees of freedom and $P$ value noted<br>*Give P values as exact values whenever suitable.* |
| ☒ | ☐ | For Bayesian analysis, information on the choice of priors and Markov chain Monte Carlo settings |
| ☒ | ☐ | For hierarchical and complex designs, identification of the appropriate level for tests and full reporting of outcomes |
| ☒ | ☐ | Estimates of effect sizes (e.g. Cohen's *d*, Pearson's *r*), indicating how they were calculated |

*Our web collection on statistics for biologists contains articles on many of the points above.*

## Software and code

Policy information about availability of computer code

| | |
|---|---|
| Data collection | Fastqs collected from 10X Genomics Chromium Single Cell 3' Expression kit were mapped with Cell Ranger v3.0.1 (10X Genomics). MERFISH imaging data was collected using custom Python (v3.9) code to control the microscope (available here from the Zhuang Lab: https://github.com/ ZhuangLab). Quantitative Real-Time PCR data was collected with CFX Manager v3.1. Data from mouse heart sections were collected with NDP View 2 software (Hamamatsu) and QuPath v0.4.3. |
| Data analysis | The pipeline for generating the encoding probes used in the MERFISH studies is available from: https://github.com/bil022/ProbeDesign. The pipeline for processing the MERFISH dataset including cell segmentation and assigning barcodes to cells is available from: https://github.com/ epigen-UCSD/merfish_tools. Custom code used for analyzing the scRNA-seq and MERFISH datasets in this study is available from: https://github.com/ChiLab-UCSD/Heart_MERFISH_analysis. Other packages used in data analysis include: Cellpose (v1.0.2); Seurat (v4.0.1); DoubletFinder (v2.0); SCCAF (v0.0.10); scArches (v0.5.9); scVI (v1.0.3); pySCENIC (v0.12.1); CellChat (v1.6.1); Waddington-OT (v1.0.8); MERlin (v0.6.1); Scanpy (v1.8); scikit-learn (v0.22); python (v3.9); R (v4.2.0) |

For manuscripts utilizing custom algorithms or software that are central to the research but not yet described in published literature, software must be made available to editors and reviewers. We strongly encourage code deposition in a community repository (e.g. GitHub). See the Nature Portfolio guidelines for submitting code & software for further information.

# Data

Policy information about availability of data

All manuscripts must include a data availability statement. This statement should provide the following information, where applicable:
- Accession codes, unique identifiers, or web links for publicly available datasets
- A description of any restrictions on data availability
- For clinical datasets or third party data, please ensure that the statement adheres to our policy

Data availability statement is included in the manuscript, which states:
Raw sequencing data for the in vivo studies is available from dbGAP under accession number (phs002031). Raw sequencing data for the in vitro studies is available from CIRM CESCG (https://cirm.ucsc.edu) under accession number (chiCardiomyocyte1). Processed scRNA-seq data is accessible on the UCSC Cell Browser (https://cells.ucsc.edu/?ds=hoc). MERFISH imaging data is available through Dryad (doi:10.5061/dryad.w0vt4b8vp).  The human reference genome sequences (hg38) can be downloaded from ncbi_refseq: https://hgdownload.soe.ucsc.edu/goldenPath/hg38/bigZips/genes/.

# Human research participants

Policy information about studies involving human research participants and Sex and Gender in Research.

| | |
|---|---|
| Reporting on sex and gender | Sex and gender were not considered in the study design and were not collected. |
| Population characteristics | Eleven hearts between the ages of 9 to 16 post conception weeks. Age was considered as a covariate-relevant population characteristic. |
| Recruitment | The heart samples were collected by the University of California, San Diego (UCSD) Perinatal Biorepository's Developmental Biology Resource (DBR). All donors gave informed consent for the collection of these tissues by medical termination. Age of the sample was measured using the crown rump length (CRL) method, and all samples were screened for and found to be absent of structural fetal abnormalities. Tissue samples were collected and transported in buffer containing 10 mM HEPES pH 7.8, 130 mM NaCl, 5 mM KCl, 10 mM Glucose, 10 mM 2,3-Butanedione monoxime (BDM), 10 mM Taurine, 1 mM EDTA, and 0.5 mM NaH2PO4, and overall morphology was checked under a stereotaxic dissection microscope (Leica). |
| Ethics oversight | Heart samples were collected in strict observance of the legal and institutional ethical regulations. The heart samples were collected under a University of California, San Diego (UCSD) Human Research Protections Program Committee Institutional Review Board (IRB)-approved protocol (IRB #081510) by the UCSD Perinatal Biorepository's Developmental Biology Resource (DBR), and all experiments were performed within the guidelines and regulations set forth by the IRB (IRB #101021, registered with the DBR). Ethical requirements for data privacy include that sequence-level data (e.g. fastq files) be shared through controlled-access databases. |

Note that full information on the approval of the study protocol must also be provided in the manuscript.

# Field-specific reporting

Please select the one below that is the best fit for your research. If you are not sure, read the appropriate sections before making your selection.

☒ Life sciences          ☐ Behavioural & social sciences          ☐ Ecological, evolutionary & environmental sciences

For a reference copy of the document with all sections, see nature.com/documents/nr-reporting-summary-flat.pdf

# Life sciences study design

All studies must disclose on these points even when the disclosure is negative.

| | |
|---|---|
| Sample size | Sample size of at least two was chosen to provide sufficient material for scRNA-seq assays, and ensure replication of the results with affordable cost. For MERFISH, three replicate sections from a 13 post conception week heart and one section of 15 post conception week ventricles were imaged for MERFISH, providing a total of ~280,000 cells which was provided a sufficient number of single-cell profiles and gave sufficient statistics for the effect sizes of interest. All other experiments, including hPSC and mouse experiments, have at least three independent biological replicates which gave sufficient statistics for the effect sizes of interest. Sample sizes were not predetermined utilizing statistical methods and sample size was determined empirically. |
| Data exclusions | No data was excluded. |
| Replication | Reported scRNA-seq results were replicated from two biological replicates for each stage of development. Reported MERFISH results were replicated using three biological sections from one 13 post conception week heart, and reported ventricle results from additional 15 post conception week ventricles, and correlation analyses were conducted to ensure the consistency between the replicates. Reported mouse results were replicated from three animals under each condition. All attempts at replication were successful. |
| Randomization | Data for scRNA-seq and MERFISH was collected from all available samples and no randomization was necessary. For the studies utilizing |

| Randomization | human pluripotent stem cell lines, treatment with NRG1 was randomly assigned. For the animal studies, animals were randomly chosen from each genotype and timepoint. |
|---|---|
| Blinding | The investigators were not blinded during collection as no subjective measurements were taken. Blinding during analysis was not necessary as all of the results were analyzed with the use of unbiased analysis and software tools that are not affected by the sample. |

# Reporting for specific materials, systems and methods

We require information from authors about some types of materials, experimental systems and methods used in many studies. Here, indicate whether each material, system or method listed is relevant to your study. If you are not sure if a list item applies to your research, read the appropriate section before selecting a response.

## Materials & experimental systems

| n/a | Involved in the study |
|---|---|
| ☐ | ☒ Antibodies |
| ☐ | ☒ Eukaryotic cell lines |
| ☒ | ☐ Palaeontology and archaeology |
| ☐ | ☒ Animals and other organisms |
| ☒ | ☐ Clinical data |
| ☒ | ☐ Dual use research of concern |

## Methods

| n/a | Involved in the study |
|---|---|
| ☒ | ☐ ChIP-seq |
| ☐ | ☒ Flow cytometry |
| ☒ | ☐ MRI-based neuroimaging |

## Antibodies

| Antibodies used | Antibody: Alexa Fluor® 647 Mouse Anti-Cardiac Troponin T<br>Supplier: BD Biosciences<br>Cat No: 565744<br>Clone: 13-11 (RUO) |
|---|---|
| Validation | Alexa Fluor® 647 Mouse Anti-Cardiac Troponin T: https://www.bdbiosciences.com/en-eu/products/reagents/flow-cytometry-reagents/research-reagents/single-color-antibodies-ruo/alexa-fluor-647-mouse-anti-cardiac-troponin-t.565744 |

## Eukaryotic cell lines

Policy information about cell lines and Sex and Gender in Research

| Cell line source(s) | H9-hTnnTZ-pGZ-D2 human pluripotent stem cell (hPSC) line was purchased from WiCell. An additional TNNT2:NLS-mKATE2-T2A-BsdR RUES2 hPSC cardiomyocyte reporter line was generated that specifically expresses the mKATE2 fluorescent protein containing a nuclear localization signal (NLS-mKATE2) in differentiated cardiomyocytes, as detailed in the methods. |
|---|---|
| Authentication | H9-hTnnTZ-pGZ-D2 and TNNT2:NLS-mKATE2-T2A-BsdR RUES2 hPSC reporter transgenic lines were authenticated with Short Tandem Repeat (STR) profiling analysis and immunofluorescence. |
| Mycoplasma contamination | Cell lines tested negative for mycoplasma contamination by PCR |
| Commonly misidentified lines (See ICLAC register) | None used in this study |

## Animals and other research organisms

Policy information about studies involving animals; ARRIVE guidelines recommended for reporting animal research, and Sex and Gender in Research

| Laboratory animals | The information of Tcf21-CreERT2; Sema3c fl/fl mice are included in the manuscript, within the methods.<br>All animals used for timed matings were aged 8-10 weeks (female) or 8-10 weeks (male) of age. E12.5, E14.5, E17.5, and P1 mouse embryos were collected for histological analysis. Mice were housed on a 12 hour light/dark cycle (6am-6pm light cycle), with a temperature between 20-22 degrees Celsius, and a humidity range of 30-70%. |
|---|---|
| Wild animals | The study did not involve wild animals. |
| Reporting on sex | Both male and female embryos were used in this study; the embryos were not genotyped to determine the sex. |
| Field-collected samples | The study did not involve samples collected from the field. |
| Ethics oversight | All protocols concerning animal use were approved by the Institutional Animal Care and Use Committee (IACUC) at UCSD and were accredited by the Association for Assessment and Accreditation of Laboratory Animal Care (AAALAC). |

Note that full information on the approval of the study protocol must also be provided in the manuscript.

## Flow Cytometry

### Plots

Confirm that:

☒ The axis labels state the marker and fluorochrome used (e.g. CD4-FITC).

☒ The axis scales are clearly visible. Include numbers along axes only for bottom left plot of group (a 'group' is an analysis of identical markers).

☒ All plots are contour plots with outliers or pseudocolor plots.

☒ A numerical value for number of cells or percentage (with statistics) is provided.

### Methodology

| | |
|---|---|
| Sample preparation | As described in the Methods section. Briefly, the single cells were dissociated and were resuspended in PBS supplemented with 5% FBS and sorted on a Sony SH800 sorter. |
| Instrument | Sony SH800 sorter |
| Software | Proprietary Sony SH800 Software and FlowJo (v10) |
| Cell population abundance | Not applicable because we sorted as many live single cells as necessary to complete downstream scRNA-seq processing |
| Gating strategy | Single cells were gated based on SSC and BSC. Live cells were gated based on DAPI (DAPI negative). |

☒ Tick this box to confirm that a figure exemplifying the gating strategy is provided in the Supplementary Information.

