## [Peer Review File · Nature]

Manuscript Title: Spatially-Organized Cellular Communities Form the Developing Human Heart

Reviewer Comments & Author Rebuttals

Reviewer Reports on the Initial Version:

Referees' comments:

Referee #1 (Remarks to the Author):

In this manuscript the authors used scRNA-Seq combined with MERFISH analysis and a computational approach to generate a spatiotemporal gene expression profile with cellular resolution in the developing human heart. The combination of these 2 approaches is novel and can be used to show the local distribution of the newly identified subpopulations of cell types.

It is a very well written and clearly presented manuscript that uses human hearts to identify the spatial distribution of several subpopulations of cells. Despite it providing interesting data, the study remains fairly descriptive or predictive (in case on the ligand-receptor paring), with a very limited amount of validation experiments.

Some specific comments

- The human samples that were used in the study should be described in more detail. Since the authors aim to make a statement regarding the identity of cardiac cell types in the developing hearts it would be good to look at gene expression changes at different ages to see whether the different populations indeed change over time. MERFISH could then also be used to determine whether age induces changes in the local distribution of the different subpopulations of cells.
- Validation experiments for known marker genes are lacking and should be used to confirm the validity of the subpopulations of cells.
- The authors should perform some validation experiments in specific regions of adult heart tissue under both healthy and stressed conditions.
- It remains unclear how the authors decided on the 238 genes that were studied by MERFISH. This should be explained in more detail.
- The authors used computational approaches to identify distinct cell-types and what they call distinct 'cellular communities'. These approaches appear powerful to identify specific subpopulations such as conduction system cardiomyocytes vs ventricular cardiomyocytes. However, it is unclear whether the subcommunities in the ventricular wall represent meaning biologically distinct regions with borders, or are more of an artificial subdivision of cells that form a continuous transmural transition of gene expression profiles.
- The mouse study provides functional evidence for the biological relevance of Sema3c – plexin signaling during ventricular wall morphogenesis. However, this model should be studied in more detail. Can the authors provide a comprehensive description of the phenotypic analysis of the mutant mice? Mutant hearts appear to show developmental delay, this could be an alternative cause of differential thickness

of compact and trabecular layers. At what stage is the difference in wall thickness first detectable?

Referee #2 (Remarks to the Author):

Cardiac cell diversity plays an important role in the development of the heart. Molecular and cellular interaction between diverse cardiac cell types is essential for the creation of complex morphological structures critical for heart development and functions. The present study by Farah et al. aims to characterize the identity of cardiac cell types in the developing human heart. Authors performed single-cell RNA sequencing (scRNA-seq) on human hearts between 9 to 16 post-conception weeks (PCW) in replicate. Authors have also tried to integrate the single cell RNA-sequencing with high-resolution multiplexed error-robust fluorescent in situ hybridization (MERFISH) to identify the cellular communities contributing to the formation of distinct cardiac structures. This is an interesting manuscript with a nice characterization of these different cell populations in the heart. However, multiple studies have performed single-cell RNA-sequencing to transcriptionally define distinct cell populations of the human heart (PMID: 32971526, PMID: 32403949, PMID: 33990324, PMID: 30759401, PMID: 31835037, PMID: 33184181). Considering these published works, the new information generated in the current manuscript is important but not very relevant. Most of these cell populations are defined at the transcription level and that is not very reliable considering scRNA-seq studies require tissue and cell dissociations causing loss of cell identity. More functional and molecular characterization is needed to suggest that they are rare (new) cell populations of the hearts.

This is an interesting manuscript, but several issues need to be addressed. Some of the major issues are described below.

1. What if the diversity seen in the cardiac population is due to the different stages of human heart development as stages vary between 9 to 16 PCW and expression significantly changes during this early phase of cardiac development?
2. What is the rationale for dissecting the chambers and IVS for scRNA-sequencing? At these stages the hearts are very small so why not use the whole heart single cell suspension and perform scRNA-sequencing in an unbiased manner?
3. Authors claim that they have sequenced a relatively high number of cells compared to prior cardiac developmental studies^{10–12}, enabling the identification of more rare cardiomyocytes including BMP2+ non-chamber cardiomyocytes²⁰ (ncCM). BMP2+ cells have been characterized previously as AVC populations (PMID: 27840109). It is not clear, what new information has been generated here if they are related to previously described cardiac cell populations and their spatiotemporal location.
4. Similarly, authors show that non-chambered cardiomyocytes primarily segregated into RSPO3/MSX2+

atrioventricular canal/node (ncCM-AVC-like) and SHOX2/TBX18+ inflow tract/pacemaker cardiomyocytes (ncCM-IFT-like), which differentially expressed ISL1, PITX2, and TBX3, known transcription factors involved in regulating pacemaker sinoatrial node versus inflow tract development. Thus, these findings provide new developmental insight into not only previously reported chamber-related cardiomyocytes^{7–11} but also more specialized cardiomyocytes critical for regulating electrical cardiac conduction.

5. It is not clear why authors performed MERFISH studies on 12-13 PCW human hearts while the scRNA-sequencing was performed on 9 to 16 PCW human hearts.

6. In Figure 1g, Gene imputation performance was validated spatially by comparing normalized gene expression profiles of marker genes measured by MERFISH with the corresponding imputed gene expression profiles. However, these gene expression profiles do not match the previously reported profiles. For example, *Tbx18* is strongly expressed in IVS and ventricular walls and left ventricles (conflicting results published in these two Nature papers, PMID: 19369973 and PMID: 18480752). This raises concerns regarding the reliability of the gene expression profiles measured by MERFISH. Also, only 238 MERFISH target genes were used. Also, the expression of missing genes in MERFISH was determined using transcriptomic data from corresponding scRNA-seq cells

7. Semaphorin-Plexin signaling directs ventricular cardiomyocyte organization is superficially described. Figure 5e, more stages should have been analyzed to see the progress of LVNC. It is not clear why no repulsive effect of Sema6A and 6B was observed on inner-LV cells in Figure 5b but they were able to block Sema3c effects in Figure 5c.

8. Class 3 semaphorins are secreted and Class 6 semaphorins are membrane-bound. It is not clear how Sema6A and 6B were able to block Secreted Sema3c functions.

9. In other cell types such as neural crest, Sema6A and 6B repel cells, while Sema3C attracts cells depending upon the differential expression of Sema6A-B receptor Plexin-A2 and Sema3C receptors Plexin-D1/Neuropilin1-2. In the present study, It is not clear how the same set of receptors PlexinA2/A4 are responding differently (attractive for Sema3C and repulsive for Sema6A/B). What are the underlying mechanisms for this differential response?

Referee #3 (Remarks to the Author):

The authors present a very exciting study on human developmental heart, which is complete with single cell analysis and spatial data (MERFISH). However, their analysis is superficial, and they are missing a great opportunity to characterise more in depth the niches they discovered.

Below my suggestions.

Major comments:

In lines 205 - 206, the authors claimed that they have identified "novel" cell types that are yet to be defined. This sounds contradictory. It is possible that the authors have found cell types that were not identifiable using their methodology rather than a new cell type. If the authors want to claim the discovery of new cell types they should provide evidence, especially if they "discovered" them by clustering. Otherwise, they should reword this.

The authors mention often the term "blood cells" to refer to cells from myeloid and lymphoid lineages. It would be correct to use immune cells, progenitors (both RBC or WBC) than group them into a single "blood" category.

In lines 220 - 223 the authors mention the potential that their discoveries have for cardiac biology. One of them is cell-cell interactions. With the richness of data that they mention, it is rather disappointing to see that they did not perform a CCI analysis, nor inferred any cell-specific GNR analysis when there are so many tools available to do this. Their analysis is mostly based on clustering and marker identification, which is biased. The clustering approach does not provide any metric to assess their accuracy, like SCCAF (<https://www.nature.com/articles/s41592-020-0825-9>). Without a more detailed functional analyses of the cell types they have identified, their study does not warrant support for novelty.

Minor comments:

The authors mention that they generated a comprehensive human heart cell atlas; however, it would be more accurate to say they develop a "comprehensive developmental human heart cell atlas".

It would provide solid proof if the authors would have compared the probes identified with NS-Forest2 with a newer method like SpaPros (<https://github.com/theislab/spapros>).

Author Rebuttals to Initial Comments:

Referees' comments:

Referee #1 (Remarks to the Author):

In this manuscript the authors used scRNA-Seq combined with MERFISH analysis and a computational approach to generate a spatiotemporal gene expression profile with cellular resolution in the developing human heart. The combination of these 2 approaches is novel and can be used to show the local distribution of the newly identified subpopulations of cell types. It is a very well written and clearly presented manuscript that uses human hearts to identify the spatial distribution of several subpopulations of cells. Despite it providing interesting data, the study remains fairly descriptive or predictive (in case on the ligand-receptor paring), with a very limited amount of validation experiments.

We thank the Reviewer for their thoughtful and positive assessment of our manuscript and their comments/suggestions, which have helped strengthen the manuscript.

Some specific comments

1 - The human samples that were used in the study should be described in more detail. Since the authors aim to make a statement regarding the identity of cardiac cell types in the developing hearts it would be good to look at gene expression changes at different ages to see whether the different populations indeed change over time. MERFISH could then also be used to determine whether age induces changes in the local distribution of the different subpopulations of cells.

We thank Reviewer 1 for their inquiry into the age-related gene expression changes within the cardiac cell types of the developing heart. Our analysis of the scRNA-seq data reveals significant age-related changes that occur within specific cardiovascular lineages as evidenced by the presence of distinct cell populations at specific ages (please see in Extended Data Figs. 3-7, the correlation between cell subpopulation dot plot and age bar graph in panel d as well as heatmaps of gene expression to these age-specific cell subpopulations in panel e). For example, ventricular cardiomyocytes (vCM) partitioned into populations correlating to developmental age (vCM-Early and vCM-Late) (Extended Data Fig. 3d, e). To further investigate how different cell populations may change with age, we have also examined their gene expression changes over time by performing a gene regulatory network analysis utilizing WGCNA (Please see response to Reviewer 2, major point 1 and Reviewer 3, major point 3). As a result, these analyses revealed gene modules that correlated with age for each cell class (Extended Data Fig. 9). In particular, we discovered that gene modules which positively correlate with age (i.e., upregulate over time) are related to cell type differentiation and function (e.g., muscle contraction for cardiomyocytes, collagen fibril organization for fibroblasts). On the other

hand, gene modules that negatively correlate with age (i.e., downregulate over time) are related to general cell processes such as splicing, translation and cell cycle. These findings are consistent with our scRNA-seq findings that revealed that many identified cardiac lineages exhibit limited proliferation over time while becoming more specialized in order to accommodate the function of each cardiac structure (Extended Data Figs. 3-5). Finally, in addition to our original MERFISH imaging of 13 PCW hearts, we also performed MERFISH imaging in 15 PCW hearts to spatially corroborate our age-related scRNA-seq findings and to determine whether age may lead to changes in the spatial distribution of different cell subpopulations. In contrast to 13 PCW ventricles, we particularly discovered that 15 PCW ventricles did not contain hybrid vCM subpopulations in either the left or right ventricles (Extended Data Fig. 19a-d). Corresponding to this disappearance of hybrid vCMs, we observed that compact vCMs extended further across the ventricular wall depth, whereas trabecular vCMs appeared to be located closer to the lumen of 15 PCW ventricles when compared to 13 PCW ventricles (Extended Data Fig. 19f). Thus, we have included these additional age-related WGCNA and MERFISH analyses to further support our findings (p. 11, lines 220-228 and p. 25-26, lines 557-570; Extended Data Figs. 9, 10, and 19 and Supplementary Table 4).

2 - Validation experiments for known marker genes are lacking and should be used to confirm the validity of the subpopulations of cells.

We appreciate Reviewer 1's suggestion to perform validation experiments for known marker genes in order to confirm the validity of the subpopulations of cells. To spatially validate the subpopulations of cells identified from our scRNA-seq studies, we performed multiplexed error-robust fluorescent *in situ* hybridization (MERFISH), which is a single-cell genome-scale imaging method (PMID: 30385464, PMID: 34616063, PMID: 35771910). To confirm the MERFISH-based transcriptomic imaging and the associated MERFISH identified cells, we further performed single molecule fluorescent *in situ* hybridization (smFISH) studies for known marker genes of many of the cell subpopulations including cardiomyocytes (*TTN*, *NR2F1*, *MYH7*, *GJA5*), fibroblasts (*DCN*), smooth muscle cells (*MYH11*, *PDGFRB*), epicardial/epicardial-derived cells (*TBX18*), valve interstitial cells (*PENK*), lymphatic endothelial cells (*LYVE1*), and endocardial/blood endothelial cells (*PECAM1*). Comparing MERFISH findings with smFISH results revealed a strong correlation of RNA transcript levels for each identified gene between the two imaging techniques (Pearson correlation coefficient = 0.92, Response Figure 1). Confirming these results, the spatial expression pattern of each marker gene is similar between MERFISH and smFISH imaging studies (Extended Data Fig. 11d). We have included these additional validation experiments to further support our findings (p. 13, lines 262-264; Extended Data Fig. 11d).

Response Figure 1. Correlation analyses of genes imaged by MERFISH and smFISH. a, Pearson correlation of the counts of the 11 genes imaged by MERFISH and smFISH reveals strong correlation between the two imaging methods (Pearson correlation coefficient = 0.92).

3 - The authors should perform some validation experiments in specific regions of adult heart tissue under both healthy and stressed conditions.

We thank Reviewer 1 for suggesting to perform validation experiments in adult heart tissues under both healthy and stressed conditions. Toward this end, we compared our findings with recently published snRNA-seq datasets from healthy/non-failing and diseased adult human heart tissues, which were primarily from the ventricle chamber

[Redacted text]

Because the Chaffin et al. (PMID: 35732739) snRNA-seq data provided the most specific cell population markers, particularly for vCMs, among these recently published adult human heart snRNA-seq datasets, we focused our analyses and comparison between our developing human ventricle scRNA-seq datasets and the Chaffin et al. snRNA-seq datasets (Extended Data Fig. 21). Consistent with previous reports that the adult ventricle comprises primarily compact myocardium/vCMs (PMID: 10737851), we discovered that the majority of healthy/non-failing adult vCMs were compact vCMs as detected by the *RABGAP1L* gene marker, but not necessarily the *HEY2* marker, suggesting that *HEY2* could possibly be a marker for early developing compact vCMs, but *RABGAP1L* may be a marker for more mature adult-like vCMs. On the other hand, trabecular vCMs were not detected in these healthy/non-failing adult vCMs using the trabecular vCM markers, *IRX3* and *GJA5*. Further analyses of diseased/heart failure adult vCMs revealed that these vCMs were also primarily compact vCMs with minimal

presence of trabecular vCMs, as similarly observed in healthy/non-failing vCMs. Thus, we have included these additional validation experiments in adult cardiac ventricles, under both healthy/non-failing and diseased conditions, to further support our findings (p. 26-27, lines 571-588; Extended Data Fig. 21).

[Redacted text and figure]

4 - It remains unclear how the authors decided on the 238 genes that were studied by MERFISH. This should be explained in more detail.

We apologize if we were unclear on how we decided on the 238 genes for our MERFISH studies. To clarify how we chose these genes, we have expanded the “MERFISH gene selection and probe library design and construction” section in the Methods in order to provide a more detailed explanation of the MERFISH gene selection, including how the final 238 target genes for MERFISH could be used to transcriptionally rederive the cell classes that were identified using the most variable genes from the scRNA-seq studies (Extended Data Fig. 8b). The expanded explanation has been included in the revised Methods as a new separate section entitled “MERFISH gene selection” (p. 63-65, lines 1230-1272), which is now followed by the “MERFISH probe library design and construction” section (p. 65-67, lines 1274-1311).

5 - The authors used computational approaches to identify distinct cell-types and what they call distinct ‘cellular communities’. These approaches appear powerful to identify specific subpopulations such as conduction system cardiomyocytes vs ventricular cardiomyocytes. However, it is unclear whether the subcommunities in the ventricular wall represent meaning biologically distinct regions with borders, or are more of an artificial subdivision of cells that form a continuous transmural transition of gene expression profiles.

We thank Reviewer 1 for their positive comments as well as their inquiries about the cellular community detection algorithm utilized in the manuscript. To understand how cardiac cells may coordinate to form and regulate the heart, we developed a community detection algorithm in order to identify cell populations which may interact with each other within local cellular neighborhoods based on their spatial proximity. To this end, the community detection algorithm initially defines the neighboring cell populations for each given individual cell of the heart within a 150 μm radius (i.e., cell zone), (Fig. 2a). These cell zones/neighborhoods are then clustered into communities based on the similarity of the content of the neighboring cell populations within each cell zone using a statistically guided clustering algorithm for community detection (Extended Data Fig. 16a). Thus, these communities (and subcommunities) are statistically-defined sets of spatially neighboring (and aggregating) cell populations which likely represent biologically distinct regions of interacting cell populations based on their spatial proximity. As a result, many of the communities identified in the whole heart community detection analysis correlate to known biologically significant cardiac structures as described (Fig. 2, p. 20-22, lines 435-477), and likewise many of the subcommunities in the ventricular wall also represent known biological regions, thus providing evidence

that the algorithm is able to define biologically meaningful regions/communities of the heart including potentially newly identified communities. However, because the community detection algorithm assigns each cell to only one community, this analysis may lead to creating communities with distinct borders rather than borders which may transition between communities. Thus, the combination of viewing both defined communities and the spatial distribution of individual cell populations, such as cell populations of the ventricular wall, provides a more holistic representation of how cardiac cell populations may interact and organize to create the functional structures of the heart. Thus, we have provided this explanation and limitation of the community detection algorithm analysis in the manuscript (p. 35, lines 774-782).

6 - The mouse study provides functional evidence for the biological relevance of Sema3c – plexin signaling during ventricular wall morphogenesis. However, this model should be studied in more detail. Can the authors provide a comprehensive description of the phenotypic analysis of the mutant mice? Mutant hearts appear to show developmental delay, this could be an alternative cause of differential thickness of compact and trabecular layers. At what stage is the difference in wall thickness first detectable?

We appreciate Reviewer 1's concerns and suggestion to provide a more comprehensive phenotypic analysis of the mutant mice in order to determine when differences in wall thickness is first detectable and whether they may be due to developmental delays. Toward this end, we included additional phenotypic analyses of WT and *Tcf21-CreERT2; Sema3c fl/fl* knockout mouse embryos at the following stages: E12.5, E14.5, and P1 (complementing our original E17.5 studies). Differences in ventricular wall thickness were first detected at E14.5, when the trabecular layer in the LV was significantly larger in *Tcf21-CreERT2; Sema3c fl/fl* hearts than WT hearts (Extended Data Fig. 28). At E17.5, *Tcf21-CreERT2; Sema3c fl/fl* hearts exhibited hypertrabeculated and thinner compact myocardial ventricular walls compared to hearts from wild-type mice (Fig. 5e, f). These differences in ventricular wall thickness continued to P1, supporting that these differences in the thickness of the trabecular and compact layers are due to a developmental defect rather than a developmental delay (Extended Data Fig. 28). Thus, we have included the findings of these additional mouse experiments (p. 33, lines 728-734; Extended Data Fig. 28) as well as expanded our description of how we performed our phenotypic analyses of ventricular wall thickness ('Animal studies' section of the Methods, p. 62-63, lines 1210-1220).

Referee #2 (Remarks to the Author):

Cardiac cell diversity plays an important role in the development of the heart. Molecular and cellular interaction between diverse cardiac cell types is essential for the creation of complex morphological structures critical for heart development and functions. The present study by

Farah et al. aims to characterize the identity of cardiac cell types in the developing human heart. Authors performed single-cell RNA sequencing (scRNA-seq) on human hearts between 9 to 16 post-conception weeks (PCW) in replicate. Authors have also tried to integrate the single cell RNA-sequencing with high-resolution multiplexed error-robust fluorescent in situ hybridization (MERFISH) to identify the cellular communities contributing to the formation of distinct cardiac structures. This is an interesting manuscript with a nice characterization of these different cell populations in the heart. However, multiple studies have performed single-cell RNA-sequencing to transcriptionally define distinct cell populations of the human heart (PMID: 32971526, PMID: 32403949, PMID: 33990324, PMID: 30759401, PMID: 31835037, PMID: 33184181). Considering these published works, the new information generated in the current manuscript is important but not very relevant. Most of these cell populations are defined at the transcription level and that is not very reliable considering scRNA-seq studies require tissue and cell dissociations causing loss of cell identity. More functional and molecular characterization is needed to suggest that they are rare (new) cell populations of the hearts.

This is an interesting manuscript, but several issues need to be addressed. Some of the major issues are described below.

We thank the Reviewer for their interest and constructive assessment of our findings, and helpful comments that improve the manuscript.

1. What if the diversity seen in the cardiac population is due to the different stages of human heart development as stages vary between 9 to 16 PCW and expression significantly changes during this early phase of cardiac development?

We appreciate Reviewer 2's inquiry with regards to the diversity of cell populations that we observed and whether some of this diversity may be due to age-related changes in gene expression. Indeed, Reviewer 2 is correct that some of the diversity of identified cell populations are due to changes in gene expression during aging (please see Reviewer 1, major point 1 response), whereas in other cases, it is due to specific developmental cell types (i.e. dorsal mesenchymal protrusion/DMP cells present primarily at 9 weeks, Extended Data Fig. 4), which may differentiate later into other specific cardiovascular lineages. Thus, because some cardiac cell populations represent developmental states of cell types, we decided to globally use the term 'cell population' rather than 'cell type' to define identified cell clusters from our analyses. Furthermore, in response to Reviewer 1, major point 1 and germane to Reviewer 2's inquiry, we have investigated how the gene expression of related cell populations may change with age utilizing WGCNA, and subsequently identified gene modules that correlate with age for each related cell population (Extended Data Figs. 9, 10, please see Reviewer 1, major point 1 and Reviewer 3, major point 3 responses). Overall, these findings support that

some of the diversity of the cell populations identified may be due to developmental cell state changes of specific cardiac lineages. Consequently, we have clarified these points and included the age-related WGCNA analyses in the manuscript to help strengthen our findings (please see p. 11, lines 220-228; Extended Data Figs. 9, 10 and Supplementary Table 4).

2. What is the rationale for dissecting the chambers and IVS for scRNA-sequencing? At these stages the hearts are very small so why not use the whole heart single cell suspension and perform scRNA-sequencing in an unbiased manner?

We thank Reviewer 2 for their inquiry regarding the sample preparation for the scRNA-seq studies. The rationale for performing scRNA-seq studies on dissected cardiac chambers and interventricular septum (IVS) was to increase the likelihood for identifying more cell types/states (including rarer populations), especially in smaller underrepresented regions such as the atria, which constitutes a smaller proportion of the total number of cells in the heart (please see Reviewer 2, major point 4 response). Thus, we have revised our manuscript to include this explanation (p. 6, lines 101-106).

3. Authors claim that they have sequenced a relatively high number of cells compared to prior cardiac developmental studies^{10–12}, enabling the identification of more rare cardiomyocytes including *BMP2+* non-chamber cardiomyocytes²⁰ (ncCM). *BMP2+* cells have been characterized previously as AVC populations (PMID: 27840109). It is not clear, what new information has been generated here if they are related to previously described cardiac cell populations and their spatiotemporal location.

We thank Reviewer 2 for their comment about the significance of our *BMP2+* non-chamber cardiomyocyte (ncCM) findings, particularly with regards to what new information that they provide to the biomedical community/field. While a *BMP2+* ncCM population has been previously characterized in the atrioventricular canal (AVC) in developing mouse hearts (PMID: 27840109), they have not been as well characterized in human hearts. Thus, our studies provide evidence and molecular characterization of *BMP2+* ncCM populations of the AVC of human hearts, which to our knowledge has not been as well described in prior human cardiac developmental single cell studies (PMID: 30759401, PMID: 31835037, PMID: 33184181, PMID: 35732239, PMID: 32810435, PMID: 36563664). Furthermore, this *BMP2+* ncCM population of the AVC, marked and characterized by *BMP2+/RSPO3+/MSX2+*, is distinct from a *BMP2+* ncCM population that we discovered in the inflow tract (please see response to Reviewer 2, major point 4), highlighting that we have revealed that *BMP2+* ncCMs represent two distinct non-chamber cardiomyocyte populations. However, in addition to the *BMP2+/RSPO3+/MSX2+* AVC ncCM population, we have also discovered a *BMP2-* CM population near the AVC, which was molecularly defined as *BMP2-/CNN1+/CRABP2+* and mapped to the

atrioventricular valve leaflets (Fig. 1d, Extended Data Figs. 11e, 13a, vCM-LV/RV-AV). These valve leaflet vCMs have not been well characterized in mouse or human hearts, and thus illustrate the new information provided by our scRNA-seq and MERFISH studies. Accordingly, we have revised and included these additional points in the manuscript to help clarify the significance of our *BMP2+* ncCM population findings and highlight how our scRNA-seq studies may provide additional insights into cell types contributing to the atrioventricular canal region (p. 7-8, lines 141-154, and p. 15-16, lines 316-333).

4. Similarly, authors show that non-chambered cardiomyocytes primarily segregated into *RSPO3/MSX2+* atrioventricular canal/node (ncCM-AVC-like) and *SHOX2/TBX18+* inflow tract/pacemaker cardiomyocytes (ncCM-IFT-like), which differentially expressed *ISL1*, *PITX2*, and *TBX3*, known transcription factors involved in regulating pacemaker sinoatrial node versus inflow tract development. Thus, these findings provide new developmental insight into not only previously reported chamber-related cardiomyocytes^{7–11} but also more specialized cardiomyocytes critical for regulating electrical cardiac conduction.

Based on the previous issue raised by Reviewer 2 as well as discussions with the Editor, our understanding of this concern is that Reviewer 2 is similarly inquiring about the biological significance and value of our finding that *BMP2+* non-chamber cardiomyocyte (ncCM) segregated into *RSPO3/MSX2+* atrioventricular canal/node (ncCM-AVC-like) and *SHOX2/TBX18+* inflow tract/pacemaker cardiomyocytes (ncCM-IFT-like). Similar to the *BMP2+* ncCM population of the AVC, the cell populations composing the inflow tract (IFT) including sinoatrial node also remains to be fully defined in human hearts (please see response to Reviewer 2, major point 3). Thus, the significance of our findings is that they provide insight and molecular definition into cell populations residing in these non-chambered regions of the human heart, which include *BMP2+* ncCMs which can be divided more specifically into *BMP2/RSPO3/MSX2+* ncCM-AVC-like and *BMP2/SHOX2/TBX18+* ncCM-IFT-like cell populations (Fig. 1d, Extended Data Figs. 3d, e, 11e, 13a). Additionally, these findings also highlight the value of dissecting the chambers and IVS prior to performing scRNA-seq, which enabled us to identify these rarer ncCM populations of the AVC and IFT (Extended Data Fig. 3d, please see Reviewer 2, major point 2 response). Finally, the MERFISH studies further spatially mapped the ncCM-IFT-like cell populations in precise locations of the developing human heart, and particularly showed that *BMP2/SHOX2/TBX18/ISL1+* ncCMs are enriched in the inflow tract of the RA versus the LA (Fig. 1d, Extended Data Figs. 3d, e, 11e, 13a), supporting that these spatially mapped cells are likely pacemaker CMs comprising the sinoatrial node as previously suggested in mouse hearts (PMID: 26786210). Thus, we have revised our manuscript to include these clarifications (p. 7-8, lines 141-154, and p. 15, lines 316-325).

5. It is not clear why authors performed MERFISH studies on 12-13 PCW human hearts while the scRNA-sequencing was performed on 9 to 16 PCW human hearts.

We appreciate Reviewer 2's inquiry about the developmental stages examined for our MERFISH and scRNA-seq studies. A major goal for our studies is to understand how diverse cardiac cell types coordinate to create complex morphological structures critical for heart function with a particular interest in ventricular wall morphogenesis. Toward this end, we initially performed scRNA-seq to investigate and identify the specific cell lineages comprising the developing human heart from 9-16 PCW, a time frame when the ventricle undergoes rapid expansion and remodeling of its wall (PMID: 14612588). To examine how these cardiac cell types may spatially organize during ventricular morphogenesis, we particularly focused on the dynamic but relatively underappreciated process of ventricular wall compaction, which has been reported to occur at ~12 PCW in human hearts (PMID: 14612588). Thus, we performed MERFISH at ~12 PCW to spatially map and examine specific cell populations organizing during the consolidation of the inner trabecular layer with the outer compact layer (i.e., ventricular wall compaction). Additionally, in response to Reviewer 1, major point 1, we have expanded these MERFISH studies to ~15 PCW hearts to further analyze this dynamic process. Overall, we have revised the manuscript to include these explanations (p. 11-12, lines 235-239) and have also added MERFISH studies at ~15 PCW (p. 25-26, lines 557-570; Extended Data Fig. 19).

6. In Figure 1g, Gene imputation performance was validated spatially by comparing normalized gene expression profiles of marker genes measured by MERFISH with the corresponding imputed gene expression profiles. However, these gene expression profiles do not match the previously reported profiles. For example, *Tbx18* is strongly expressed in IVS and ventricular walls and left ventricles (conflicting results published in these two Nature papers, PMID: 19369973 and PMID: 18480752). This raises concerns regarding the reliability of the gene expression profiles measured by MERFISH. Also, only 238 MERFISH target genes were used. Also, the expression of missing genes in MERFISH was determined using transcriptomic data from corresponding scRNA-seq cells

We appreciate Reviewer 2's inquiry into the performance of the gene imputation algorithm performed in this study. Regarding *Tbx18* expression, the following papers, PMID: 19369973 and PMID: 18480752, examined the expression pattern of *Tbx18* in the developing mouse heart at various stages and observed that *Tbx18* is primarily expressed on the epicardial surface at early developmental stages (~E10), and in both the epicardial surface and myocardial wall at later developmental stages (~E16). Thus, the *TBX18* expression (both measured and imputed) that we observed in our studies is consistent with the aforementioned mouse heart studies, as we observed that *TBX18* is expressed on the epicardial surface, with some expression within the ventricular wall of the human heart (Figure 1g). However, to further validate this result, we examined *TBX18*

expression by using an alternative imaging strategy (single molecule FISH, smFISH) and observed a similar spatial gene expression pattern of *TBX18* using either MERFISH or smFISH imaging approaches (Extended Data Fig 11d). Complementing these findings, a recently published paper examining spatial gene expression in the developing human heart at an earlier stage than our studies revealed that *TBX18* is expressed mainly on the epicardial surface, similar to the early developmental expression of *Tbx18* in mouse hearts (PMID: 31835037). As for the concern about using 238 genes to perform the gene imputation, we performed a bootstrap analysis where we analyzed the integrated scRNA-seq and MERFISH datasets at several quantities of genes (25, 50, 75, 100, 150, and 200 genes) and then compared the MERFISH gene predictability score for each quantity (Extended Data Fig. 15e). We discovered that increasing the number of genes used for the imputation improved the MERFISH gene predictability scores up to 150 genes, after which the predictability scores plateaued (Extended Data Fig. 15e), thus supporting that the set of 238 genes used for our MERFISH studies is above the number of genes (150) that is needed for optimizing the accuracy of this method for imputing the spatial expression of genes not imaged in MERFISH. However, we agree with the Reviewer that imputed gene expression of missing genes in MERFISH using corresponding scRNA-seq cells may be a limitation of the method. Thus, we have clarified these points in the Discussion (p. 35-36, lines 786-792).

7. Semaphorin-Plexin signaling directs ventricular cardiomyocyte organization is superficially described. Figure 5e, more stages should have been analyzed to see the progress of LVNC. It is not clear why no repulsive effect of *Sema6A* and *6B* was observed on inner-LV cells in Figure 5b but they were able to block *Sema3c* effects in Figure 5c.

We thank Reviewer 2 for their suggestion to examine more developmental stages of the mouse model in order to track the progress of LVNC in the *Tcf21-CreERT2; Sema3c fl/fl* knockout mice. Thus, in addition to the original mice that we examined at E17.5, we have also phenotypically analyzed WT and *Tcf21-CreERT2; Sema3c fl/fl* knockout mouse embryos at the following stages: E12.5, E14.5, and P1. Differences in ventricular wall thickness were first detected at E14.5 when the trabecular layer in the LV was significantly greater in the *Tcf21-CreERT2; Sema3c fl/fl* hearts than in WT hearts. These differences further progressed at E17.5 and P1 where *Tcf21-CreERT2; Sema3c fl/fl* ventricles exhibited a LVNC phenotype with hypertrabeculation and thinner compact myocardium compared to the ventricular wall of wildtype mice (Fig. 5e, f, Extended Data Fig. 28). Thus, we have included the findings of these additional mouse experiments (p. 33, lines 728-734; Extended Data Fig. 28, please also see Reviewer 1, major point 6 response). With regards to the effects of *SEMA6A* and *6B* in Figure 5b, *SEMA6A* and *6B* were mixed in the intermediate-LV CC-like layer, but not directly in the inner-LV CC-like layer where *PLXNA2/A4+* trabecular-like hPSC-vCMs were bioprinted. Since *SEMA6A* and *6B* ligands do not come into direct contact with *PLXNA2/4+* trabecular-like hPSC-vCMs, these ligands are unable to block/repel these hPSC-vCMs in the inner LV CC-like

layer. On the other hand, in Figure 5c, PLXNA2/A4+ trabecular-like hPSC-vCMs migrate from the inner-LV CC-like layer to the intermediate-LV CC-like layer because of SEMA3C (in Tier 2). However, under conditions where SEMA6A and 6B are mixed in the intermediate-LV CC-like layer (Tier 1), these migrating PLXNA2/A4+ vCMs are blocked/repelled from moving further into the intermediate-LV CC-like layer Tier 1 when they come into contact with SEMA6A and/or 6B. Thus, we discuss these points in the manuscript (p. 36-37, lines 797-820).

8. Class 3 semaphorins are secreted and Class 6 semaphorins are membrane-bound. It is not clear how Sema6A and 6B were able to block Secreted Sema3c functions.

We apologize that our explanation may not have been initially clear for how SEMA6A and 6B may be able to block SEMA3C function. Based on a combination of our MERFISH and experimental results, we have created a model in which trabecular ventricular cardiomyocytes (vCMs) that express PLXNA2/A4 may be initially attracted to SEMA3C secreted from ventricular cardiac fibroblasts in the intermediate/compact layer; however, as these migrating trabecular vCMs reach the intermediate/compact layer, they encounter endothelial cells expressing membrane-bound SEMA6A and 6B, which compete with SEMA3C to block/repel the further migration of trabecular vCMs within the intermediate/compact layer (see Fig. 5g). This model is consistent with other studies that suggest that class 6 semaphorins may block the response of class 3 semaphorins in specific cellular contexts (PMID: 20484647, PMID: 15814794). For example, previous studies have shown that Sema6A and 6B can block a PlxnA2-mediated attractive response of hippocampal mossy fiber axons *in vivo* (PMID: 20484647). Similarly, other studies have observed that class 3 and class 6 Semaphorins can interact with the same Plexin receptors (PMID: 15814794, PMID: 18625214), supporting a potential cross-talk between these two classes of Semaphorins that may occur through shared Plexin receptors and their respective downstream signaling pathways. Overall, these previously reported findings help support our model that SEMA6A and 6B expressed on endothelial cells may block the response of PLXNA2/A4+ trabecular vCMs to SEMA3C when these vCMs contact endothelial cells in the intermediate/compact layer. However, while we have provided experimental evidence for this model in Figure 5, we appreciate that further studies may be needed to fully understand the interactions between these two semaphorin classes. Thus, we clarified these points and the model in the manuscript, particularly in the Discussion (p. 36-37, lines 797-820), and further discuss how our findings may open up new lines of investigations to understand how different classes of Plexins and Semaphorins expressed exclusively or combinatorially on distinct cell types may mediate the morphogenesis of the heart and more specifically the ventricular wall.

9. In other cell types such as neural crest, *Sema6A* and *6B* repel cells, while *Sema3C* attracts cells depending upon the differential expression of *Sema6A-B* receptor *Plexin-A2* and *Sema3C* receptors *Plexin-D1/Neuropilin1-2*. In the present study, It is not clear how the same set of receptors *PlexinA2/A4* are responding differently (attractive for *Sema3C* and repulsive for *Sema6A/B*). What are the underlying mechanisms for this differential response?

We appreciate Reviewer 2's inquiry regarding the underlying mechanisms for *Plexin A2/A4*'s differential response to Semaphorin 3C and Semaphorin 6A/6B. Because previous studies have showed that *Plexin* receptors present on the same cell can differentially respond to various Semaphorins (PMID: 20484647, PMID: 15814794, PMID: 18625214), we believe that *PLXNA2/A4* on vCMs may also differentially interact and respond to *SEMA3C* and *SEMA6A/B* in order to direct either attractive or repulsive responses, respectively, in a context-specific manner. For instance, innervation of the suprapyramidal region by mossy fibers in mice as well as target selection of motor axons in *Drosophila* are regulated by a general principle of differential responses to opposing guidance signals including *Plexin-Semaphorin* signaling (PMID: 20484647, PMID: 9604933). In particular, *PlxnA4* can function as a direct receptor for *Sema6A* and *6B* to regulate the repulsive responses in mossy fibers, and this *Sema6A/B* mediated repulsive response can compete against an attractive response mediated by *PlxnA2* (PMID: 20484647), thus suggesting that a similar competition mechanism may occur in trabecular vCMs which express a similar set of *Plexin* receptors that respond to a similar set of Semaphorins. While the exact mechanisms for how the same set of *Plexin* receptors (*PLXNA2/4*) respond differentially to various signaling guidance cues remain to be fully elucidated, our studies support that vCMs may respond to a combination of attractive and repulsive signaling cues through specific interactions between *PLXNA2/4* and *SEMA3C-SEMA6A/B*, which in turn can direct the migration of vCMs to precise locations in the ventricular wall. Thus, we included these explanations in the Discussion (p. 36-37, lines 797-820).

Referee #3 (Remarks to the Author):

The authors present a very exciting study on human developmental heart, which is complete with single cell analysis and spatial data (MERFISH). However, their analysis is superficial, and they are missing a great opportunity to characterise more in depth the niches they discovered.

Below my suggestions.

We thank the Reviewer for their excitement and constructive comments to strengthen the manuscript, which we address below.

Major comments:

1 - In lines 205 - 206, the authors claimed that they have identified "novel" cell types that are yet to be defined. This sounds contradictory. It is possible that the authors have found cell types that were not identifiable using their methodology rather than a new cell type. If the authors want to claim the discovery of new cell types they should provide evidence, especially if they "discovered" them by clustering. Otherwise, they should reword this.

We thank Reviewer 3 for raising the point about our claim of identifying “novel” cell types in lines 205-206. We recognize how this sentence could be misinterpreted. Thus, we have removed the sentence to avoid any confusion (previously lines 205-206, now p. 10, line 206 in the revised manuscript).

2 - The authors mention often the term "blood cells" to refer to cells from myeloid and lymphoid lineages. It would be correct to use immune cells, progenitors (both RBC or WBC) than group them into a single "blood" category.

We agree with Reviewer 3 that the myeloid and lymphoid cell lineages belong to a broader “immune cell” group (i.e., white blood cells/WBCs) which is a subset of the overall blood cell category/compartments that also includes red blood cell (RBC) and platelet groups (PMID: 29364285). Thus, we apologize if we may have used “blood cells” to specifically refer to cells from myeloid and lymphoid lineages. Our intention was to use the term “blood cells” to refer to all blood cell populations including immune/WBC, RBC and platelet groups. Thus, we have clarified this point in the manuscript (see p. 10, lines 197-201) and also have reorganized Extended Data Fig. 7 to more accurately reflect the nomenclature and hierarchy of the blood cell category/compartments so that it is consistent with the organization of the other major cell compartments (cardiomyocytes, mesenchymal, etc.).

3 - In lines 220 - 223 the authors mention the potential that their discoveries have for cardiac biology. One of them is cell-cell interactions. With the richness of data that they mention, it is rather disappointing to see that they did not perform a CCI analysis, nor inferred any cell-specific GNR analysis when there are so many tools available to do this. Their analysis is mostly based on clustering and marker identification, which is biased. The clustering approach does not provide any metric to assess their accuracy, like SCCAF (<https://www.nature.com/articles/s41592-020-0825-9>). Without a more detailed functional analyses of the cell types they have identified, their study does not warrant support for novelty.

We thank Reviewer 3 for the suggestion of how our data could be further analyzed with additional bioinformatic tools. We particularly agree that there is great opportunity to analyze cell-cell interactions (CCIs) among the cell types that we identified and spatially mapped by MERFISH. Thus, we have combined the CCI tool, CellChat (PMID: 33597522, PMID: 35926050, PMID: 36790929), with our spatially-mapped cell types by MERFISH in order to identify and refine CCIs that are spatially and biologically relevant (Fig. 4d-f, Extended Data Figs. 23, 24, Supplementary Tables 16, 17). In particular, we applied this approach to discover CCIs regulating ventricular wall morphogenesis (Figs. 4, 5; Extended Data Figs. 22-28). Specifically, we identified cell types that could potentially interact based on spatial proximity using our MERFISH studies and cellular community analysis, and then performed CCI analysis on these cell types utilizing CellChat (Methods, p. 81, lines 1630-1641). Thus, in response to the CCI concern, we have not only clarified this strategy in the Results and Methods (p. 28, lines 621-629 and p. 81, lines 1630-1641) but also further implemented it to all other cellular communities to expand our analyses (Supplementary Tables 16, 17). With regards to the cell-specific GRN analysis, we have now performed a gene regulatory network analysis utilizing WGCNA on each of the cell classes to investigate the gene programs related to their development in Extended Data Fig. 9, and show network plots highlighting potential transcriptional regulators of age-related gene programs in Extended Data Fig. 10 (p. 11, lines 220-228; Extended Data Figs. 9, 10 and Supplementary Table 4, please see Reviewer 1, major point 1 and Reviewer 2, major point 1 responses). Finally, we thank Reviewer 3 for the recommendation to use SCCAF to assess the accuracy of our clustering approach. We have applied this algorithm on our clustering solutions and observed accuracies that were comparable to those reported with other human heart datasets (Extended Data Fig. 2) (PMID: 32971526). This analysis has been included as Extended Data Fig. 2, and p. 6, line 114-117 and in the Methods p. 73, lines 1445-1448.

Minor comments:

1 – The authors mention that they generated a comprehensive human heart cell atlas; however, it would be more accurate to say they develop a “comprehensive developmental human heart cell atlas”.

We appreciate Reviewer 3’s comment and revised line 201 (p. 10, line 202 in the revised version) to specify that the dataset is a ‘comprehensive developmental human heart cell atlas.’

2 - It would provide solid proof if the authors would have compared the probes identified with NS-Forest2 with a newer method like SpaPros (<https://github.com/theislab/spapros>).

We thank Reviewer 3 for their suggestion of comparing our NS-Forest2 classifier with Spapros to identify MERFISH candidate probes. Using the following pipeline: https://spapros.readthedocs.io/en/latest/tutorials/spapros_tutorial_basic_selection.html, we applied Spapros on the same cell compartments (cardiomyocyte, neuronal, blood, mesenchymal, and endothelial) that were used for our NS-Forest2 classifier analyses. Because Spapros initially generated only 80 candidate gene probes, we re-ran Spapros again to generate a larger list of 170 genes so that a similar number of genes could be compared between Spapros and NS-Forest2 classifiers (159 genes). When we assess the ability/accuracy of discovered genes from each classifier to re-identify the cell subpopulations from our scRNA-seq datasets (i.e., accuracy metric, Response Figure 3), we found that both gene lists were able to identify the original cell subpopulations from our studies with similar accuracy (Response Figure 3). Thus, we have added the Spapros comparison in the manuscript (p. 12, lines 242-244, and p. 64, lines 1243-1257 and Supplementary Tables 7, 8).

Response Figure 3. Evaluation of gene probe sets identified from Spapros and NS-Forest2 classifiers. Bar graph reveals the accuracy score for the ability of each set of candidate genes to identify specific cell subpopulations from scRNA-seq datasets.

Reviewer Reports on the First Revision:

Referees' comments:

Referee #1 (Remarks to the Author):

The authors dealt with most of the issues that were raised and I have now further comments

Referee #2 (Remarks to the Author):

Multiple studies have performed single-cell RNA-sequencing to transcriptionally define distinct cell populations of the human heart (PMID: 32971526, PMID: 32403949, PMID: 33990324, PMID: 30759401, PMID: 31835037, PMID: 33184181). Considering these published works, the new information generated in the current manuscript is not very relevant. Most of these cell populations are defined at the transcription level and that is not very reliable considering scRNA-seq studies require tissue and cell dissociations causing loss of cell identity. More functional and molecular characterization is needed to support the claim that they are rare (new) cell populations of hearts.

The study remains descriptive or predictive, with a very limited amount of validated experimental data.

Some of the major concerns are listed below.

1. Majority of these cell populations are defined based on the expression of one or two genes which are not very specific in their expression pattern and when samples from 9 to 16 PCW have been used it is very difficult to determine whether they are novel cell population, or the diversity seen in the cardiac cell population is due to the different stages used in the analysis. For example, in the revised manuscript, in addition to 13 PCW ventricles, authors have also performed MERFISH imaging in 15 PCW hearts to spatially corroborate age-related scRNA-seq findings and to determine whether age may lead to changes in the spatial distribution of different cell subpopulations. The authors discovered that 15 PCW ventricles did not contain hybrid vCM subpopulations in either the left or right ventricles compared to 13 PCW ventricles. This hybrid vCM also seems to be missing at other developmental stages such as 9 and 11 PCW (Extended Data Figure 19e). Hybrid vCMs are defined as HEY2+/IRX3+. HEY2+ cells are defined as compact vCMs and IRX3+ trabecular vCMs. Looking at the gene expression data presented in Extended Data Figure 17a-b, one can see HEY2 expression in trabecular vCMs and IRX3 expression in compact vCMs. During chamber formation, compact layer vCMs migrate to trabecular layer vCMs. As hearts from different stages have been used in the analysis it is not clear whether Hybrid vCMs are novel cell populations with specific functions at P13 PCW or different stages of proliferating and migrating vCMs.

2. Authors say that the major goal of the present study is to understand how diverse cardiac cell types coordinate to create complex morphological structures critical for heart function. Considering that they

already have performed scRNA-sequencing on 9 to 16 PCW human hearts, it would have been great to see how these cell populations progress from 9 PCW to 16 PCE.

3. Study remains descriptive, and no functional experiment was performed on any population identified in this study. Most of the conclusions are solely based on gene expression data. Too reliant on one single set of experiments.

4. Semaphorin-Plexin signaling directs ventricular cardiomyocyte organization is superficially described. The authors have included more developmental stages to show the progression of LVNC. As per the presented data, it seems *Sema3C* is expressed only in cardiac fibroblasts. Authors should show what is the efficiency of *Sema3C* deletion in the heart. In Figure 5b-c, the authors should have included positive controls to show that *SEMA3D*, *SEMA6A*, and *SEMA6B* used in the experiments are functional.

5. It is not clear why no effect of *Sema6A* and *6B* was observed on trabecular-like hPSC-vCMs when added in the two different tiers (tier I, II) of the intermediate-LV CC-like layer (Figure 5b, 6A/6A and 6B/6B panels) but they were able to block *SEMA3C* effects in Figure 5c. The authors explain that “in Figure 5c, *PLXNA2/A4+* trabecular-like hPSC-vCMs migrate from the inner-LV CC-like layer to the intermediate-LV CC-like layer because of *SEMA3C* (in Tier 2). However, under conditions where *SEMA6A* and *6B* are mixed in the intermediate-LV CC-like layer (Tier 1), these migrating *PLXNA2/A4+* vCMs are blocked/repelled from moving further into the intermediate-LV CC-like layer Tier 1 when they come into contact with *SEMA6A* and/or *6B*.” As *SEMA6A/6B* are membrane-bound, they need to be in physical contact with *PLXNA2/A4+* vCMs. It is not clear why we do not see any blocked/trapped cells in Tier 1. There are barely a few cells in the No *SEMA/3C* control. Authors could have used Figure 5b *SEMA3C/3C* setup and added *SEMA6A* or *6B* together with *3C* in Tier2 to see migration only in Tie1 and not Tier2 compared to both in the *3C/3C* combination.

6. Authors suggest that *SEMA3C/D* is originating from the intermediate-LV CC and influence the spatial re-allocation of *PLXNA2/A4+* trabecular vCMs. Data presented in Extended Data Figure 25d-e does not support this conclusion as *SEMA3C* and *SEMA3D* expression can be seen throughout the heart both in compact and trabecular layers. Similarly, *PLXNA2/A4* expression is also observed throughout the heart. The expression intensity may be a bit higher in the trabecular layer compared to the compact layer. What is the optimum level of expression required for *SEMA* ligands to repel or attract *PLXNA2/A4* expressing vCMs? Why *SEMA3C* will not attract *PLXNA2/A4+* cells in the compact vCM?

7. Similarly, expressions of *Sema6A/6B* are throughout the heart. Why *Sema6A/6B* will not repel *PLXNA2/A4+* cells in the compact vCM as per the model presented?

8. Authors conclude that *SEMA3C* function as a key attractive guidance cue for driving the migration of *PLXNA2/PLXNA4+* trabecular vCMs into the intermediate and outer layers of the ventricle during ventricular compaction. Are there any in vivo data authors can provide to support this model? Most published work suggests that compact layer CMs extend into the trabecular layer (PMID: 29743679) and not the other way around.

Referee #3 (Remarks to the Author):

I am very grateful to the authors for making significant changes to the first version of the manuscript. I think this has improved the quality of their, already, excellent work. However; now that the analysis is in a proper state, there are some concerning issues with it.

Below you will find my comments:

1) Identifying cell types and cell-states: In lines 106 - 112 they authors describe how they use their scRNA-Seq data to cluster the cells and identify cellular compartments and cell states. However, given the existence of, as of July 17th 2023, three human heart cell atlas (one with spatial information), this reviewer doesn't understand why the authors have not leveraged the annotation of the heart cell states. Using label transfer with scANVI or scArches, the authors can easily link their cellular compartments to previously described cell types and states. This approach will allow them to confirm their results computationally, and to truly identify novel cell states. At worst, it will be just confirmation of their findings. If the authors disagree with the application of these annotations, they should state why.

2) Gene regulatory networks: In the first round of reviews, I suggested the authors should further characterise their cell states using cell-cell interactions (CCI) and gene regulatory networks (GRN). I am very grateful for the authors to have considered this important point! However, I am confused by their tool of choice WGCNA. This tool is a correlation network inference method that looks for gene co-expression networks. This is not a gene regulatory inference network. More appropriate tools to do this are pySCENIC (or their newly publish SCENIC+ counterpart in R) and cell oracle. The issue here is that you need to identify regions and their target genes using databases of curated transcription factors (TF). Personally, this reviewer does not support the use of WGCNA in any droplet-based single-cell studies because it tends to capture a lot of false-positive correlation patterns in sparse scRNA-Seq data. The ideal scenario will be, as recently published studies have done, to use joint scRNA-Seq and scATAC-Seq data to integrate them and infer GRNs from their cell states, however; this may not be accessible to all research labs. Nevertheless, both tools provide a curated database that the authors could use to properly infer GRNs from their data.

3) BMP2+ cardiomyocytes: In lines 146 - 152 mention the presence of an exciting population expressing BMP2, ISL1 and TBX3 genes. How is this population compared with the recently published populations of the spatially-resolved niches for pacemaker cells (Kanemaru K, 2023 <https://www.nature.com/articles/s41586-023-06311-1>)? A comparative analysis of your data with their publicly available dataset will be beneficial to confirm the novelty of these cells.

4) Novelty of cell types: In lines 202 - 206, the authors claim to have identified novel cell types involved in heart morphogenesis. This may be so, but without a proper comparative analyses with the publicly

available atlases, it is hard to assess this. The SCCAF-guided clustering ensures that their populations are supported by their data, but to claim that these states are novel, they need to compare to what has been published. The authors have the data and the computational tools to perform this analysis, which will make their claim for novelty much stronger.

5) Cellular trajectory analysis: It is quite surprising to this reviewer that the authors missed the opportunity to characterise potential cellular trajectories with the populations that they identified as proliferative. Given the importance of their work, I would suggest to assess the implementation of a trajectory analysis with tools such as cellrank, and provide an overview of potential CCI and GRN in each stage of transition.

Author Rebuttals to First Revision:

The reviewers and the editorial team continue to find the study of interest for the readership of Nature. While we are willing to wave mouse models regarding the SEMA data and claims of a novel population of cells, major concerns have been raised regarding the data analysis. The mapping, analysis, and comparison of the scRNA-seq data in the study is a major reason for our consideration, and these concerns bring the soundness of that data into question.

Having said this, should future experimental data and theoretical analysis allow you to address these concerns we would be happy to look at a revised manuscript (unless, of course, something similar has by then been accepted at Nature or appeared elsewhere).

We thank the Editors and Reviewers for their careful consideration of our manuscript and for their overall positive assessment. The comments and constructive suggestions have helped us to improve the manuscript. We have provided a response to all Reviewer comments in bold, as well as new data and revisions in the manuscript to address their concerns, as detailed below for each Reviewer.

Referees' comments:

Referee #1 (Remarks to the Author):

The authors dealt with most of the issues that were raised and I have now further comments

We again thank Reviewer 1 for their helpful suggestions, which have strengthened the findings of our manuscript.

Referee #2 (Remarks to the Author):

Multiple studies have performed single-cell RNA-sequencing to transcriptionally define distinct

cell populations of the human heart (PMID: 32971526, PMID: 32403949, PMID: 33990324, PMID: 30759401, PMID: 31835037, PMID: 33184181). Considering these published works, the new information generated in the current manuscript is not very relevant. Most of these cell populations are defined at the transcription level and that is not very reliable considering scRNA-seq studies require tissue and cell dissociations causing loss of cell identity. More functional and molecular characterization is needed to support the claim that they are rare (new) cell populations of hearts.

The study remains descriptive or predictive, with a very limited amount of validated experimental data.

We thank the Reviewer for their evaluation of our findings, and helpful comments that improve the manuscript.

Some of the major concerns are listed below.

1. Majority of these cell populations are defined based on the expression of one or two genes which are not very specific in their expression pattern and when samples from 9 to 16 PCW have been used it is very difficult to determine whether they are novel cell population, or the diversity seen in the cardiac cell population is due to the different stages used in the analysis. For example, in the revised manuscript, in addition to 13 PCW ventricles, authors have also performed MERFISH imaging in 15 PCW hearts to spatially corroborate age-related scRNA-seq findings and to determine whether age may lead to changes in the spatial distribution of different cell subpopulations. The authors discovered that 15 PCW ventricles did not contain hybrid vCM subpopulations in either the left or right ventricles compared to 13 PCW ventricles. This hybrid vCM also seems to be missing at other developmental stages such as 9 and 11 PCW (Extended Data Figure 19e). Hybrid vCMs are defined as HEY2+/IRX3+. HEY2+ cells are defined as compact vCMs and IRX3+ trabecular vCMs. Looking at the gene expression data presented in Extended Data Figure 17a-b, one can see HEY2 expression in trabecular vCMs and IRX3 expression in compact vCMs. During chamber formation, compact layer vCMs migrate to trabecular layer vCMs. As hearts from different stages have been used in the analysis it is not clear whether Hybrid vCMs are novel cell populations with specific functions at P13 PCW or different stages of proliferating and migrating vCMs.

We appreciate Reviewer 2's inquiry about the cell populations that we have identified and defined in our manuscript using both scRNA-seq and spatial transcriptomics. These cell populations were initially curated from our scRNA-seq based on not just the expression of one or two genes but several hundreds/thousands of differentially expressed genes among these cell populations (Supplementary Table 3); however, for ease of referencing,

we have used one or two specific genes as examples for each cell population in each cluster/subcluster analyses. As noted by Reviewer 2, some of the cell populations are cell types at specific states/stages of heart development as evidenced by distinct cell populations of certain cell types being present at specific stages of heart development (e.g., specific ventricular cardiomyocyte (vCM) cell populations that are only present at specific developmental stages, Extended Data Figs. 3-8). As a result, we have been careful to use particular cell nomenclature such as “cell populations”, “cell subpopulations”, etc. rather than globally using “cell types” to define cell clusters identified from our scRNA-seq analysis. To determine how novel these identified cell populations are, we have now compared our data with those from a recently published heart atlas (Kanemaru et al., *Nature* 2023, PMID: 37438528, which includes other published scRNA-seq heart datasets) using scArches as requested by Reviewer 3 (please see Reviewer 3 major point 1 and major point 4 responses). With regards to the hybrid vCM subpopulation, this vCM subpopulation is defined by not only the co-expression of the compact vCM marker *HEY2* and trabecular vCM maker *IRX3* but other gene markers specific to compact or trabecular vCMs, including *GJA5*, *CGNL1*, and *DHRS3* (Extended Data Fig. 20a, b). We agree that it is interesting that this vCM subpopulation is enriched at 13 PCW when ventricular wall compaction occurs but substantially reduced at 9, 11 and 15 PCW. Thus, these findings highlight the utility of our molecular and spatial examination of cell populations across different developmental stages as well as the transient nature of cell populations during heart development, particularly when vCMs transition from one state/stage to another during ventricular wall compaction. Consistent with these findings, recent mouse lineage tracing studies have also suggested the existence of a hybrid population of vCMs during ventricular wall compaction but the precise identification of these vCMs remains to be illuminated (PMID: 28729659, p. 27, lines 588-593). Thus, it will be interesting to investigate in the future whether our identified hybrid vCMs from human hearts are similar to those suggested in the mouse hearts. Finally, while we appreciate that compact vCMs migrate to become trabecular vCMs during the early cardiac developmental process of trabeculation when cardiac chambers form (PMID: 29743679), our studies are focused on the later cardiac developmental process of ventricular wall compaction after cardiac chambers have been created but when they are remodeling (see Reviewer 2, major point 8 response). Thus, we apologize if these points were not made clear. Consequently, we have clarified them in the manuscript (p. 37-38, lines 828-832) as well as modified our conclusions of our identified cell populations, which are now substantiated with our comparison between our cell populations and those recently reported in a human heart cell atlas which includes other published scRNA-seq human heart datasets (p. 10-11, lines 205-216, Extended Data Fig. 9).

2. Authors say that the major goal of the present study is to understand how diverse cardiac cell types coordinate to create complex morphological structures critical for heart function. Considering that they already have performed scRNA-sequencing on 9 to 16 PCW human hearts, it would have been great to see how these cell populations progress from 9 PCW to 16 PCE.

We appreciate Reviewer 2's suggestion to analyze the developmental progress of our identified cardiac cell populations from 9 PCW to 16 PCW. To this end, we have provided detailed analysis of our scRNA-seq data from 9-16 PCW, which revealed significant age-related gene expression changes within specific cardiac lineages (Extended Data Figs. 3-8). For example, ventricular cardiomyocytes (vCM) subdivided into populations correlating to developmental age (vCM-Early and vCM-Late) (Extended Data Fig. 3d, e), suggesting a developmental progression from 9 PCW to 16 PCW. To further understand how these cardiac lineages progress during development, we have analyzed the gene regulatory networks of these cardiac lineages that span across these developmental stages utilizing pySCENIC (please see response to Reviewer 3, major point 2). As a result, these analyses revealed regulons (i.e., coordinately regulated gene programs) that correlate with age for each cell class (Extended Data Figs. 11, 12). In particular, we discovered that regulons which upregulate over time are related to cell type differentiation and function (e.g., muscle contraction for cardiomyocytes, collagen fibril organization for fibroblasts), whereas regulons that downregulate over time are related to general cell processes such as splicing, translation and cell cycle. Finally, we have also included a cellular trajectory analysis on vCMs that enabled pseudotime ordering of the gene regulatory networks and cell-cell interactions during the developmental progress of vCMs from 9-16 PCW (please see Reviewer 3 major point 5 response). Thus, we have included these additional pySCENIC and trajectory analyses to further support our findings and show how identified cell populations progress from 9-16 PCW (p. 11-12, lines 237-253; Extended Data Figs. 11-13 and Supplementary Tables 4-6).

3. Study remains descriptive, and no functional experiment was performed on any population identified in this study. Most of the conclusions are solely based on gene expression data. Too reliant on one single set of experiments.

We appreciate Reviewer 2's interest for functional studies that further characterize cell populations from our scRNA-seq studies. To this end, we have provided a high-resolution MERFISH spatial mapping of cell populations from our scRNA-seq in order to confirm and further spatially characterize and define cell population identities. As such, these findings have illuminated specific cell populations located in distinct regions of the developing heart, which may participate in cardiac developmental events that remain to be fully elucidated. Because these events include the process of ventricular wall compaction, which may be clinically relevant and significant to left ventricular non-compaction cardiomyopathies, we further functionally examined cell populations involved in this developmental process using a combination of conditional/inducible (Cre-mediated) mouse genetic studies and *in vitro* hPSC studies, which is in line with Reviewer 2's suggestion to not rely on one set of experiments (please see Fig. 5, Extended Data Figs. 29-31). Thus, we have clarified that a combination of functional

experimental studies was performed to help validate the role of identified cell populations in ventricular wall compaction, and have also included that these studies offer new opportunities to further investigate in the future this clinically-relevant developmental process (p. 38, lines 845-853).

4. Semaphorin-Plexin signaling directs ventricular cardiomyocyte organization is superficially described. The authors have included more developmental stages to show the progression of LVNC. As per the presented data, it seems Sema3C is expressed only in cardiac fibroblasts. Authors should show what is the efficiency of Sema3C deletion in the heart. In Figure 5b-c, the authors should have included positive controls to show that SEMA3D, SEMA6A, and SEMA6B used in the experiments are functional.

We thank the Reviewer for recommending controls for the mouse Sema3C deletion and hPSC Semaphorin studies. To examine the efficiency of the mouse Sema3C deletion within the heart, we performed qPCR for Sema3C on E18.5 ventricles and discovered that Sema3C mRNA in the mutant hearts was reduced to 30% of the amount observed in wildtype hearts, akin to what was previously reported (PMID: 26053665) (Response Fig. 1a). To measure and validate the activities of the commercial semaphorin proteins used for our studies (SEMA3C, SEMA3D, SEMA6A, and SEMA6B), we performed functional ELISAs and found that they all bind to the NRP1 receptor with similar binding affinities (Response Fig. 1b).

Response Figure 1. Control experiments evaluating semaphorin-related reagents. **a**, Gene expression of Sema3C was measured in E18.5 wildtype (WT) and Sema3C knockout (KO) mouse hearts using qPCR. Error bars are SEM. *** $p < 0.01$ by one-way ANOVA. **b**, The activity of each semaphorin protein was measured by its binding ability to NRP1 in a functional ELISA. Error bars are SEM.

5. It is not clear why no effect of Sema6A and 6B was observed on trabecular-like hPSC-vCMs when added in the two different tiers (tier I, II) of the intermediate-LV CC-like layer (Figure 5b, 6A/6A and 6B/6B panels) but they were able to block SEMA3C effects in Figure 5c. The authors explain that “in Figure 5c, PLXNA2/A4+ trabecular-like hPSC-vCMs migrate from the inner-LV CC-like layer to the intermediate-LV CC-like layer because of SEMA3C (in Tier 2). However, under conditions where SEMA6A and 6B are mixed in the intermediate-LV CC-like layer (Tier 1), these migrating PLXNA2/A4+ vCMs are blocked/repelled from moving further into the intermediate-LV CC-like layer Tier 1 when they come into contact with SEMA6A and/or 6B.” As SEMA6A/6B are membrane-bound, they need to be in physical contact with PLXNA2/A4+ vCMs. It is not clear why we do not see any blocked/trapped cells in Tier 1. There are barely a few cells in the No SEMA/3C control. Authors could have used Figure 5b SEMA3C/3C setup and added SEMA6A or 6B together with 3C in Tier2 to see migration only in Tie1 and not Tier2 compared to both in the 3C/3C combination.

We appreciate Reviewer 2’s inquiries about the role of SEMA6A and 6B in our experimental studies and overall model. For Figure 5b, there are no appreciable effects of SEMA6A or 6B on inner-LV CC-like layer trabecular-like hPSC-vCMs when these Semaphorins are added to the intermediate-LV CC-like Tier 1 or 2 layer because neither Semaphorin is able to attract these inner-LV CC-like layer hPSC-vCMs into the intermediate-LV CC-like layer. Consequently, these trabecular-like hPSC-vCMs remain in the inner-LV CC-like layer under these conditions. In Figure 5c, we designed hPSC-vCM experiments that specifically allowed us to test whether SEMA6A or SEMA6B can block migrating trabecular-like hPSC-vCMs between the inner- and intermediate-LV CC-like layers. This required us to add SEMA3C (to induce migration) and SEMA6A and 6B (to block migration) in different combinations between the layers and Tiers as outlined in Figure 5c and Methods. We showed in the No SEMA/SEMA3C Tier 1 and 2 condition that there was an appreciable amount of trabecular-like hPSC-vCMs that was able to be induced to migrate into the intermediate-LV CC-like layer as shown and quantitated in Figure 5c, d. This allowed us an opportunity to test whether adding SEMA6A and 6B to Tier 1 (rather than No SEMA) could block this migration. As a result, we observed that these trabecular-like hPSC-vCMs did not migrate into the intermediate-LV CC-like layer under the SEMA6A/SEMA3C, SEMA6B/SEMA3C or SEMA6A-6B/SEMA3C Tier 1 and 2 conditions likely because SEMA6A and SEMA6B blocked the ability of these vCMs to advance into intermediate-LV CC-like layer when they came in contact with SEMA6A/6B at the border of the inner- and intermediate-LV CC-like layers (Fig. 5c, d). These findings are consistent with previous studies showing that secreted ligands typically have higher binding affinities than membrane-bound ligands, and allow for long range interactions, whereas ligands bound to a membrane can cluster together and strengthen individually weak protein-protein interactions (PMID: 25321392, PMID: 28340336, PMID: 24006364), thus potentially further explaining why PLXNA2/A4+ trabecular-like vCMs are unable to migrate into the intermediate-LV CC-like Tier 1 layer, which contains a high concentration of SEMA6A/6B. Nonetheless, we appreciate that future studies, particularly those testing gradients of these Semaphorins (and Plexins) as alluded by the Reviewer, will be helpful

for further illuminating mechanistically how these Semaphorins and Plexins may direct the organization of distinct cardiac cell populations during ventricular wall morphogenesis. Thus, we have clarified these points, and now discuss how our findings may lead to new lines of investigations into the mechanisms of the Semaphorin-Plexin interactions in the manuscript (p. 35, lines 774-779 and p. 38, lines 832-853). Furthermore, we replaced Figure 5c No SEMA/SEMA3C panel with a better representative image that clearly shows that there are migrating trabecular-like hPSC-vCMs in the Intermediate-LV CC-like layer, consistent with Figure 5d quantitative data.

6. Authors suggest that SEMA3C/D is originating from the intermediate-LV CC and influence the spatial re-allocation of PLXNA2/A4+ trabecular vCMs. Data presented in Extended Data Figure 25d-e does not support this conclusion as SEMA3C and SEMA3D expression can be seen throughout the heart both in compact and trabecular layers. Similarly, PLXNA2/A4 expression is also observed throughout the heart. The expression intensity may be a bit higher in the trabecular layer compared to the compact layer. What is the optimum level of expression required for SEMA ligands to repel or attract PLXNA2/A4 expressing vCMs? Why SEMA3C will not attract PLXNA2/A4+ cells in the compact vCM?

We thank Reviewer 2 for their comments and questions about our semaphorin-plexin signaling model and the expression of *SEMA3C/D* and *PLXNA2/A4* across the ventricular wall. We agree that these Semaphorins and Plexins are expressed in a complementary gradient across the ventricular wall when measured by smFISH and quantified along the wall depth (Extended Data Fig. 28d, e – previously Extended Data Fig. 25d, e). In particular, *PLXNA2/A4* expression is higher in the Inner-LV CC trabecular layer but lower in Outer/Intermediate-LV CCs but *SEMA3C/D* is conversely lower in the Inner-LV CC but higher in Outer/Intermediate-LV CCs. These expression gradients are consistent with the spatial gradients of cells expressing the respective ligand/receptor (e.g., *SEMA3C/D*+ compact vFibro and *PLXNA2/4*+ trabecular vCMs - Fig. 4g, h, Extended Data Fig. 28a, b). This gradient of expression of SEMA ligands is in line with other models of secreted semaphorins displaying similar gradients of expression for directing cortical neuron migration and axon guidance (PMID: 18059265, PMID: 22368082). In regards to the optimal level of expression of SEMA ligands, previous studies have shown that these ligands act in a cell context-specific manner (PMID: 18625214, PMID: 22368082). Thus, discovering the optimal level would need to be derived for each specific cellular context; however, these studies, while interesting, may be beyond the scope of the current study. With regards to why SEMA3C would not attract PLXNA2/A4+ cells in the Outer-LV CC compact layer, our model predicts that PLXNA2/A4+ cells would be attracted to SEMA3C in the Outer-LV CC, but are blocked/prevented from further migrating in the Outer-LV CC due to the presence of SEMA6A/B expressing endothelial cells (see Reviewer 2, major point 7). Thus, we have clarified these points and model, and further included in the Discussion how our findings may lead to new lines of investigations into the underlying

mechanisms of how Semaphorin-Plexin interactions may regulate ventricular wall morphogenesis (p. 38, lines 832-853).

7. Similarly, expressions of *Sema6A/6B* are throughout the heart. Why *Sema6A/6B* will not repel *PLXNA2/A4+* cells in the compact vCM as per the model presented?

We thank Reviewer 2 for their inquiry about *SEMA6A/6B*, which is related to their previous question about *SEMA3C/D* and *PLXNA2/A4+* trabecular vCMs (Reviewer 2, major point 6). Similar to the expression of *SEMA3C/D* and *PLXNA2/A4*, we also observed that *SEMA6A* and *6B* are also expressed as a gradient along the ventricular wall when measured and quantified along the wall depth by smFISH (Extended Data Fig. 28d, e – previously Extended Data Fig. 25d, e). Specifically, the expression of *SEMA6A* and *6B* is highest in the Outer/Intermediate-LV CCs but lower within the Inner-LV CC. This expression gradient correlates with the spatial gradient of blood endothelial cells that express these ligands (Fig. 4g, Extended Data Fig. 28a, b), and are in line with other systems that display similar gradients of semaphorins for regulating the migration of neuronal cells (PMID: 18059265, PMID: 22368082, please see response to Reviewer 2, major point 6). Overall, these combined *SEMA-PLXN* findings are consistent with a model where *PLXNA2/A4+* trabecular vCMs are attracted by *SEMA3C* to the Outer/Intermediate-LV CCs but are prevented from further migrating into the Outer-LV CC when they come into direct contact with *SEMA6A/6B* expressing endothelial cells present within the Outer-LV CC (Fig. 5g). As such, these interactions result in a gradient of *PLXNA2/A4+* trabecular vCMs that progressively decrease along the wall depth with few cells present within the Outer-LV CC (Fig. 4g, Extended Data Fig. 28a, b). Thus, we have clarified these points and model in the Discussion (p. 38, lines 832-853).

8. Authors conclude that *SEMA3C* function as a key attractive guidance cue for driving the migration of *PLXNA2/PLXNA4+* trabecular vCMs into the intermediate and outer layers of the ventricle during ventricular compaction. Are there any *in vivo* data authors can provide to support this model? Most published work suggests that compact layer CMs extend into the trabecular layer (PMID: 29743679) and not the other way around.

We appreciate Reviewer 2's comments about providing *in vivo* data to support the role of *SEMA3C* as a key attractive guidance cue for driving the migration of *PLXNA2/PLXNA4+* trabecular vCMs. We also agree with Reviewer 2 that most published studies on ventricular wall development have investigated the early cardiac developmental process of trabeculation, which is when compact layer cardiomyocytes extend into the trabecular layer during early heart development (PMID: 29743679). However, to provide examples of how our data may offer new insights into biological processes that remain to be further elucidated, we have used our data to shed new light into the less well-studied process of

ventricular wall compaction which occurs later in heart development when the trabecular layer reduces but the compact layer expands (PMID: 14612588, p. 37-38, lines 828-850). As such, our data reveals that SEMA3C, which is expressed by cardiac fibroblasts residing in the Outer- and Intermediate-LV CCs, may attract Inner-LV CC PLXNA2/PLXNA4+ trabecular vCMs to this Outer-LV CC compact layer. To provide *in vivo* data to support this model, we have investigated a mouse genetic model that shows that genetically deleting Sema3C in cardiac fibroblasts leads to hypertrabeculation and a thinner compact ventricular wall compared to hearts from wildtype mice (Fig. 5, Extended Data Fig. 31). This Sema3C knockout mouse phenotype is consistent with the inability of trabecular vCMs to migrate to the Outer/Intermediate-LV CC in the absence of Sema3C, thus supporting the role of SEMA3C attracting Inner-LV CC PLXNA2/PLXNA4+ trabecular vCMs to the Outer-LV CC. Thus, we have clarified these points in the manuscript and included in the Discussion that additional *in vivo* studies in the future will be interesting to further elucidate underlying cellular mechanism for how Semaphorins and Plexins may regulate ventricular wall morphogenesis (p. 34, lines 750-763, p. 38, lines 850-853).

Referee #3 (Remarks to the Author):

I am very grateful to the authors for making significant changes to the first version of the manuscript. I think this has improved the quality of their, already, excellent work. However; now that the analysis is in a proper state, there are some concerning issues with it.

We very much appreciate Reviewer 3's thoughtful and positive assessment of our manuscript and their additional comments/suggestions, which has greatly strengthened our findings and conclusions.

Below you will find my comments:

1) Identifying cell types and cell-states: In lines 106 - 112 they authors describe how they use their scRNA-Seq data to cluster the cells and identify cellular compartments and cell states. However, given the existence of, as of July 17th 2023, three human heart cell atlas (one with spatial information), this reviewer doesn't understand why the authors have not leveraged the annotation of the heart cell states. Using label transfer with scANVI or scArches, the authors can easily link their cellular compartments to previously described cell types and states. This approach will allow them to confirm their results computationally, and to truly identify novel cell states. At worst, it will be just confirmation of their findings. If the authors disagree with the application of these annotations, they should state why.

We thank Reviewer 3 for their recommendation to link our cell compartments/populations to cell types and states previously described. To this end, we computationally related our cell population data with those recently published (Kanemaru et al., *Nature* 2023, PMID: 37438528) by applying the label transfer approach scArches. We discovered that several of our identified cell populations match those previously described, including vascular- and neuronal-related cell populations (Extended Data Fig. 9). Additionally, we further discovered that developing and adult hearts also consist of some cell populations that are specific for each corresponding heart condition (adult versus developing). For instance, adult hearts contain specific lymphoid and adipocyte lineages, whereas developing hearts contain several distinct cell subpopulations including those from cardiomyocytes and fibroblasts, which partially map to adult heart cell populations (Extended Data Fig. 9). Thus, we have included these label transfer analyses and confirmation of our findings in the manuscript (p. 10-11, lines 205-216, Extended Data Fig. 9).

2) Gene regulatory networks: In the first round of reviews, I suggested the authors should further characterise their cell states using cell-cell interactions (CCI) and gene regulatory networks (GRN). I am very grateful for the authors to have considered this important point! However, I am confused by their tool of choice WGCNA. This tool is a correlation network inference method that looks for gene co-expression networks. This is not a gene regulatory inference network. More appropriate tools to do this are pySCENIC (or their newly publish SCENIC+ counterpart in R) and cell oracle. The issue here is that you need to identify regions and their target genes using databases of curated transcription factors (TF). Personally, this reviewer does not support the use of WGCNA in any droplet-based single-cell studies because it tends to capture a lot of false-positive correlation patterns in sparse scRNA-Seq data. The ideal scenario will be, as recently published studies have done, to use joint scRNA-Seq and scATAC-Seq data to integrate them and infer GRNs from their cell states, however; this may not be accessible to all research labs. Nevertheless, both tools provide a curated database that the authors could use to properly infer GRNs from their data.

We greatly appreciate the feedback from Reviewer 3 about the utility of WGCNA for inferring GRNs and the recommendation to use pySCENIC or SCENIC+ instead. Given that our paper provides single cell transcriptomic data, we utilized pySCENIC to infer regulons for each cell class to examine how their gene expression changes over time. As a result, these analyses revealed regulons that correlated with age for each cell class (Extended Data Figs. 11, 12, please see Reviewer 2, major point 2 response). Thus, we have replaced our previous WGCNA analyses with these new age-related pySCENIC analyses to further support our findings (p. 11-12, lines 237-246; Extended Data Figs. 11, 12 and Supplementary Table 5).

3) BMP2+ cardiomyocytes: In lines 146 - 152 mention the presence of an exciting population expressing BMP2, ISL1 and TBX3 genes. How is this population compared with the recently published populations of the spatially-resolved niches for pacemaker cells (Kanemaru K, 2023 <https://www.nature.com/articles/s41586-023-06311-1>)? A comparative analysis of your data with their publicly available dataset will be beneficial to confirm the novelty of these cells.

We thank Reviewer 3 for their recommendation to compare our *BMP2+* cardiomyocyte data with the recently published cell populations of the spatially-resolved niches for pacemaker cells (Kanemaru et al., *Nature* 2023, PMID: 37438528). To this end, we linked our cardiac cell population data with those from Kanemaru et al. using a cell label transfer strategy as suggested by Reviewer 3, major point 1 (e.g. scArches, please see Reviewer 3, major point 1 response). We discovered that developing *BMP2+* non-chamber cardiomyocytes (ncCMs), which included ncCM-IFT-like and ncCM-AVC-like, exhibited higher expression of developmental transcription factors (e.g., IFT-like – *SHOX2*, *TBX18*; AVC-like – *TBX3*, *MSX2*) and lower expression of ion channels (e.g., IFT-like/AVC-like – *CACNA1D*, *HCN1*) and sarcomeric proteins (IFT-like – *MYH11*, AVC-like – *MYH9*) compared to adult pacemaker cardiomyocytes, which also no longer express *BMP2* (Extended Data Fig. 9c). These results are consistent with previous findings that as cells of the cardiac conduction system develop and mature, the expression of cardiac pacemaker transcription factors decreases and becomes more restricted (PMID: 30042181). Additionally, in line with the developing and maturing of pacemaker cells, ion channels that are involved in pacemaker activity, also appear to increase from fetal to adult human hearts (Extended Data Fig 9c) (PMID: 25623957). Thus, we have included these additional comparative analyses in the manuscript to further enhance our findings (p. 10-11, lines 205-216, Extended Data Fig. 9).

4) Novelty of cell types: In lines 202 – 206, the authors claim to have identified novel cell types involved in heart morphogenesis. This may be so, but without a proper comparative analyses with the publicly available atlases, it is hard to assess this. The SCCAF-guided clustering ensures that their populations are supported by their data, but to claim that these states are novel, they need to compare to what has been published. The authors have the data and the computational tools to perform this analysis, which will make their claim for novelty much stronger.

We are grateful for Reviewer 3's recommendations to perform a comparative analysis of our data with those from publicly available cell atlases in order to assess the novelty of the cell populations identified from our studies. Using the scArches cell label transfer algorithm as suggested by Reviewer 3 (please see Reviewer 3 major point 1 response),

we have compared our identified cell populations to those reported in publicly available datasets. In particular, we utilized the treeArches framework within scArches (PMID: 37502708) to allow for an unbiased identification of potentially new cell populations that are not present in the reference cardiac dataset. As a result, we discovered cell populations that appear to be specific for the developing heart (p. 10-11, lines 209-216; Extended Data Fig. 9), and include those which represent developmental cell states for certain cell types (i.e., multiple potential cell states of developing ventricular cardiomyocytes – Extended Data Fig. 3, 9) as well as cell types specific for the developing heart (i.e., M20 cell – dorsal mesenchymal protrusion/DMP-like cell which mainly appears at 9 PCW - Extended Data Fig. 4, 9). Thus, these analyses have allowed us to be more circumspect about our interpretation of the novelty of our identified cell populations as suggested by the Reviewers. Thus, we have added the comparative analyses to the manuscript and modified our conclusions of the cell populations identified from our analyses in human fetal hearts (p. 10-11, lines 205-216, Extended Data Fig. 9).

5) Cellular trajectory analysis: It is quite surprising to this reviewer that the authors missed the opportunity to characterise potential cellular trajectories with the populations that they identified as proliferative. Given the importance of their work, I would suggest to assess the implementation of a trajectory analysis with tools such as cellrank, and provide an overview of potential CCI and GRN in each stage of transition.

We appreciate Reviewer 3's suggestion to characterize potential cellular trajectories with our cell populations that we identified as proliferative in our original submission. Thus, we performed a cellular trajectory analysis on ventricular cardiomyocyte (vCM) populations from 9-16 PCW because they exhibit multiple proliferative and developmental cell populations, and are also the primary biological focus of our manuscript. Utilizing the Waddington-OT trajectory analysis tool, which employs optimal transport analysis, as suggested by Reviewer 3, we characterized the trajectory of vCMs as they progressed from 9 PCW to 16 PCW, and discovered that the proliferative populations, which correlate with earlier developmental time points (Extended Data Fig. 3), also associate with earlier pseudotime stages (Extended Data Fig. 13a-c). To provide an overview of potential cell-cell interactions (CCIs) and gene regulatory networks (GRNs) of vCMs transitioning during development, we integrated the cellular trajectories of developing vCMs with corresponding vCM GRNs that we inferred using pySCENIC (as kindly suggested by Reviewer 3 in major point 2), as well as CCI results that we generated. Specifically, the vCM pseudotime was projected from our cellular trajectories onto our inferred vCM GRNs and CCIs by determining the expression-weighted pseudotime of each respective transcription factor and receptor/ligand expressed by vCMs for corresponding GRN and CCI analyses as previously described (Fleck et al., *Nature*, 2022, PMID: 36198796). This analysis allowed us to (pseudotime) order the

transcriptional regulators and receiving interactions of vCMs to better understand the factors important for their development (Extended Data Fig. 13d, e). We found that SEMA-PLXN interactions mapped to later vCM transition stages which correspond to the later developmental stage upon which we performed MERFISH on (13 PCW) (Extended Data Fig. 13e), supporting our SEMA-PLXN related findings in the manuscript. Additionally, comparing this Waddington-OT-based pseudotemporal ordering of vCM CCIs to those by PAGA trajectory analysis support an improvement in the vCM trajectory analysis using Waddington-OT (Response Figure 2). Thus, we kindly thank Reviewer 3 again for their recommendation to use an optimal transport-based trajectory tool instead of PAGA. Overall, Reviewer 3's thoughtful suggestions have been particularly helpful for improving our analyses and providing a comprehensive overview of how ventricular cardiomyocytes may develop during human heart development. These findings have been now included to improve our manuscript and its conclusions (p. 11-12, lines 221-253; Extended Data Figs. 10-13).

Response Figure 2. Pseudotime ordering of vCM CCIs using PAGA versus Waddington-OT trajectory analyses. Dot plots of interactions received by vCMs were pseudotime ordered based on PAGA or Waddington-OT trajectory analyses

Reviewer Reports on the Second Revision:

Referees' comments:

Referee #2 (Remarks to the Author):

As previously mentioned, multiple recent studies have been performed on single-cell RNA-sequencing to transcriptionally define distinct cell populations of the human heart. In the context of these published works, there is no novelty or robustness of the data presented in the current manuscript. The new cardiac cell populations identified in the revised manuscript should be at least characterized robustly at cellular and molecular (transcriptionally and protein) and functional levels to support the claims made by the authors. The revised manuscript remains descriptive or predictive, with a very limited amount of validated experimental data. Too reliant on one single set of experiments.

My two major concerns remain are -

1. In the absence of robust characterization, the whole study is solely dependent on the poor characterization of these cardiac cell populations at the transcription level and that is not very reliable considering scRNA-seq studies require tissue and cell dissociations causing loss of cell identity. In addition, the authors have used samples from 9 to 16 PCW. Considering dynamic changes in embryonic gene expression during the early stages of cardiac development it is very difficult to determine whether they are novel cell populations, or the diversity seen in the cardiac cell population is due to the different stages used in the analysis.

2. The data relating to Semaphorin-Plexin signaling to direct ventricular cardiomyocyte organization is still superficially described. The authors show that Sema3C is expressed only in cardiac fibroblasts. Authors demonstrate that SEMA3C/D originates from the intermediate-LV CC and influences the spatial re-allocation of PLXNA2/A4+ trabecular vCMs. However, the data presented in the manuscript does not support this conclusion as SEMA3C/SEMA3D and PLXNA2/A4 expression can be seen throughout the heart both in compact and trabecular layers. My sincere concern is that authors are trying to align the computation data generated by sc-RNA transcriptomics on developmentally variable (9-16 PCW) cardiac samples with unbiased protein expression data and they are not supportive of the model/mechanism presented.

Referee #3 (Remarks to the Author):

This reviewer thanks the authors for being so receptive to the comments and suggestions provided. After revising their latest version of the manuscript and checking their results, I am happy to say that they have managed to address all my concerns.

The manuscript reads much better and their analysis has improved considerably. I have no further comments.

Author Rebuttals to Second Revision:

Referees' comments:

Referee #2 (Remarks to the Author):

As previously mentioned, multiple recent studies have been performed on single-cell RNA-sequencing to transcriptionally define distinct cell populations of the human heart. In the context of these published works, there is no novelty or robustness of the data presented in the current manuscript. The new cardiac cell populations identified in the revised manuscript should be at least characterized robustly at cellular and molecular (transcriptionally and protein) and functional levels to support the claims made by the authors. The revised manuscript remains descriptive or predictive, with a very limited amount of validated experimental data. Too reliant on one single set of experiments.

We thank the Referee for their review of our manuscript.

My two major concerns remain are -

1. In the absence of robust characterization, the whole study is solely dependent on the poor characterization of these cardiac cell populations at the transcription level and that is not very reliable considering scRNA-seq studies require tissue and cell dissociations causing loss of cell identity. In addition, the authors have used samples from 9 to 16 PCW. Considering dynamic changes in embryonic gene expression during the early stages of cardiac development it is very difficult to determine whether they are novel cell populations, or the diversity seen in the cardiac cell population is due to the different stages used in the analysis.

We appreciate Referee 2's comment and concern about scRNA-seq studies requiring tissue and cell dissociations. Thus, we have used high-resolution MERFISH on intact tissue sections to spatially map cell populations from our scRNA-seq and confirm their identities. As a result, we have discovered specific cell populations located in distinct regions of the developing heart that may contribute to the dynamic events of heart development. Additionally, as recommended by Referee 3 in previous reviews, we have included a comparative analysis of our data with those from publicly available heart cell atlases in order to assess the novelty of the cell populations identified from our studies. As such, we identified cell populations that appear to be specific for the developing heart (Supplementary Fig. 8 – previously Extended Data Fig. 9), and include those which represent developmental cell states for certain cell types (i.e., multiple potential cell states of developing ventricular cardiomyocytes – Supplementary Figs. 3, 8 – previously Extended Data Figs. 3, 9) as well as cell types specific for the developing heart (i.e., M20 cell – dorsal mesenchymal protrusion/DMP-like cell which mainly appears at 9 PCW - Supplementary Figs. 4, 8 – previously Extended Data Fig. 4, 9).

2. The data relating to Semaphorin-Plexin signaling to direct ventricular cardiomyocyte organization is still superficially described. The authors show that Sema3C is expressed only in cardiac fibroblasts. Authors demonstrate that SEMA3C/D originates from the intermediate-LV CC and influences the spatial re-allocation of PLXNA2/A4+ trabecular vCMs. However, the data presented in the manuscript does not support this conclusion as SEMA3C/SEMA3D and PLXNA2/A4 expression can be seen throughout the heart both in compact and trabecular layers. My sincere concern is that authors are trying to align the computation data generated by sc-RNA transcriptomics on developmentally variable (9-16 PCW) cardiac samples with unbiased protein expression data and they are not supportive of the model/mechanism presented.

We appreciate Referee 2's comments about our Plexin-Semaphorin findings in the ventricular wall. While SEMA3C/SEMA3D and PLXNA2/A4 may be observed within the ventricular walls, they are expressed in a complementary gradient (Extended Data Fig. 11d, e – previously Extended Data Fig. 28d, e). PLXNA2/A4 expression is higher in the Inner-LV CC trabecular layer but lower in Outer/Intermediate-LV CCs. However, SEMA3C/D is lower in the Inner-LV CC but higher in Outer/Intermediate-LV CCs. These expression gradients closely correlate with the spatial gradients of cells expressing these respective ligands/receptors (e.g., SEMA3C/D+ compact vFibro and PLXNA2/4+ trabecular vCMs - Fig. 4g, h, Extended Data Fig. 11a, b – previously Extended Data Fig. 28a, b). This gradient of expression of SEMA ligands is consistent with other models of secreted semaphorins displaying similar gradients of expression for directing cortical neuron migration and axon guidance (PMID: 18059265, PMID: 22368082). Supporting our model of how these Plexins and Semaphorins may interact to direct ventricular wall morphogenesis, we have provided both *in vitro* human pluripotent stem cell and *in vivo* mouse genetic studies (Figure 5, Extended Data Fig. 12, Supplementary Figs. 18, 19). In addition to providing the aforementioned data, we have also included in the Discussion that additional *in vivo* studies in the future will be interesting to further support and elucidate underlying cellular mechanisms for how Plexins and Semaphorins may regulate ventricular wall morphogenesis.

Referee #3 (Remarks to the Author):

This reviewer thanks the authors for being so receptive to the comments and suggestions provided. After revising their latest version of the manuscript and checking their results, I am happy to say that they have managed to address all my concerns.

The manuscript reads much better and their analysis has improved considerably. I have no further comments.

We again thank Referee 3 for their helpful suggestions, which have strengthened the findings of our manuscript.